# A single-particle mechanofluorescent sensor

Narges Ahmadi [1], Jieun Lee [1], Chirag Batukbhai Godiya[1], Jong-Man Kim [2] ✉ & Bum Jun Park [1] ✉

Monitoring mechanical stresses in microchannels is challenging. Herein, we report the development of a mechanofluorescence sensor system featuring a fluorogenic single polydiacetylene (PDA) particle, fabricated using a co-flow microfluidic method. We construct a stenotic vessel-mimicking capillary channel, in which the hydrodynamically captured PDA particle is subjected to controlled fluid flows. Fluorescence responses of the PDA particle are directly monitored in real time using fluorescent microscopy. The PDA particle displays significant nonlinear fluorescence emissions influenced by fluid viscosity and the presence of nanoparticles and biomolecules in the fluid. This nonlinear response is likely attributed to the torsion energy along the PDA's main chain backbone. Computational fluid dynamic simulations indicate that the complete blue-to-red transition necessitates ~307 μJ, aligning with prior research. We believe this study offers a unique advantage for simulating specific problematic regions of the human body in an in vitro environment, potentially paving the way for future exploration of difficult-to-access areas within the body.

Stenosis arising from substantial plaque deposition can locally increase blood flow rates and impart severe mechanical shear stress in narrowed blood vessel walls[1,2]. As a result, plaque can rupture or break down to produce small particles or emboli, which can lodge downstream in brain blood vessels where they can provoke neurological symptoms or strokes. To quantitatively understand the effect of flow-induced stress on stenotic vessels, extensive experimental and computational studies have been conducted exploring the hemodynamics of blood flow in simulated blood vessels[3–6]. In particular, recent advances made in developing microfluidic control technologies have enabled facile and accurate measurements to be made on flowing blood[7,8]. For example, in the context of methods to probe stenotic vessels, Jain et al. developed a microfluidic approach that enables real-time monitoring of hemostasis and platelet formation in model stenosis arteriolar vessels[9]. Also, Costa et al. used a 3D printer-based method to construct stenosed coronary arteries and utilized these mimics to evaluate the effect of vessel geometry on platelet aggregation[10]. Moreover, Korin et al. utilized the abnormally high shear stress around narrowed regions of stenosed vessels as the basis for the design of shear-activated drug delivery of target nanotherapeutic agents[11].

Along with those aimed at understanding and applying hemodynamic phenomena in microfluidic channels that mimic stenotic vessels, a large effort has been given to the use of computational fluid dynamics (CFD) for quantification of flow patterns and flow-induced stresses under atherosclerosis conditions[3,12–18]. However, very few approaches have been developed to experimentally support the results of these simulations[14,19–21]. In particular, to the best of our knowledge studies that directly visualize and quantify flow-induced mechanical stress inside stenotic vessel-mimicking flow channels have not been reported. In the investigation described below, which is part of our continuing efforts aimed at developing polydiacetylene (PDA)-based smart sensors[22–28], we designed and fabricated a mechanical fluorescence sensor that smartly responds to flow-induced stress in microfluidic channels. The effort to develop this system was comprised of three parts, the first of which involved fabrication of highly conjugated PDA particles with a uniform size distribution by utilizing a co-flow microfluidic method (Fig. 1a) and assessing their physico-chemical properties and fluorescence (FL) responses. Note that the PDAs, in general, have unique optical properties associated with a blue-to-red color transition and becoming FL emissive when it is subjected to various external physical, chemical, and biochemical stimuli[29–37]. In

[1]Department of Chemical Engineering (BK21 FOUR Integrated Engineering Program), Kyung Hee University, Yongin-Si, Gyeonggi-do 17104, South Korea. [2]Department of Chemical Engineering, Hanyang University, Seoul 04763, South Korea. ✉e-mail: jmk@hanyang.ac.kr; bjpark@khu.ac.kr

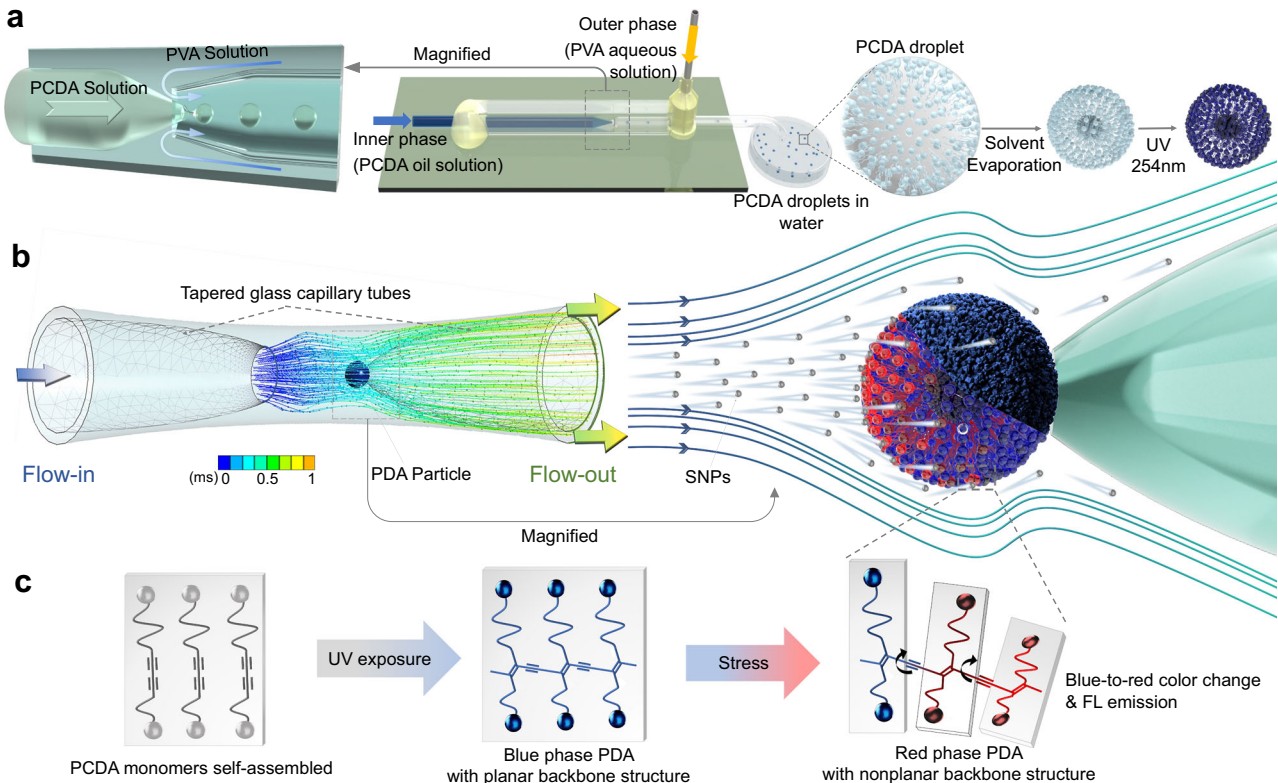

**Fig. 1 | Schematic overview of the experimental procedure. a** Fabrication of polydiacetylene (PDA) particles using the co-flow microfluidic method. The inner phase consists of 10,12-pentacosadiynoic acid (PCDA) monomers dissolved in chloroform, and the outer phase is an aqueous polyvinyl alcohol (PVA) solution. The magnified region on the left shows the generation of oil-phase PCDA droplets at the inlet orifice. After collection, the PCDA droplets are dried by solvent eva-poration, followed by photopolymerization under 254 nm UV exposure. **b** Operation of the single particle mechano-fluorescence (FL) sensor system showing silica nanoparticles (SNPs) bombardment on a captured PDA particle in a stenotic vessel-mimicking microchannel and the subsequent blue-to-red transition of stressed portions. The color bar denotes the residence time of SNPs within the channel. **c** Formation of blue-colored PDA from self-assembled diacetylene (DA) monomers and subsequent blue-to-red color transition and FL emission of the PDA particle under external stimuli.

the second part of the study, we constructed a prototype of a single PDA particle mechano-FL sensor system that has a narrow geometry like those of stenotic vessels (Fig. 1b). We elucidated the effect of flow-induced mechanical stress on the FL response of this sensor (Fig. 1c) and assessed its dependency on multiple experimental variables such as flow rate, solution viscosity, and the concentration of solid objects (i.e., silica nanoparticles (SNPs), red blood cells (RBCs) and yeast cells). We identified the key variable conditions using SNPs and subsequently applied these to the experiments with RBCs and yeast cells. Lastly, CFD simulations were performed to identify the relationship between the amount of mechanical energy absorbed by the PDA particle and the experimental FL intensity change.

## Results

### Fabrication of the PDA particle and evaluation of its FL response to stresses

The monodispersed stimulus-responsive fluorogenic PDA particle employed in this effort was fabricated using the co-flow microfluidic method. In the process, chloroform droplets containing 10 wt.% 10,12-pentacosadiynoic acid (PCDA) monomers were periodically generated along the flow of an aqueous outer phase (Fig. 1a and Supplementary Fig. 1)[38,39]. Typically, by using flow rates of the aqueous and chloroform phases of 400 μL·min⁻¹ and 60 μL·min⁻¹, respectively, micro-emulsion droplets having a mean diameter of ~400 ± 8.07 μm (Fig. 2a) were gen-erated. The PCDA-chloroform droplets were collected in a 2 wt.% poly-vinyl alcohol (PVA) water-filled petri dish that was subjected to chloroform removal by standing under ambient conditions for 12 h. The dried PCDA particles in water were then photopolymerized using a hand-

held laboratory UV lamp emitting 254 nm at an intensity of 0.4 mW·cm⁻². During this 10 min process, the container was gently shaken to ensure uniform UV exposure and effective photopolymerization. As a result, dark blue PDA particles with a uniform size distribution were produced, having a mean diameter of $2R \approx 200 \pm 2.08$ μm, as shown in Fig. 2b. Upon inspection of scanning electron microscope (SEM) images, it is apparent that the surface of the PDA particles features fuzzy domains, as depicted in Fig. 2c. This characteristic morphology has a resemblance to the structure of giant lipid/PDA vesicles, prepared using a simple solvent evaporation method[40].

Importantly, due to the finite penetration depth of the UV irra-diation, the PDA particles underwent partial polymerization[41]. As evi-dent from the confocal images of the red phase PDA particles in Fig. 2d, the polymerized shell is clearly visible with an approximate thickness of $h_{shell} \approx 10.0 \pm 0.03$ μm, as averaged over nine different particles. This corresponds to a maximum volume degree of polymerization of ~27%. Furthermore, SEM analyzes of bisected PDA particles reaffirmed the partial polymerization of the PDA particles. The interior of the bisected particle appeared to be filled with a solid substance before the chloroform treatment (Fig. 2e) but appeared hollow after the treat-ment (Fig. 2f). Given that PCDA monomers are soluble in chloroform, these results demonstrate that the center region of the particle remained unpolymerized and contained residual PCDA monomers.

The characteristic FL responses of the generated PDA particles to external stresses were evaluated prior to creating the mechano-FL sensor system. The thermo-FL behavior of the PDA particles was monitored using a FL microscope equipped with a thermal stage. As shown in Fig. 2g, the normalized FL intensity averaged over several

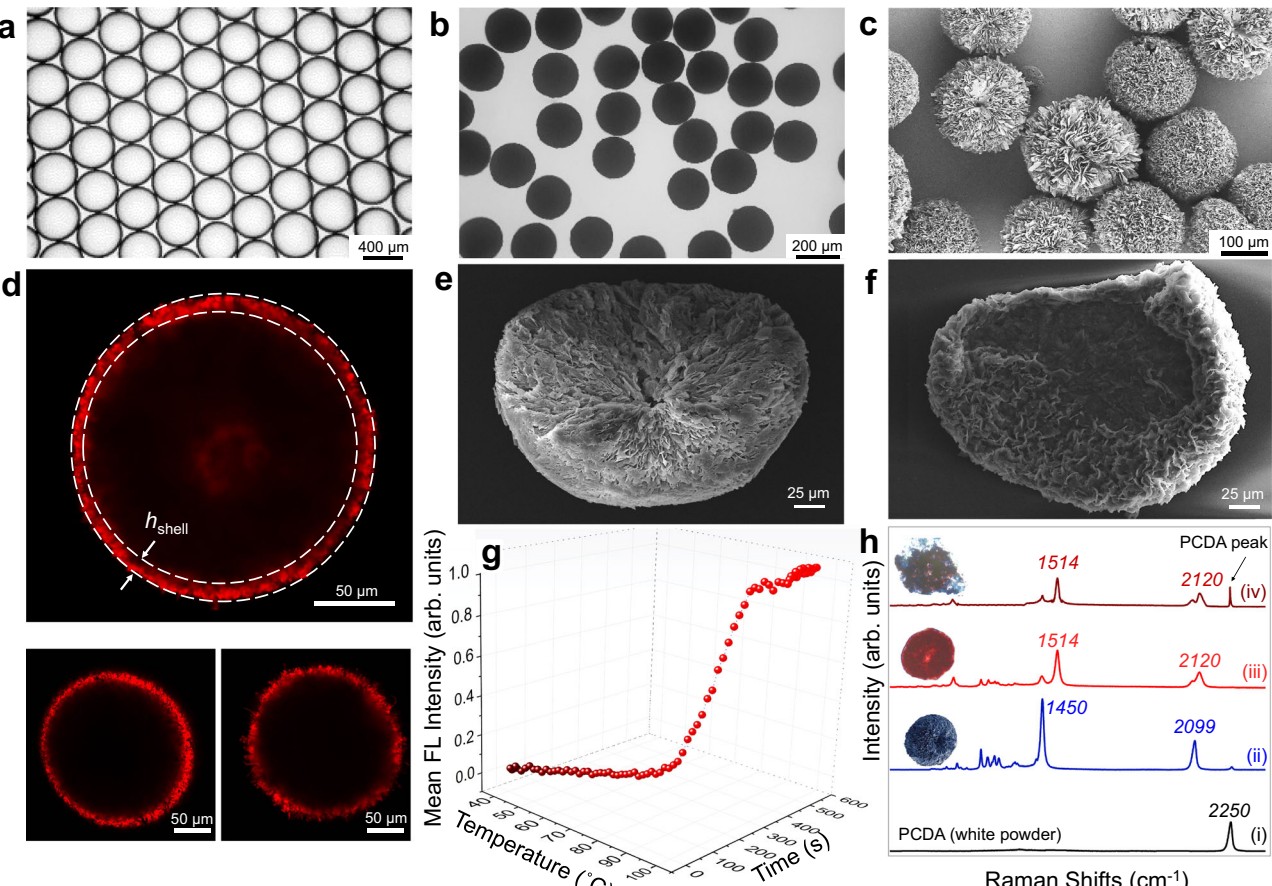

**Fig. 2 | Structural and optical properties of PDA particles. a, b** Optical microscopic images of PCDA-chloroform emulsions in water (**a**) and solid PDA particles in water (**b**). **c** Scanning electron microscope (SEM) image of PDA particles. **d** Confocal microscope image of a red phase PDA particle heated at 80 °C for 10 min, revealing a polymerized shell of a thickness $h_{shell}$. Two additional images from different particles are shown below. **e, f** SEM images of a bisected PDA particle before (**e**) and after (**f**) chloroform treatment. **g** Thermo-FL response of PDA particles subjected to gradual heating from 40 °C to 100 °C. **h** Raman spectra of PCDA monomers (i), a blue PDA particle (ii), a red PDA particle produced by heating at 80 °C for 10 min (iii), and a mechanically crushed PDA particle (iv).

PDA particles nonlinearly increases as the incubation temperature increases from 40 to 100 °C at a rate of 10 °C·min⁻¹. A minor intensity increase occurs in the ~40 to ~70 °C range and then a steep increase occurs when the temperature increases from ~70 to ~90 °C at which the intensity plateaus and the PDA blue-to-red color transition is complete. Note that the FL intensity barely exhibit significant changes after the normalized FL intensity reaches ~1. This is because the PDA response occurs irreversibly, meaning that beyond this point, additional stresses unlikely result in a substantial change in the FL response. The observed thermally induced-FL response matches that observed in a previous study where a PDA powder underwent a significant colorimetric transition over the temperature range of ~65–100 °C[42,43]. The nonlinear FL intensity response to temperature suggests that a critical energy point exists for creating the level of distortion of the PDA backbone structure required for substantial FL emission.

The blue-to-red color transition of the PDA particles is a consequence of conformational changes within the highly conjugated polymer backbone[44–47]. In general, the blue phase represents a PDA state with a planar backbone structure, whereas the red phase exhibits a twisted, nonplanar geometry (Fig. 1c). Interestingly, some research has challenged the initial belief that disorder is a necessary condition for the red phase formation in PDA[48,49]. Raman spectroscopy was used to confirm that the generated PDA has a π-conjugated PDA backbone and evaluate changes taking place upon formation and stressing the PDA particles[32]. As can be seen in the spectrum displayed in Fig. 2h and Supplementary Fig. 2, the acetylenic (C ≡ C) stretching band of PCDA

monomer is located at 2250 cm⁻¹, and the bands associated with C ≡ C and C = C stretching in the alternating ene-yne backbone at 2099 cm⁻¹ and 1450 cm⁻¹, respectively, arise following polymerization. Thermal perturbation and crushing of the PDA lead to shifts of the alkyne and alkene bands to higher frequencies (2120 cm⁻¹ and 1514 cm⁻¹, respectively), a pattern that has been observed in a variety of PDAs subjected to stresses[22,43]. Note that the PCDA monomer peak observed in the crushed sample consistently demonstrates the presence of unpolymerized PCDA within the PDA particle. The typical mechanochromism of the PDA particles, manufactured through the microfluidic method, was further confirmed by subjecting them to common mechanical stresses, such as shearing, poking, and pressing. Each of these stresses induced FL emission, as shown in Supplementary Fig. 3.

## Optimization of the mechano-FL sensor system

In the next phase of this investigation, we developed a single PDA particle mechano-FL sensor system having an optimized channel geometry, fluid flow rate, and exposure time to flows. When pure water at a flow rate of 25 mL·min⁻¹ (maximum tolerated by experimental setup) was introduced into a blue PDA particle-containing device without or with an injection tube (Supplementary Fig. 4a, b), an observable FL response did not take place. In a study aimed at uncovering a mimic of blood components (e.g., RBCs) in a model of the vascular stenosis environment, a FL response arose when water containing 16.96 wt.% silica nanoparticles (SNPs, which are smaller than RBCs) was introduced to a device incorporating a narrow outer tube at a rate of 25 mL·min⁻¹ for

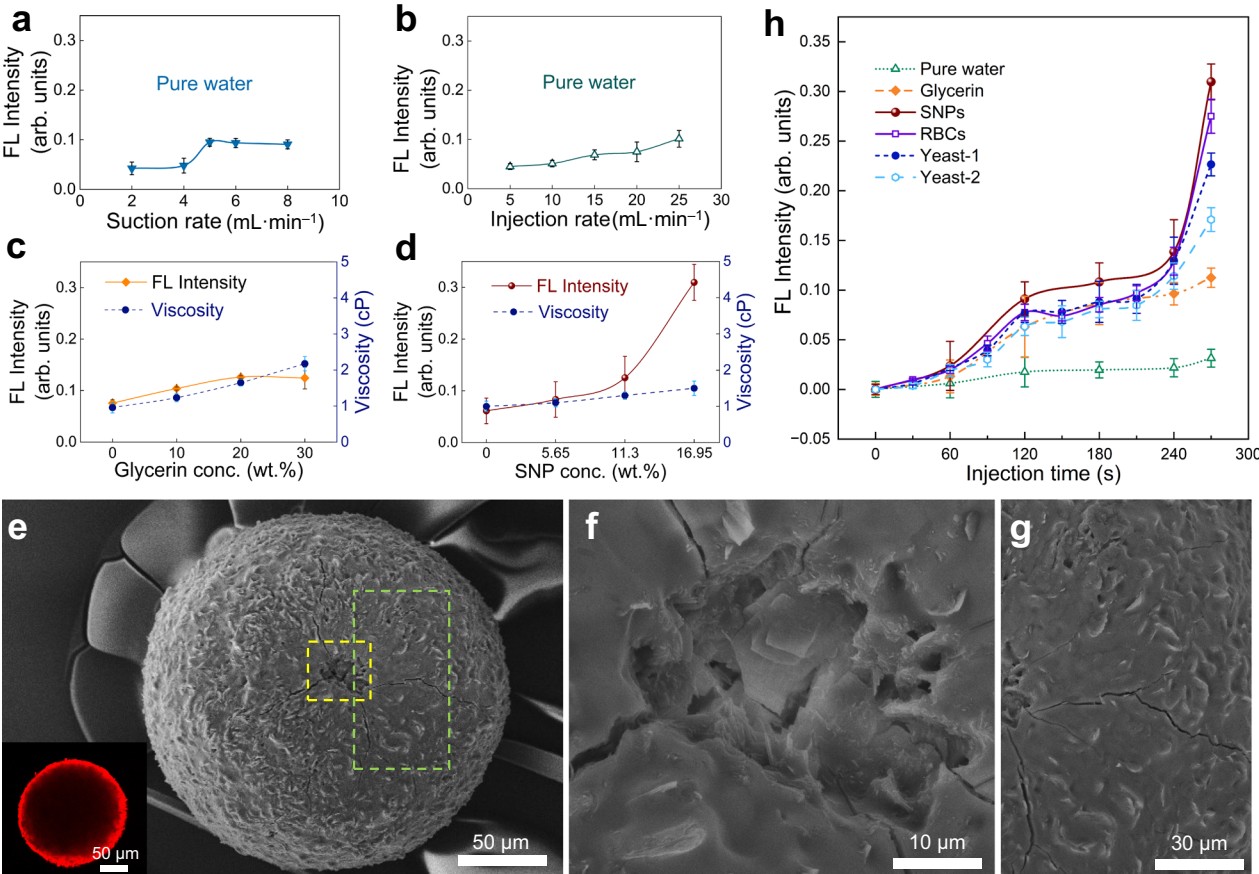

**Fig. 3 | Evaluation of the mechano-FL sensor system. a–d** Normalized mean FL intensities of the PDA particles as a function of suction rate (**a**), injection rate (**b**), glycerin concentration (**c**), and SNP concentration (**d**). In each case, parameters were 4.5 min injection time, 25 mL·min⁻¹ injection rate, and 5 mL·min⁻¹ suction rate, except when used as variables. **e** SEM image of a PDA particle after SNPs bombardment for 4.5 min and the inset is the corresponding confocal fluorescence microscopic image. **f**, **g** Magnified SEM images of the regions indicated by the yellow (**f**) and green (**g**) dashed boxes of the image in panel (**e**). **h** FL emissions of PDA particles under the given experimental conditions depending on flow injection time. The injection and suction rates were 25 mL·min⁻¹ and 5 mL·min⁻¹, respectively. The injection time refers to the period of a PDA particle exposed to fluid flows for a given time. The data points and error bars in panels (**a–d**) and **h** indicate the mean FL intensity values and standard deviations from at least three independent experiments, respectively.

an exposure time (flow injection time) of 4.5 min (270 s) and a suction rate of 5 mL·min⁻¹ to retain the PDA particle (Supplementary Fig. 4c and Supplementary Fig. 5). In addition, we observed that simple incubation of the PDA particles in stationary aqueous mixtures containing commercially available LUDOX TM-40, LUDOX HS-30, LUDOX AM-30, and LUDOX LS-30 resulted in detectable FL intensities (Supplementary Fig. 6a, b and Supplementary Table 1). This effect is likely caused by the presence of anionic components in the mixtures, which promote a pH triggering PDA response[50–53]. The use of a 20 mM phosphate buffer significantly reduced the FL intensity of the PDA particles, with values in the range of 0.0098 – 0.0132 arb. units for all types of SNPs, which were sufficiently minor (Supplementary Fig. 6c and Supplementary Table 1). Consequently, we chose to use LUDOX TM-40 (diameter ~19.6 ± 5.6 nm, Supplementary Fig. 7), which displayed the relatively lowest FL intensity in the buffer environment. However, the use of other types of SNPs would not have resulted in any significant differences in the mechano-FL sensor experiment. Additionally, before proceeding with the mechano-FL sensor experiments, we confirmed that all the solutions used in the experiments did not induce significant changes in the FL emission of the PDA particles under static conditions.

### Evaluation of the vascular mimicking mechano-FL sensor system

In the designed vascular mimicking device, a single PDA particle is immobilized in the tip of the holding tube by using a suction syringe pump to create a negative pressure to counteract the fluid flow. Thinking that a strong negative pressure created by the suction system might cause mechanical stress, especially by pushing the round-shaped tip of the holding tube into the PDA (Supplementary Fig. 4c) and a consequent enhancement of FL intensity. However, we found that injection of pure water at a rate of 25 mL·min⁻¹ for a flow injection time of 4.5 min, using suction rates of 2, 4, 5, 6, and 8 mL·min⁻¹, has a negligible effect ( < 0.1 arb. units) on the FL intensity (Fig. 3a). Therefore, we selected 5 mL·min⁻¹ as the optimized suction rate in the system that was subjected to further investigation.

The injection flow rate was another parameter that we anticipated could influence FL emission intensity response of the immobilized PDA particle. To probe this feature, pure water was introduced into the mechano-FL sensor system at rates of 5, 10, 20, and 25 mL·min⁻¹ (constant 5 mL·min⁻¹ suction rate), a flow rate range that is similar to the blood flow rate of 3–26 mL·min⁻¹ in 0.8–1.8 mm arteries in the human finger[54]. After the 4.5 min injection time, the mean FL emission intensity gradually increases from 0.045 to 0.101 arb. units as the injection rate is increased (Fig. 3b). These observations suggest that flow-induced mechanical stress on the PDA is directly proportional to the flow rate. However, similar to the suction rates, the injection rates in the range explored only negligibly affect FL emission and, thus, 25 mL·min⁻¹ was used as the optimized injection rate in subsequent studies.

The effect of solution viscosity on the PDA FL response was assessed next. For this purpose, aqueous solutions containing 10, 20,

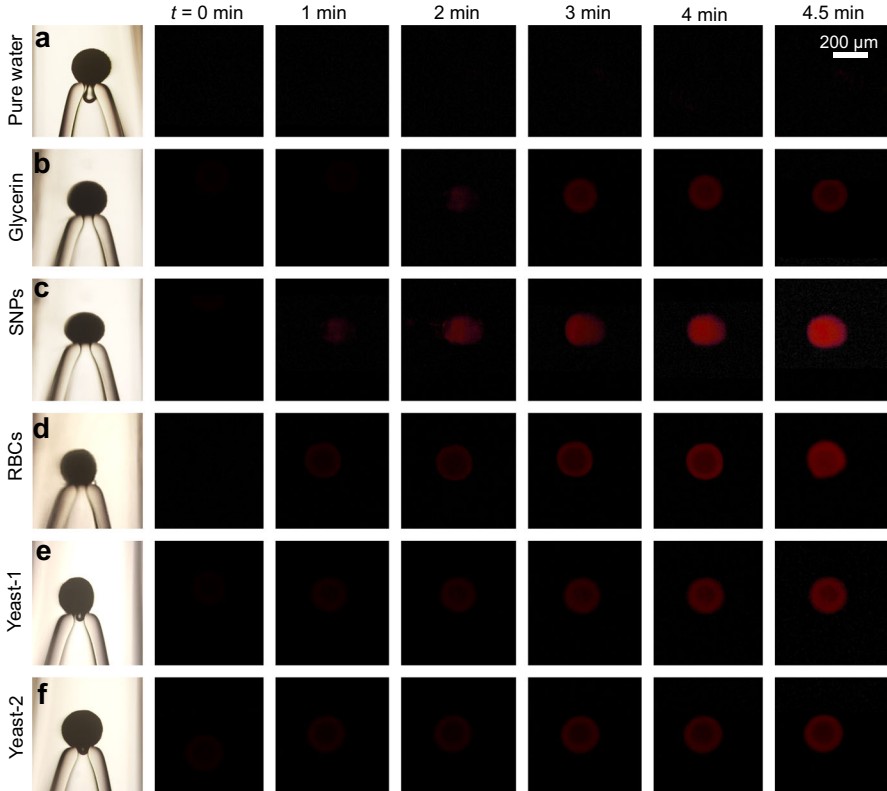

**Fig. 4 | FL images of the PDA particle in the mechano-FL sensor system.**
**a**–**d** Representative FL images illustrating the variation in PDA particle FL emission upon injecting different types of solutions: pure water (**a**), 20 wt.% glycerin (**b**), 16.95 wt.% SNPs (**c**), $1.54 \times 10^5 \, \mu L^{-1}$ red blood cells (RBCs) (**d**), $4.80 \times 10^4 \, \mu L^{-1}$ yeast (**e**), and $2.40 \times 10^4 \, \mu L^{-1}$ yeast (**f**), corresponding to the results depicted in Fig. 3h. The images were captured at an injection rate of 25 mL·min$^{-1}$ and a suction rate of 5 mL·min$^{-1}$, with the flow injection time varied.

and 30 wt.% glycerin were injected at an injection rate of 25 mL·min$^{-1}$ for an injection time of 4.5 min using a suction rate of 5 mL·min$^{-1}$. As demonstrated by the plot in Fig. 3c, injection of 20 and 30 wt.% results in relatively strong FL emissions of about $0.1262 \pm 0.0025$ arb. units and $0.1245 \pm 0.0213$ arb. units, respectively. Given that the viscosities of these solutions are proportional to the glycerin concentration, the similarities of these FL intensity responses might possibly result from the fact that both fluids do not cause sufficient mechanical energy in the PDA backbone to reach the critical value needed to promote a substantial FL change.

Lastly, we explored the effects of SNP and biomolecule (i.e., RBCs and yeast cells) concentrations on the FL response of the PDA particle, expecting that higher mechanical stress would be imparted when higher concentrations are used. First, we assessed the FL intensity changes brought about by using aqueous SNP suspensions with concentrations of 0, 5.65, 11.3, and 16.95 wt.%, while keeping the other parameters constant (i.e., injection rate of 25 mL·min$^{-1}$, suction rate of 5 mL·min$^{-1}$, and flow injection time of 4.5 min). It should be noted that we used the 5.65 and 11.3 wt.% SNP suspensions because their viscosities are the same as those of respective 10 and 20 wt.% aqueous glycerin solutions, thereby enabling us to separate the effects of particle concentration and viscosity. As shown in Fig. 3d, the FL intensity drastically increases with increasing SNP concentrations, and it reaches $0.3096 \pm 0.01802$ arb. units at 16.95 wt.%. The results show that the FL response is more greatly governed by SNP concentration (Fig. 3c, d). The PDA particle was isolated after bombardment by 16.95 wt.% SNPs for 4.5 min. As can be seen in the image in Fig. 3e–g, local indentations are present mainly at the center region of the PDA particle, where it should be most heavily bombarded by the SNPs. Also, the images show that several cracks radiate outward from this central area. Interestingly, numerous fuzzy domains present on the intact PDA

particle, as shown in Fig. 2c, appear to have been mechanically etched as a consequence of mechanical stress applied to the PDA surface during the SNPs injection. Finally, the confocal image shown in the inset of Fig. 3e reveals that the entire surface region of the PDA particle fluoresces after injection of the SNPs for 4.5 min. This illustrates how the localized mechanical stress, when continuously applied and accumulated, can affect the entire region of the PDA particle.

The time-dependent response of the mechano-FL sensor system was examined by continuously monitoring FL intensities at various time points following injection (Figs. 3h and 4). We found that as the flow injection time increases the consequent FL intensity increases. For example, in contrast to the negligible effect caused by pure water (Fig. 4a) and relatively small FL intensity changes for the case of 20 wt.% glycerin (Fig. 4b), injection of 16.95 wt.% SNP promotes the most prominent time dependent FL intensity increases. Interestingly, as the SNP injection time is increased, the area of FL emission also gradually increases (Fig. 4c). Given that the impact location of SNPs remains constant throughout the injection, the continuous SNPs collision likely results in accumulated mechanical stress, causing the entire particle to fluorescence. Notably, using the 16.95 wt.% SNP solution, the FL intensity increases nonlinearly (Fig. 3h).

We conducted further experiments to elucidate the effect of physical impact of SNPs on PDA particles on FL emission. The time-dependent FL response at different SNPs concentrations (5.65, 11.3, and 16.95 wt.%) was compared, as shown in Supplementary Fig. 8a. It was observed that the FL intensity value rapidly increased with time at higher concentrations, while the change was relatively minor at lower concentrations. The FL intensity change was then re-plotted against the number of injected SNPs, resulting in the data for the three different concentrations converging to a single master curve (Supplementary Fig. 8b). Additionally, we extended the injection time of

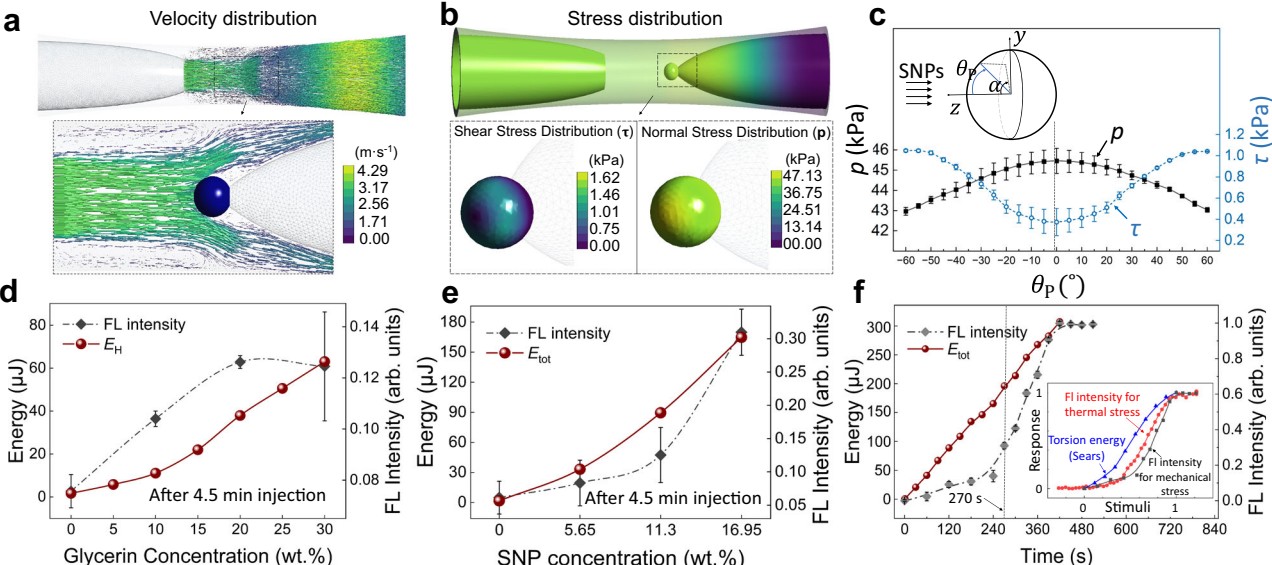

**Fig. 5 | Computational fluid dynamics (CFD) analysis of the mechano-FL sensor system. a, b** Examples showing distributions of flow velocity (**a**) and stress (**b**) for the 16.95 wt.% SNP suspension. **c** Variation of shear and normal stresses on the PDA surface as a function of the polar angle $\theta_P$. Error bars represent the standard deviation over the stress values across the azimuthal angle $\alpha$. **d, e** Plots of $E_H$ and $E_{tot}$ as functions of glycerin (**d**) and SNPs (**e**) concentrations, respectively, at an injection rate of 25 mL·min$^{-1}$ over 4.5 min. Corresponding experimentally measured FL intensity profiles are overlaid for comparison. **f** Time-dependent profiles of $E_{tot}$

and FL intensity for the 16.95 wt.% SNP suspension injected at 25 mL·min$^{-1}$ over an extended duration of 8.5 min. Inset indicates the comparison of the nonlinear property of normalized FL intensity profiles from the mechano-FL sensor experiment and the thermal stress analysis (from Fig. 2g) against the normalized torsion energy $E_{tor}$ from a previous study[56]. The $x$-axis is normalized from 0 to 1, aligning maximum values of FL intensity and $E_{tor}$ to 1. For panels **d–f** each experimental data point and corresponding error bar represent the mean intensity value of a minimum of three independent trials and its standard deviation, respectively.

16.95 wt.% SNPs to 8.5 min (510 s) and found that the normalized FL intensity approached 1, indicating near saturation (Supplementary Fig. 9). Moreover, a 16.95 wt.% SNPs solution was injected and halted at 4.5 min (270 s), and the FL intensity was continuously monitored even after stopping the injection. It was found that the FL change after stopping was negligible (Supplementary Fig. 10). Based on these additional analyzes, we confirmed that the FL intensity values are primarily influenced by the number of SNPs colliding with the PDA particles. Furthermore, the gradual changes in PDA mechanochromism, along with the halt of FL intensity changes upon the removal of mechanical stress, underscore the important role that continuous application and accumulation of mechanical stress play in mechanochromism. This might be closely related to the nonlinear behavior of PDA mechanochromism[55,56]. For more information on the injected fluids used in this study, one can refer to Supplementary Table 2.

**Effect of biomolecule collisions on PDA FL response**

Interestingly, injection of biomolecules (Supplementary Fig. 11a, b) also promotes similar FL responses. For these experiments, we used ovine RBCs because they are commercially available, and they have a biconcave morphology and size (~4 μm in ovine and ~7 μm in humans) that are similar to human RBCs, and a low aggregability[57]. After injecting ovine RBCs with a number density of ~1.54 × 10$^5$ μL$^{-1}$ (Supplementary Table 3) for 4.5 min under the standard conditions, we observed a FL intensity change of ~0.2751 ± 0.0171 arb. units (Fig. 3h). This change is significantly stronger than that promoted by 30 wt.% glycerin (Fig. 3c). Specifically, given that the viscosity of the RBC solution (~2.5 cP, Supplementary Fig. 11c) is comparable to that of the 30 wt.% glycerin solution, the ~3 times greater intensity change promoted by the RBCs (Figs. 3h and 4d) is directly related the collisions of solid objects, even when soft, with the PDA particle. Note that these observations were made using conditions that were comparable to present in human blood where the RBC number density is ~5 × 10$^6$ μL$^{-1}$ and the viscosity is ~3.5–5.5 cP[58]. Additional experiments using yeast cells yielded similar results. Injection of solutions of yeast cell with

concentrations of ~4.8 × 10$^4$ and ~2.4 × 10$^4$ μL$^{-1}$ (Supplementary Table 3) into the mechano-FL sensor system caused FL intensity responses of ~0.2265 ± 0.01147 and ~0.171 ± 0.012 arb. units (Figs. 3h and 4e, f), which are also greater than those promoted by glycerin solutions. Here again, because the viscosities of the two yeast solutions are ~1.3 and ~1.2 cP (Supplementary Fig. 11c), the collision effect clearly governs the response of this mechano-FL sensor system. Similar nonlinear responses were also found using the biomolecules (Fig. 3h), consistently demonstrating that a critical energy level exists for sufficient distortion of the PDA backbone to create a significant FL response[55,56].

**CFD Simulations of the mechano-FL sensor system**

We utilized CFD simulations to formulate a rational interpretation of the experimental observations described above (Supplementary Fig. 12). For this purpose, calculations were conducted to quantitatively determine the total energy $E_{tot}$ through hydraulic pressure and SNP collision absorbed by a PDA particle. The $k$-omega shear stress transport ($k$–$\omega$ SST) turbulence model was utilized because of its excellent capability of handling flow transition from laminar to turbulent regimes, and to accurately account for strong pressure gradients near walls (see the method section for more detailed explanation)[59,60]. The dimensions of the mechano-FL sensor system used in experiments were employed in the CFD simulation (Supplementary Fig. 13), and detailed descriptions of equations and modeling parameters are provided in Supplementary Tables 4 and 5.

The results of these calculations clarify the dynamic behavior of the mechano-FL sensor system, as illustrated in Fig. 5a–c. Specifically, Fig. 5a provides a detailed visual representation of the flow velocity patterns observed when injecting a 16.95 wt.% SNP solution with a flow velocity of 3.317 m·s$^{-1}$ (Supplementary Table 2). The maximum flow velocities are found to be approximately 3 m·s$^{-1}$ in the microchannel following impact with the PDA particle surface. Figure 5b visualizes the corresponding shear and normal stress distributions. Notably, the normal stress peaks at a polar angle of $\theta_P = 0°$, which is the point where

the SNPs collide perpendicularly with the PDA particle, as depicted in Fig. 5c. In contrast, the shear stress reaches a minimum value at the same polar angle ($\theta_P = 0°$) and increases as the angle $\theta_P$ widens. Note that the stress scale obtained in these calculations is comparable to the scales obtained through the surface forces apparatus for PDA Langmuir films upon the blue-to-red transition (i.e., -100 kPa for normal stress and -3 kPa for shear stress)[61].

The hydraulic energy $E_H$ absorbed by the PDA particle is proportional to the viscosity of the fluid. As depicted in Fig. 5d, the upward trend in calculated $E_H$ values as the glycerin concentration increases from 0 to 30 wt.% is consistent with a typical characteristic of Newtonian fluid flows. The summary of fluid properties with varied viscosities is presented in Supplementary Table 6. Note that the shape of glycerin concentration dependent $E_H$ profile does not completely match the corresponding experimentally determined FL intensity profile. This deviation likely arises because the conjugated structures of the PDA particle are not accounted for the CFD simulation. Additionally, the maximum calculated $E_H$ values for 20 to 30 wt.% glycerin are -50 and -62 µJ, respectively, whereas the experimental FL intensity only shows a negligible change upon injection of these solutions. This result suggests that the conjugated PDA backbone may resist structural distortion caused by the flow-induced mechanical stress.

The calculations show that bombardment by SNPs leads to an increase in $E_{tot}$ absorbed by the PDA particle, considered to be the sum of the computed $E_H$ and SNP collision energy $E_{col}$ calculated using the Hertz contact equation (Supplementary Table 7)[62]. $E_{tot}$ demonstrates an increasing trend as the SNP concentration increases, spanning 0, 5.65, 11.3, and 16.95 wt.% when injected for 4.5 min (Fig. 5e) and with an extended injection duration of 8.5 min at 16.95 wt.% concentration (Fig. 5f). Similarly, because the conjugated structure of the PDA was not considered in this simulation, the $E_{tot}$ values do not accurately reflect the observed nonlinear behavior of FL intensities dependent on the SNP concentration. Nevertheless, this computational data suggests that the $E_{tot}$ required to achieve a complete blue-to-red transition for a single PDA particle is approximately 307 µJ. The activation energy $E_{act}$ for irreversible blue-to-red transitions in a variety of PDAs has been estimated to be -17.6–22.5 kcal·mol⁻¹[46,47]. For instance, for a three-layer PDA film fabricated from the PCDA monomers, the predicted $E_{act}$ was approximately 17.6 kcal·mol⁻¹. Though our 200 µm PDA particles might differ from the previously reported PDA conditions, we presumed $E_{act} \approx 17.6–22.5$ kcal·mol⁻¹ to roughly estimate $E_{act}$ for our PDA particles. Considering -27% polymerization (Fig. 2d) yields $0.27 \times \frac{\rho_{PDA} \times V_{PDA}}{MW_{PCDA}} E_{act} = 289 - 369$ µJ, where the PDA density is $\rho_{PDA} \approx 1.3 \times 10^3$ kg·m⁻³, the PDA volume is $V_{PDA} = \frac{4\pi R^3}{3}$, and the molecular weight of PCDA is $MW_{PCDA} = 0.375$ kg·mol⁻¹. This result aligns well with the energy requirement of -307 µJ for the complete blue-to-red transition, as derived from our calculations. Additionally, we performed CFD calculations and experiments for different sizes of PDA particles. Given the polymerization degree of -74% for 50 µm and -37% for 140 µm PDA particles (Supplementary Fig. 14a), the energy required for the complete blue-to-red transition was -12.4 µJ and -141 µJ, respectively. These values are in good agreement with the predictions based on $E_{act}$ values (Supplementary Fig. 14b). Furthermore, it was observed that the blue-to-red transition occurred more rapidly as the size of the PDA particles decreased (Supplementary Fig. 14c–f), demonstrating sensitivity tunability for PDA particles.

The nonlinear FL intensity profile possesses a resemblance to the previously reported torsional energy $E_{tor}$ calculations, which also exhibit nonlinear increases for various types of PDA materials[55,56]. Therefore, we compared the normalized $E_{tor}$ profile with the normalized FL intensity profiles derived from both the mechano-FL sensor experiment (Fig. 5f) and the thermal stress analysis (Fig. 2g). As displayed in the inset of Fig. 5f, the experimental FL intensity's nonlinear trajectory seems fairly consistent with the shape of $E_{tor}$. This similarity allows us to reasonably postulate a close correlation between these two factors. Notably, recent experimental results have demonstrated that lateral stress applied parallel to the PDA backbone significantly influences the mechanochromism of thin PDA films[63]. Therefore, further experimental and/or theoretical studies are needed to clarify how torsional and lateral stresses contribute relative to each other depending on the PDA structures under various conditions.

## Discussion

Mechanical stress is a key factor in the incidence of plaque rupture in blood arteries and the formation of emboli, both of which lead to severe cardiovascular and neurological disorders. Despite extensive study in the area of hemodynamics to comprehend the impact of mechanical stress on stenotic vessels, visualizing and accurately quantifying flow-induced mechanical stress inside stenotic channels in real-time remains challenging. To address this, we performed a proof-of-concept study to demonstrate the potential of a single PDA particle-based mechano-FL sensor system for effectively analyzing and visualizing flow-induced mechanical stresses in stenotic vessel-mimicking microfluidic channels. Using uniformly sized PDA particles fabricated through the co-flow microfluidic method, we could directly monitor mechanical stress. Among various parameters, such as fluid velocity, viscosity, and the concentration of particulate objects, the most significant factor influencing the FL emission of the PDA particle was the physical impact of solid particles. Notably, we observed nonlinear behaviors in FL intensity. This behavior closely resembled the previously reported energy changes based on the torsion angle of the PDA backbone. Given these similarities, we hypothesized a direct link between the two phenomena. However, considering our experimental conditions, which utilized spherical particles with a polymerized layer of -10 µm composed of PCDA monomers, clear quantification of torsion energy and its relation to the mechanical energy applied to the PDA particle will be crucial in future studies.

We believe this study represents a significant advancement in the development of minimally invasive force/stress measurement in in vitro systems that simulate hard-to-reach spaces within the human body. However, we recognize that several aspects in our current system need refinement. For instance, positioning the PDA probe particle within a narrowed channel could influence fluid behaviors. As illustrated in Supplementary Fig. 15, one potential solution is to embed the PDA particles into simulated vessel walls using materials such as hydrogels or polydimethylsiloxane (PDMS). To more accurately simulate actual blood flows in stenosed vessels, a 3D-printed vascular stenosis model can be employed[10]. The sensitivity of the PDA particles as sensors could be enhanced by exploring different diacetylene monomers[43,64,65]. A more straightforward approach might involve adjusting the UV exposure time for polymerization, thus controlling the degree of PDA particle polymerization. Given the diverse application requirements, our microfluidic setup allows us the flexibility to produce PDA particles of different sizes, ranging from tens to hundreds of micrometers. This can be achieved by adjusting the flow rates fed into the microfluidic device or by altering the geometry of the capillary tubes constituting the device[66].

Lastly, the system we developed offers a promising experimental methodology to further understandings of the unique phenomena related to PDAs that remain elusive. For instance, we still lack a clear understanding of how mechanical stress, when applied locally, propagates through the PDA's conjugated system. When considering a linearly arranged PDA main chain backbone of finite length, it is unlikely to expect that mechanical energy introduced at one end would spontaneously propagate, like dominos, to the opposite end. This notion is evident from what we observed experimentally in Supplementary Fig. 10. Therefore, we surmise that for the energy applied locally to a PDA particle to spread across the entire region, it would

necessitate a consistent and accumulating external energy input. In our pursuit to understand the propagation of mechanical stress at micro and/or macroscopic scales, we plan to utilize optical laser tweezers apparatus[67–69]. This can optically trap a polymeric micro-particle and apply oscillatory impacts to the surface of a PDA particle anchored to a substrate. Given that optical laser tweezers can precisely control the applied force down to the piconewton scale, we believe that sufficient mechanical stress can be exerted on thin-layered PDA particles. Furthermore, we plan to initiate a single collision of a PDA particle (or other types of particles) to another PDA particle positioned within the mechano-FL sensor channel. By manipulating variables such as size, rigidity, and collision speed, we aim to compare the area where FL is emitted to the actual collision area.

## Methods

### Materials

GFS Chemicals (USA) provided the 10,12-pentacosadiynoic acid (PCDA) monomer. Colloidal silica nanoparticles (SNPs; LUDOX TM-40, LUDOX HS-30, LUDOX AM-30, LUDOX LS-30) and polyvinyl alcohol (PVA, molecular weight of 13000–23000 g/mol, 87–89% hydrolyzed) were purchased from Sigma-Aldrich (USA). Polydimethylsiloxane (PDMS, SYLGARD™ 184 silicone elastomer kit) was obtained from Dow Coring (USA). Chloroform ( > 99.5%, HPLC grade) was purchased from Thermo Fisher Scientific (USA). Glycerin (extra pure) was obtained from Duksan chemicals (Korea). Anhydrous monobasic sodium phosphate ($NaH_2PO_4$, 99% extra pure) and anhydrous dibasic sodium phosphate ($Na_2HPO_4$, 99% extra pure) were obtained from Daejung chemicals (Korea). Ultrapure water (resistivity >18.2 MΩ cm) was used for all aqueous phases. Defibrinated sheep blood was purchased from Biozoa Biological Supply (Korea), and yeast cells (Saccharomyces cerevisiae, type I) were acquired from Sigma-Aldrich (USA).

### Fabrication of the microfluidic device

The glass capillary microfluidic device (Supplementary Fig. 1) was constructed on a glass slide (76 × 52 mm, Matsunami Glass Ind. Ltd., Japan)[38,70]. One side of a glass capillary tube (World Precision Instruments, USA) with inner and outer diameters of I.D. = 0.58 and O.D. = 1.0 mm was tapered using a micropipette puller (Model P-1000, Sutter Instruments, USA). The tapered tip of the circular capillary tube (CC-1, length = 50 mm) was then cut and ground to a proper size (I.D. ≈ 400 μm) using a sandpaper. The other end region of CC-1 was gently heated and slightly bent. The tapered side of CC-1 was then inserted into a piece of a square capillary tube (SC-1) (I.D. = 1.05 mm, length = 20 mm, AIT glass, USA), which was placed on a glass slide (52 × 76 mm, Matsunami, Japan). A threaded hub dispensing needle (20 gauge, Weller, Germany) was positioned on a gap between SC-1 and CC-1, and sealed with 5 min epoxy glue (Devcon, USA). Another 50 mm long circular capillary tube (CC-2) was tapered to yield I.D. ≈ 130 μm and inserted into the SC-1 from the opposite side until the orifice of CC-2 slightly entered CC-1. Lastly, all gaps were completely sealed with epoxy glue and allowed to cure at room temperature for ~12 h.

### Fabrication of PDA particles using the co-flow microfluidic method

PDA particles were prepared utilizing the co-flow microfluidic capillary system (Fig. 1a and Supplementary Fig. 1)[38,70]. Sol.A was an aqueous solution containing 2 wt.% PVA, used as a stabilizer, and Sol.B was a 10 wt.% PCDA monomer solution, which consisted of 0.163 g PCDA dissolved in 1 mL of chloroform. Sol.B was filtered through a poly-tetrafluoroethylene syringe filter (pore = 0.45 μm, Whatman, USA) prior to usage. The prepared solutions were loaded into two gas-tight syringes, and separately delivered into the microfluidic device using two syringe pumps. Sol.A was injected through a needle positioned in the gap between CC-1 and SC-1, and Sol.B was added to CC-2. The injection rate of each fluid phase was carefully adjusted to ensure the stable generation of PCDA-chloroform droplets with a desired size and size distribution. Once the optimal conditions were identified, the flow rates used in this study were established as 400 μL·min⁻¹ for Sol.A and 60 μL·min⁻¹ for Sol.B. The produced PCDA-chloroform droplets were then directed along CC-1 and collected in a 2 wt.% PVA water-filled glass petri dish for 5 min. To facilitate the evaporation of chloroform from the individual droplets, they were stored under dark conditions for ~12 h, leading to the formation of solid PCDA particles. Photo-polymerization of the PCDA particles, dispersed in the water-filled petri dish, was conducted under 254 nm UV irradiation (1 W, VL-6LC, Vilber Lourmat, France) at a power density of 0.4 mW·cm⁻² for 10 min. During this process, the petri dish was gently shaken to ensure uniform irradiation to the particles. Finally, the obtained PDA particles were washed with water, and then stored in fresh water. The particles were kept under dark conditions at 4 °C until they were needed for sub-sequent mechano-FL experiments.

### Fabrication of the mechano-FL sensor device

A 50 mm long circular capillary tube (I.D. = 0.58 mm and O.D. = 1.0 mm) was used as a holding tube (Fig. 1b, Supplementary Figs. 4 and 5). Three device geometries were prepared. For the first geometry, one of the end regions of the holding tube was fire-polished to render a round surface with a hole (I.D. ≈ 100 μm). The rounded side was inserted into a piece of a square outer capillary tube (I.D. = 1.05 mm and length = 20 mm) and carefully aligned using an optical microscope, such that it is centered vertically and horizontally. Then, the capillary tubes were attached on a glass slide using epoxy glue. A 20 gauge dispensing needle was placed on a gap formed between the two tubes and sealed with epoxy glue. The opposite end of the outer tube was used for fluid injection (Supplementary Fig. 4a). For the second geometry, a 50 mm long tapered circular capillary tube (I.D. ≈ 400 μm) was carefully inserted into the open end of the outer tube and placed the tube tip with ~1 mm distance from the holding tube tip (Supplementary Fig. 4b). Fluid was injected through this tapered capillary tube. For the third geometry, the outer square capillary tube was narrowed to mimic a stenotic vessel. The center region of the outer tube was locally heated and stretched using micropipette puller until its inner diameter reached ~800 μm. During this process, the geometry of the outer tube became circular. Then, the fire-polished holding tube and the tapered capillary tube were inserted into the outer tube (Supplementary Figs. 4c and 5). To perform the mechano-FL sensing experiment, the device was filled with water. A single PDA particle was added to the injection tube and captured at the holding tube by applying a negative pressure, which was controlled by the suction rate using a syringe pump (Harvard Apparatus, USA). In our experimental setup, the PDA particles could be discharged into the gap between the outer tube and the holding tube. As a result, the mechano-FL sensor device can be readily reused for subsequent experiments.

### Measurement of fluid viscosity

To determine the viscosity of the fluids, we utilized a Brookfield rotary viscometer (LV DV-I PRIME, Brookfield Engineering Laboratories, Inc., USA) equipped with a low viscosity cylindrical spindle (LV-1 standard, Brookfield Engineering Laboratories, Inc., USA). The measurements were performed at least three times to ensure accuracy and reproducibility.

### Characterization of the PDA particles

The morphology of the PDA particles was analyzed using an optical microscope (IX83, Olympus, Japan) and a field-emission scanning electron microscopy (FE-SEM, LEO SUPRA 55, Carl Zeiss, Germany). For visualizing the interior of PDA particles, they were dispersed within polydimethylsiloxane (PDMS), followed by curing the sample for two days under ambient conditions after adding a 10 wt.% curing agent. Subsequently, the PDMS-embedded PDA particle was carefully

bisected with a sharp razor. The bisected PDA particle was submerged in chloroform for an hour, dried, and finally imaged using SEM. The thermal response of PDA particles was examined with a microscope hot-stage (HS82, Mettler-Toledo Inc., USA). The blue-to-red transition of the PDA particles was analyzed using a Raman spectroscopy (inVia Basis, Renishaw, UK) equipped with an adapted research-grade Leica DM 2700 microscope and a high-power near-infrared Ranishaw diode laser (>250 mW at 785 nm). Note that the use of 785 nm laser was beneficial to reduce background effects and maintain relatively high signal intensity. Confocal microscopy measurements were performed using an Olympus IX83/FV3000 inverted confocal laser microscope (Olympus, Japan), equipped with a charge-coupled device (CCD) camera (DP80, Olympus, Japan). An objective with a numerical aperture of $NA = 0.45$ (LUCPlanFL N 20×, Olympus, Japan) and a 561 nm diode laser were used in this study.

## Mechano-FL sensing data acquisition

In order to capture FL response images, a digital single-lens reflex (DSLR) camera (EOS 700, Canon, Japan) mounted on an optical microscope (Ti-U, Nikon, Japan) was employed. The microscope was equipped with a metal halide illuminator (PhotoFluor LM-75, 89 North, USA) offering a full output spectrum from 340 nm to 800 nm. To selectively isolate red FL emission region, a Nikon FL filter (C-FL Epifluorescence TRITC filter, Nikon, Japan) was used. All FL measurements and observations were performed in a dark environment to eliminate potential interference from external light sources, ensuring the accuracy of the FL signals. Additionally, the PDA particles were only exposed to the FL laser during the FL measurements, and the DSLR camera settings were kept consistent (i.e., ISO 100 for camera sensitivity and 0.4 s exposure time) during the capturing of FL images. Note that each PDA particle was utilized only once for the mechano-FL sensor experiments; no reused particles were incorporated into any subsequent FL sensor experiments. All the FL measurements were conducted independently at least three times. The mean intensity values, along with their respective standard deviations, were reported in the main text and Supplementary Information. By maintaining these stringent conditions, we ensured the minimization of any potential errors from FL measurements.

To quantify the FL values, each acquired FL image was processed to separate its color components—red, green, and blue (RGB)—using the 'Splitting Channels' function in ImageJ software[71]. During this process, the color of each pixel, determined by the intensity combination across the RGB channels, was separated into individual grayscale images representing each color channel. In these grayscale images, pixel intensity corresponds to the strength of the respective color in the original image. Subsequently, the mean gray value of the red channel was acquired and normalized within a range of 0 to 1 using the maximum intensity value of 255. In the mechano-FL sensor experiments, images obtained from different PDA particles under identical experimental conditions were processed in the same manner. Note that because the 570–700 nm emission filter was used, any signals from the green and blue channels were barely detectable. The normalized red intensity values derived from the multiple measurements were averaged, and the resulting mean values were used for further data analysis to ensure consistency.

## Preparation of RBC suspension

RBCs contained in defibrinated sheep blood were separated from blood plasma and buffy coat by centrifuging at 13,680 g for 2 min. After removing the supernatant, obtained RBCs were resuspended in 0.01 mol·L$^{-1}$ phosphate-buffered saline (PBS, pH 7.4, Sigma-Aldrich, USA) and centrifuged. This washing procedure was repeated three times. The cell number density was determined using a bright-line hemocytometer (Sigma-Aldrich, USA) under an optical microscopy (IX83, Olympus, Japan) (Supplementary Fig. 11b). The RBC suspension

with a concentration of ~$9.24 \times 10^5$ cells·μL$^{-1}$ was stored at a refrigerator. Viscosity and pH of this RBC solution were ~5.8 cP and ~7.44, respectively. Prior to use of RBCs for the mechano-FL sensor system, the RBC solution was diluted with a 20 mM sodium phosphate buffer (pH = 7.0), resulting in pH = 7.15 and final concentration of $1.54 \times 10^5$ cells·μL$^{-1}$.

## Preparation of yeast cell suspension

The yeast cell suspension was prepared using baker's yeast cells (Saccharomyces cerevisiae) with a diameter of 5–10 μm[72]. 1 wt.% dry yeast was dispersed in a sterile medium containing ultrapure water and cultivated at 30 °C for 3 days. The medium was supplemented with 0.1 % (v/v) bovine serum albumin (BSA, Sigma-Aldrich, USA) to nourish the cell culture and prevent yeast cells from aggregating[73]. The measured pH value and viscosity of the resulting suspension were 4.97 and 1.5 cP, respectively. This solution was diluted with a 20 mM sodium phosphate buffer (pH = 7.00) to prepare two samples with different cell concentrations: yeast-1 ($4.80 \times 10^4$ cells·μL$^{-1}$, pH = 6.99) and yeast-2 ($2.4 \times 10^4$ cells·μL$^{-1}$, pH = 7.03) (Supplementary Fig. 11c). The cell concentration was measured with a bright-line hemocytometer under an optical microscopy (IX83, Olympus, Japan) (Supplementary Fig. 11a).

## Numerical simulation

The CFD simulation (Supplementary Fig. 12) of the mechano-FL sensor system was carried out using the widely used finite volume method (FVM) with Ansys-Fluent (Academic Workbench 2021R2) due to its versatility, robustness, and accuracy. We refrained from using the meshless method due to potential challenges such as convergence and stability issues, difficulty in enforcing boundary conditions, and high computational costs for large numbers of nodes. We evaluated the effects of injection flow rate, viscosity, and SNP concentration on the absorbed energy by the PDA particle. The computational domain is subdivided into control volumes, and differential equations are integrated to produce a set of algebraic equations. These approaches are intrinsically conservative and can describe various arbitrary geometries[74]. The ANSYS meshing tool was used to create meshes for the presented control volumes. The unstructured triangular/tetrahedral hybrid mesh elements were chosen in this study due to their flexibility and capacity to cope with complex geometries. For simplicity, the PDA particle was fixed at the holding tube without introducing the negative pressure on it. The governing equations for numerical analyzes of fluid flows and SNPs motions inside the microchannel were solved using the Ansys-Fluent FVM CFD codes. To reduce the computational cost but maintain reasonable accuracy, the Eulerian approach with the $k$-$\omega$ SST turbulence model was implemented for the steady-state incompressible Newtonian fluid flow, in which pressure and velocity distributions were determined via mass and momentum conservation equations[59,60]. The $k$-$\omega$ SST model combines the strengths of the low Reynolds $k$-$\varepsilon$ turbulence model in free-stream flows with the benefits of the $k$-$\omega$ turbulence model near walls. This approach can address certain limitations of both the pure $k$-$\omega$ and $k$-$\varepsilon$ models, resulting in enhanced accuracy and stability over a wide range of flow conditions[75,76].

In the case of particle injection, the discrete phase model (DPM) was employed, which relies on the Eulerian-Lagrangian approach as one of the CFD methodologies for numerical simulations of fluid-solid systems[77]. The particle motions are tracked based on the Lagrangian approach[78]. Supplementary Table 4 lists the governing equations for the fluid flows, particle motions, and Reynolds number[79]. Supplementary Table 5 summarizes the DPM simulation variables. The two-way coupling was used to calculate the collision energy, in which the dispersed and continuous phases are influenced with each other[80]. The particles were injected into the simulation domain from an inlet at 393.15 K and in the direction normal to the inlet surface. The SIMPLE

algorithm with pressure–velocity coupling and the second-order upwind scheme were implemented to discretize all of the convective terms in the transport equations. The default values for the under-relaxation factors for momentum, pressure, and turbulent kinetic energy in Fluent are 0.70, 0.30, and 0.80, respectively. Initial boundary conditions for the inlet were specified as pressure-based steady-state flow injection with a uniform velocity distribution. At the outlet, the pressure-outlet condition was set with restricted backflow and zero-gauge pressure. All fluid walls were subjected to a no-slip boundary condition, and the roughness of the PDA surface was fixed to 2 μm.

## Data availability

All data are present in the paper and the Supplementary Information. Any additional information can be obtained from the corresponding authors upon request. Source data are provided with this paper.

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

## Acknowledgements

This study was supported by the National Research Foundation (NRF) of Korea, No. 2021R1A5A6002853 (B.J.P), 2020R1A6A03048004 (B.J.P), 2021R1A2C2005906 (J.-M.K), and 2021M3H4A1A02051834 (J.-M.K).

## Author contributions

J.-M.K. and B.J.P. conceived and supervised the project. N.A. performed the mechano-FL experiments and the CFD simulations. N.A. and J.L. performed the microfluidic PDA fabrication. N.A., C.B.G., and B.J.P. wrote the first draft. All authors contributed to the interpretation of experimental data and read, edited, and commented on this manuscript.

## Competing interests

The authors declare no competing interests.
