## [Transparent Peer Review file · Nature Communications]

A single-particle mechanofluorescent sensor

Corresponding Author: Professor Jong-Man Kim

Version 0:

Reviewer comments:

Reviewer #1

(Remarks to the Author)

The quantification of flow-induced mechanical stress inside channels at both macro and micro scale level is extremely challenging. This manuscript (MS) proposes a mechanofluorescent sensor system that can be employed to analyze flow-induced mechanical stresses of a single polydiacetylene (PDA). This is an interesting approach that can be applied in lab and organ on-a-chip platforms to measure flow-induced stresses. The MS is generally clearly presented, organized and has potential to those in the field of biomicrofluidics and biomedical engineering. However, this MS needs several improvements that must be addressed before the acceptance.

The following points need to be addressed:

- The authors should clarify if the proposed system only works for a single PDA. How to ensure that you only have one PDA to analyze and what happens when the PDA is released from the suction region. It blocks the system? Can we use this system several times? What will happen if we have several PDA within the flow to analyze. Is this a limitation of this system? The authors should clarify the limitations of the proposed system and the way to overcome them in the near future.
- In the abstract the authors mention "...for designing minimally invasive mechanical stress measurement systems that can be placed in small and difficult-to-access areas deep within the human body". In order to make this statement, the authors should provide biocompatibility tests of the proposed system and PDA.
- Also in the abstract and along the MS the authors mention "comprised of a fluorogenic single polydiacetylene (PDA) particle, which is fabricated using the microfluidic method". Please be more specific. You should say for instance "...by using a co-flow microfluidic approach".
- A critical point is regarding the fluid dynamic results. The authors should significantly improve this part by providing the Reynolds number (Re), velocity profiles, shear stress, etc, around the measurement region. The injection flow rate does not give any relevant information about the velocities, shear and mechanical stresses in the region of interest where the flow measurements are made. Just by knowing these velocities and shear stress is possible to compare with in vivo features. In page 8, line 153 and 154 please revise this sentence "a flow rate range that is similar to velocity of 3–26 mL/min in 0.8–1.8 mm arteries in the human finger." The unit of the velocity is not mL/min. Please revise and improve.
- The authors should also provide the hematocrits(Hct) for both red blood cells (RBCs) and yeast cells. The Hct depends on the size and location of the channel so the authors should clarify the Hcts that they have used in this study. What is the relevance of using yeast cells and silica nanoparticles in comparison to RBCs. Nowadays there are particulate blood analogues easy to be made that could give information more relevant than yeast cells and silica nanoparticles.
- In page 11 the authors mention that "...the ~3 times greater intensity change promoted by the RBCs is directly related the collisions of solid objects, even when soft, with the PDA particle" and for the yeast cells "...the collision effect clearly governs the response of this mechano-FL sensor system." However, there are no images/videos, cell visualizations and results (both experimental and numerical) showing these collisions. Significant improvements should be made on this point.
- For the CFD simulations the authors decided to select a turbulent model but there is no information/discussion about the Re ? Can this numerical approach show the impact and influence of the cells collisions with the PDA particle. Is the FVM the most appropriate numerical method for this application? Why not using mesh less methods ? Please provide additional information about the selected numerical method and model.
- In general the captions of several Figures need improvement and additional information. The captions of Figure 4 and S1 should be improved. In Figure S3 is missing the scale and quantitative velocities around the region of interest.

- Should be nice to show a schematic Figure showing more details for each step for the fabrication of the mechano-FL sensor device.

Reviewer #2

(Remarks to the Author)

Summary

This paper describes a novel method for developing large polydiacetylene (PDA) particles and a microfluidic approach to measuring mechanical stress induced fluorescence. The paper describes a useful way to make these particles, but significant work is needed to clarify the mechanochromic behavior and some of their observations. It is unlikely that the developments presented will enable "mechanical stress measurement systems that can be placed in small and difficult-to-access areas deep within the human body." A glaring issue in the manuscript is that all results are presented in energy. Why aren't stresses reported here?

Major Issues:

1. The use of a DSLR with an unspecified microscope configuration is problematic. Was autofluorescence an issue? Please provide a thorough and in-depth description of the fluorescent microscopy setup. What set of fluorescent filters were used? What light source was used? Because it appears a color camera was used to assess the fluorescence and an RGB calculation was done to extract red fluorescence, these parameters matter significantly.

Furthermore:

Complementary, independent measurement of the blue and red phase should be done: ideally UV-vis spectroscopy of blue and red particles and true, monochromatic fluorescence using a high quality monochrome scientific camera. The Raman spectroscopy, does not appear to have been done in-situ nor for the bombarded particles.

How were experiment-to-experiment and day-to-day fluorescent intensity variations handled? Often the intensity from image to image is not directly comparable, and a reference color card is used.

* The use of an RGB to fluorescence is a poor indication of PDA behavior, especially without a second independent validation method. Spectroscopy or well tailored fluorescent filtering provide significantly more quantitative descriptions of the behavior. This RGB calculation should be validated with other in-situ techniques. These particles are large enough to easily be resolved by even a low quality UV-vis spectrometer. Data acquisition of the RGB-fluor. Is it not clear did they specify a region of interest and track it? Did they integrate over the image, further details on acquisition and processing are important and should be included.

2 The overall literature review is poor and focuses almost exclusively on recent and crude measurement of PDAs. I suggest referencing the following papers for the following sub-regions of PDAs mechanochromism:

* <https://pubs.acs.org/doi/10.1021/acs.nanolett.0c04027>

* <https://iopscience.iop.org/article/10.1088/0953-8984/16/23/R01>

* <https://onlinelibrary.wiley.com/doi/full/10.1002/admi.202201808>

origin of the blue to red transition

* <https://onlinelibrary.wiley.com/doi/abs/10.1002/3527607323.ch3>

* <https://pubs.acs.org/doi/10.1021/jp0638437>

* <https://www.sciencedirect.com/science/article/pii/S0009261499011379>

* <https://www.nature.com/articles/nphys196>

* <https://pubs.acs.org/doi/10.1021/la991580k>

* <https://pubs.rsc.org/en/content/articlelanding/2010/CP/B915527A>

<https://pubs.acs.org/doi/10.1021/la970831r>

<https://pubs.acs.org/doi/10.1021/la8029038>

<https://pubs.acs.org/doi/10.1021/la991580k>

<https://pubs.rsc.org/en/content/articlelanding/2007/cc/b703691d>

<https://pubs.acs.org/doi/10.1021/acs.macromol.0c00718>

3. It's unclear what the nanoparticles are doing to the PDA

* are the particles impacting the PDA particle and then deflecting or embedding themselves into the particle?

* Are they traveling through the PDA particle?

* Some interesting tests would be to take a big particles (like 100-200 micron) and smash them into the PDA particle. This is almost something that could be examined using the classroom physics billiard balls momentum calculation.

The treatment of the blue to red transition in this work is unclear and at times contradictory.

* Figure 1 panel c suggests that distortion of the polymer backbone is responsible for the blue to red transition

* Page 6 Line 108-109 suggests that the blue to red transition is due to a transition from planar to non-planar geometry.

The authors should refer to the above references to clarify the following:

It's been well established that the red phase may be even more ordered than blue phase PDA, but simply non-planar. In some cases, it may be that an increase in disorder is associated with the blue to red transition. However, this is not exclusively true. Highly ordered red chains are well documented in literature. The authors should revise figure 1 and their statements to avoid making any conclusions on backbone conformation from these macro-scale measurements without significant experiment evidence.

See: the above references on the origin of the blue to red transition, this paper on a potentially disordered PDA

<https://onlinelibrary.wiley.com/doi/abs/10.1002/adfm.201301044>

Furthermore, the authors make a very unusual series of conclusions based on their simulations regarding the torsion of PDAs. The packing and angles of PDAs at the molecular scale is not fully elucidated and is still relatively challenging to investigate even with atomistic simulations and molecular scale experiments. The CFD model and association molecular torsion needs to be re-examined quite closely. To make those conclusions, the authors need to draw a molecular picture of what this might look like. The length scales between where this occurs (angstroms) to the 100s of microns over which this

simulation is done, makes this whole assessment seemingly spurious. It is unclear how normal or shear stresses applied via particle bombardment or viscous stress is imparted onto the molecule in the form of torsion of the polymer backbone. Atomic force microscopy measurements show that "torsion" is not necessary, applying lateral stress along the PDA backbone is sufficient to induce the blue-to-red transition. It seems extremely difficult to correlate nano-scale phenomena with micron-scale continuum mechanics simulations.

Their energy calculations are also very unusual, they should report shear and normal stresses at the surfaces and compare to existing mechanical stress data found in:

<https://pubs.acs.org/doi/10.1021/acs.nanolett.0c04027>

<https://onlinelibrary.wiley.com/doi/full/10.1002/admi.202201808>

<https://pubs.acs.org/doi/10.1021/la9815998>

<https://doi.org/10.1023/A:1019113218650>

The torsional calculations are naive. From the supplemental, it seems that they assume the entire particle is polymerized in the blue phase, prior to application of mechanical stress. It is well known that significant amounts of monomer remain in PDAs as only the topochemically aligned structures are capable of polymerizing. Even in well packed, ordered Langmuir films, polymerization is generally incomplete, especially in the blue phase. Complete polymerization is usually only achieved through prolonged irradiation of the polymer and in the red phase. Hence, assuming the entire particle polymerized is completely polymerized is unrealistic, especially without supporting evidence. The easiest way to validate this is to expose to a calibration uv light source and then measure the absorbance spectra as a function of UV light exposure. The maximum intensity of the particle will very likely be in the red phase.

Are these solid particles of PDA? What is inside? Figure 2d doesn't give a complete picture of the internal structure, or what that internal structure is. The classic liposome cross section often shown in SEM images of vesicles should be utilized here. Additionally, if it is assumed that the PDA is fully polymerized, the domains should be readily viewable with a polarized light microscope. Were there domains large enough to be observed?

The concept of the blue to red transition propagating through the particle needs further examination, especially given the size of the particle. It should be readily viewable how the blue to red transition occurs spatially within the particle and yet the data they show is not really clear. Figure 4 e is the most clear, but experimental details are not clear enough. Experiment to try: what if they injected nanoparticles for a second, then stopped adding nanoparticles: does the fluorescence continue to grow? Then it propagates, does it stop after flow of nanoparticles stops?

The particles shown in Figure 2c look completely different from the particle in figure 3f. What happened to the fuzzy domains? Are they stripped off? Furthermore the references used in line 92 (37, 38) are fine, but don't really explain the fuzzy polymer domains that form, and neither reference refers to the domains they investigated as "plate-like". The authors should investigate the following reference, where giant vesicles covered in similar domains were investigated.

See: <https://onlinelibrary.wiley.com/doi/abs/10.1002/adfm.200700726>

Additional Comments regarding the manuscript that should be clarified.

The microfluidic channel approach shows promise as an interesting way to both visualize and quantify the role of shear and normal stress in driving blue to red transitions. The particle synthesis approach potentially opens the door to further quantification of mechanical stress induced fluorescence in PDAs through a wide variety of traditional single molecule and microfluidic approaches for example optical tweezers, micropipette aspiration, etc.

The role of pH salt concentration in the blue-to-red transition needs further explanation. The authors are clearly aware of the effects that basic pH (8 – 9) have on PDAs, however their literature review of this portion of the PDA field is quite narrow and would provide significant insights into why this is occurring. I would suggest investigating and potentially citing some of the following papers. PDAs can be quite sensitive to changes in ionic strength, and pH. Furthermore their work with silica nanoparticles raises several questions

see: <https://pubs.acs.org/doi/full/10.1021/ac0517794>, <https://pubs.acs.org/doi/full/10.1021/ja0299001>,

<https://www.sciencedirect.com/science/article/pii/S0927775720302429> among many many others...

When PDA particles are buffered down to around pH 7: do they aggregate? The stock solutions are quite basic. Were PDA particles and the buffer, glycerin or other solvents just added together and allowed to equilibrate? Did the PDAs slowly transform?

The particle size distributions are not given for these nanoparticles. These are commercially available. A quick search of the product pages of these nanoparticles does not yield DLS or SEM characterization. The authors should strongly consider adding DLS particle size distributions (with full cumulants analysis and multimodal size distributions properly weighted as volume, intensity and number in the supplemental), both as stock and when buffered.

Viscosity is an important parameter that is discussed in these experiments. However, there is no mention of how viscosity of any samples were measured (using say a rheometer), how it was calculated, or what references were used. Please provide references, and experimental details for all viscosity calculations. If the viscosity was measured, error and uncertainty should be reported (Figure 3 a-d).

The microfluidic setup is not described in sufficient detail. What Reynolds numbers were calculated? What flow regime was this system operated in? Significant program-specific jargon related to the "Fluent" CFD package is referenced, but standard parameters for flow behavior within the channel are not described in the manuscript or supplemental. Furthermore, this model seems awfully complicated for what could maybe be approximated with back of the envelope calculations from a fluids textbook.

The figures need significant work. The color scheme and resolution on many of the figures is so poor that the data cannot be properly interpreted. Figure 3: Markers, error bars and viscosity bars need to be revised. Markers are too big and error bars are too small. The viscosity bars are too lightly shaded to be easily seen and compared between figures. The resolution of the figures is terrible. Many micrographs are missing scale bars making interpretation difficult. Figure 5: too blurry to make out anything from the CFD simulations. The jet color scheme is misleading and a perceptually uniform colormap should be used instead. Velocity and pressure distributions are too small to be interpreted. Figure 2. panel e doesn't tell us anything that the inset doesn't already tell us. It's also too small and blurry to be useful.

With regards to the PDA particle size: what governs it? The microfluidic system? Solvent choice? Solvent choice of chloroform is interesting as it is denser than water.

Please provide a detailed description of the confocal microscope used to acquire the inset in figure 3 either in the experimental section or in the supplemental information. How were confocal images processed?

The discussion section is incredibly short and reads more like a conclusions section. Why does anyone care about this beyond some high level discussion of arterial function? I'm unsure if this will appeal to the general audience of Nat. Comm. The data availability statement is not sufficient as the raw data is all in the manuscript or supplemental. Information on how statistics and error should be provided.

Reviewer #3

(Remarks to the Author)

In this paper, the authors demonstrated a droplet microfluidics-based fabrication technique to develop fluorescent PDA particles with unique mechanosensitive properties. After thoroughly characterizing the particles, the authors explore their potential as a probe to monitor mechanical stress in the microchannel and particulate flows. In the key experiments, the authors individually trap these particles at the center of a microchannel and induce/tune the injection flow parameters, such as flow rate, flow duration, viscosity, and particle concentration. Observable mechanofluorescent response occurs when a colloidal suspension is injected (10-20% particle concentration, somewhat similar to physiological RBC flows). Empirically and based on the SEM images, it is possible to hypothesize that colloids bombard the mechanosensitive particle, transferring energy and distorting the PDA particle's backbone, thereby causing mechanofluorescent sensors. However, this response is nonlinear: typically, fluorescent signal increases after 1 minute, plateaus after 2 min, and rises again after a few minutes of exposure (Fig.3e, Fig. S5d, Fig.5f). This response is critical for its stress-sensing application in a biological flow. Finally, the authors apply an ambitious CFD model to elucidate this response, simulations estimate the transferred hydraulic and collision energies to the PDA particle, and they particularly relate collision energy to the critical torsion energy needed to cause an optimal fluorescent response.

While the reviewer thinks that the overall concept and its application are interesting, the reviewer do not think it meets the standards and requirements of Nature Communications (see major comments).

Major comments:

1. The reviewer is unable to see an obvious application for this sensor probe. For instance, the authors mentioned that "this study represents a significant advancement in the development of minimally invasive force measurement systems that can be located in hard-to-reach spaces within the human body, including implantable vascular prostheses, blood cell trauma caused by stenosis, and flow-induced thrombogenicity." How do the authors plan to translate this system to establish a mechanical stress measurement system within the human body? In the experiment, sensor particles have a certain size (200 μm diameter), and they are manually placed and trapped at the center of the microchannel flow to induce colloidal bombardment, which causes an irreversible fluorescent response measured by a microscope setup. In large vessels, these particles will probably flow along with the RBCs, in small vessels these particles will themselves induce blockage and cause high shear stresses? Furthermore, how do the authors measure the fluorescence coming from deep within human body? What would happen to the PDA particles?
2. If the authors do not plan to establish the system within a human body and only plan to apply the probe in 3D-printed, vessel-mimicking microfluidic channels, does the sphere affect the flow profile and hence affect the measurement? The probe is shaped like a sphere requiring direct contact with the fluidic flow. It is difficult to understand how the sphere (200 μm diameter) could be placed in a fluidic channel without altering the flow profile of the fluidic channels.
3. The key sensor response seems highly specific to the induced flow and does not appear to have a linear and high dynamic range.

Minor comments:

- What are the standard deviations in the plots? What is the number of particles (n) tested in each experiment?
- Would it be possible to tune the mechanofluorescent response? (e.g., more or less sensitive to stress and particle injection)
- Is there any other response after the FL intensity of 0.3 a.u? Is this about the maximum? Does the particle start to lose its mechanical integrity in the flow after that (e.g. if one performs these experiment for >10 minutes)?

Reviewer #4

(Remarks to the Author)

In their manuscript, Ahmadi et al. describe the development of monodisperse polydiacetylene (PDA) microparticles and their fluorescence responses to a flow of fluid that contains silica nanoparticles or red blood cells. The concept and the results obtained are highly novel and interesting, but there are several points that must be corrected before the manuscript can be published.

- About the preparation and characterization of PDA particles:

- 1) As far as I understand the particles fabrication protocol, the injection rates of aqueous PVA solution and PCDA monomer solution in chloroform are important. Could the authors explain what happen if these flow rates are changed? Is it possible to modify the size of the particles by modifying these flow rates or does this have an impact on the size dispersion?
- 2) What is the role of PVA in this protocol and are the obtained particles coated with PVA at the end of the process?

- 3) Could the authors give some indications about the stability of the PDA particles (colloidal stability and photostability during the fluorescence experiments)?
- 4) The photopolymerization procedure should be described more precisely: what is the power per surface unit received by the particle suspension? Could the authors provide some information about the UV-Vis absorption spectrum of the polymerized particles? This is important because it is known that for polydiacetylene the shape of the UV-Vis spectrum can be modified when the photopolymerization progresses with sometimes the apparition of red PDA phase that can change not only the absorption properties but also the fluorescence emission properties (see for example Y. Lifshitz, A. Upcher, O. Shusterman, B. Horovitz, A. Berman and Y. Golan, *Phys. Chem. Chem. Phys.*, 2010, 12, 713–722 for a description of the evolution of absorbance of PDA films with the photopolymerization rate).
- 5) As far as I understand, the authors seem to suppose that the diacetylene monomers in the particles are completely polymerized to give polydiacetylene chains. In particular, this hypothesis is used in the calculation of the PDA torsion energy. However, it has been demonstrated on several PDAs that upon UV irradiation the polymerization is far from complete. This has been shown in particular on PCDA in this reference: Spagnoli et al., *Langmuir* 2017, 33, 1419–1426. As a consequence, the authors should estimate the polymerization rate in their microparticles, and correct the PDA torsion energy calculation accordingly.
- 6) Could the authors specify in ESI which excitation wavelength they use for fluorescence experiments and also for Raman characterization? This would be useful to ensure good reproducibility of the results.

- About the sensitivity of the PDA particles to stimuli:

- 1) I am intrigued by the results displayed on figure S2. Could the authors give an estimate of the force (or the stress) applied on the PDA particle by the three methods (shearing, poking, pressing)? In particular, in the following references (*Chem. Commun.* 2019, 55 (97), 14566–14569 and *Nano Lett.* 2021, 21, 543–549 which could both usefully be cited in the article) PDA are described to be sensitive to anisotropic friction stress, which seems to be in contradiction with the results displayed on figure S2 where visually the PDA particle seems to be more sensitive to pressing than to shearing. Could the authors discuss the apparent contradiction?
- 2) Could the authors please add scale bars on figure S3?
- 3) Could the authors explain how they optimized the conditions for force sensing using the PDA particle? In particular, what happens at silica NPs concentration higher than 16.96%? And what happens at times longer than 4.5 minutes? How is the potential photobleaching of the particle taken into account during the measurements?
- 4) The authors present a model where the mechanofluorochromic response is linked to the energy absorbed by the PDA particle, with the highest fluorescence response obtained at high concentration of particles and longest observation time. The authors should detail whether or not a short injection time with a high concentration of silica particles is equivalent to a long injection time with a smaller concentration of particles. This would strongly comfort the proposed model and increase the practical usefulness of such a force sensor.
- 5) What would be the detection limit of the proposed force sensor? And does the “optimal response” presented by the authors correspond to a saturation limit of the sensor (in other words does the fluorescence signal stay similar when higher mechanical stress is applied to the PDA particle)?
- 6) The authors notice a fluorescence response after simple incubation of the PDA particle with various silica nanoparticle and justify this by a sensitivity to pH. I am very surprised that pentacosadiynoic acid can be sensitive to pH variations between 8.2 and 9.1 since it is a carboxylic acid completely on its carboxylate form in this pH range. Could the authors explain further?
- 7) About figure 3: in the text p10 l191 the authors write: “local indentations are present mainly at the center region of the PDA particle” and “complete PDA particle fluoresces in the red region after injection of the SNPs for 4.5 min, demonstrating the applied mechanical stress propagates throughout the entire particle.” Is this propagation of mechanical stress instantaneous or is it possible to observe transient states where only the central region of the particle is fluorescent? Such a mechanical stress propagation is described in *Langmuir* 2000, 16, 1270-1278 (reference which should be cited in the manuscript) however it is not instantaneous. Could the authors discuss potential differences between their work and the literature results on this specific point?
- 8) Still about the propagation of mechanical stress, have the authors tried to flow a solution with a high concentration of silica nanoparticles for a short time followed by pure water for a longer time to check how this could influence the mechanical stress detection? If there is stress propagation from the central zone of the PDA particle to the entire particle, this could be highly beneficial for the sensitivity of the device, but this could also strongly perturb the force quantification. I would like the authors to further comment on this point.
- 9) Lastly, if I understand correctly figures 1e and 2e the fluorescence enhancement upon thermal stimulus is quantified between 0 and 1 arbitrary units while the fluorescence enhancement due to the mechanical stimulus is quantified between 0 and 0.3 arbitrary units. I suppose the arbitrary units are the same for the two experiments, and I wonder why if the PDA particle is entirely converted to the red phase after mechanical stimulation the fluorescence enhancement is limited to 0.3. Could the authors explain further?
- 10) Could the authors give information about how many times the experiment was repeated for a given condition and give some details on how they calculated the error bars in figures 3 and 5?

- About the model used:

- 1) Could the authors give the numerical values of the Reynolds number in the system studied and justify the use of a turbulence model? Additionally, if the authors could provide numerical values for all the parameters they use in table S6, it would help the reader understand the model used.
- 2) I cannot find details on how E_{tors0} is calculated (values of table S7 and table S8). Could the authors give more details? If these values are directly taken from reference 47 then it is a huge approximation because the polydiacetylene studied are different.

3) Could the authors provide data, either from their experiment or from the literature, about which torsion angle should be reached on PCDA to observe the blue to red colour change? The data from reference 46 are on polyTHD, which is a completely different diacetylene while the side chain interactions are likely to play a significant role in the torsion angle values. Additionally, it is described in reference 46 that a PDA chain with a torsion angle of 4° is blue while a PDA chain with a torsion angle of 14° is red, which would be completely inconsistent with the results of this manuscript where the most significant fluorescence change is observed for torsion angle between 40° and 50° .

4) Could the authors discuss on the reversibility of the torsion at the scale of one PDA chain? If the torsion is fully reversible then a long experiment time with a little number of collisions on the PDA particle would be equivalent to a short experiment time with many collisions. Are there critical torsion angles beyond which the torsion is not reversible? Is there some thermal relaxation? It would be interesting to know the shape of the function $E_{tors}(\theta)$ as a function of θ in this respect.

Lastly, I am not convinced by the last sentence of the abstract "We anticipate that this study will provide an important foundation for designing minimally invasive mechanical stress measurement systems that can be placed in small and difficult-to-access areas deep within the human body." While I fully acknowledge the originality of the presented results, I do not see how the system described in this manuscript can be implemented within the human body (furthermore I am not aware of studies on the potential toxicity of PDA particles). Could the authors comment and potentially modify the abstract accordingly?

Author Rebuttal letter:

Response to the Reviewer's Comments
REVIEWER COMMENTS

Reviewer #1 (Remarks to the Author):

The quantification of flow-induced mechanical stress inside channels at both macro and micro scale level is extremely challenging. This manuscript (MS) proposes a mechanofluorescent sensor system that can be employed to analyze flow-induced mechanical stresses of a single polydiacetylene (PDA). This is an interesting approach that can be applied in lab and organ on-a-chip platforms to measure flow-induced stresses. The MS is generally clearly presented, organized and has potential to those in the field of biomicrofluidics and biomedical engineering. However, this MS needs several improvements that must be addressed before the acceptance.

We are very grateful for the time and attention devoted by the reviewer for the constructive feedback to this article. The comments have been carefully gone through and the manuscript was carefully revised to address all the points raised by the reviewer.

The following points need to be addressed:

1- The authors should clarify if the proposed system only works for a single PDA. How to ensure that you only have one PDA to analyze and what happens when the PDA is released from the suction region. It blocks the system? Can we use this system several times? What will happen if we have several PDA within the flow to analyze. Is this a limitation of this system? The authors should clarify the limitations of the proposed system and the way to overcome them in the near future.

Our sensor system is designed to analyze one PDA particle at a time. Once a single PDA particle is trapped in the holding tube of the mechano-FL sensor system, an experiment at certain fluid conditions is conducted. After the experiment is completed, the PDA particle is discharged into the gap between the holding tube and the outer tube of the sensor. The same device can then be reused to perform additional experiments using a new PDA particle, following the same methodology. We have clarified this point in the revised manuscript.

Added statements (p6 in SI): In our experimental setup, the PDA particles could be discharged into the gap between the outer tube and the holding tube. As a result, the mechano-FL sensor device can be readily reused for subsequent experiments.

2- In the abstract the authors mention "for designing minimally invasive mechanical stress measurement systems that can be placed in small and difficult-to-access areas deep within the human body". In order to make this statement, the authors should provide biocompatibility tests of the proposed system and PDA.

[Type here]

We appreciate the reviewer's insightful comments and questions regarding the potential applications and translatability of our mechano-FL sensor system for measuring mechanical stress within the human body. In fact, our system was initially designed to serve as an in vitro platform that simulates specific problem areas within the human body, and it also facilitates the visualization and quantification of mechanical forces applied to these regions. At present, the direct implantation of the mechano-FL sensor system into the human body may pose challenges. However, we believe that these challenges could potentially be overcome by coating the PDA particles with biocompatible materials, making in vivo applications not entirely out of reach. To avoid any misunderstandings and to clarify this point, we have revised the relevant section as suggested by the reviewer.

Before (Abstract): We anticipate that this study will provide an important foundation for designing minimally invasive mechanical stress measurement systems that can be placed in small and difficult-to-access areas deep within the human body.

After (Abstract): We believe this study offers a unique advantage for simulating specific problematic regions of the human body in an in vitro environment, potentially paving the way for future exploration of difficult-to-access areas within the body.

Before (Discussion): This study represents a significant advancement in the development of minimally invasive force/stress measurement systems that can be located in hard-to-reach small spaces within the human body, including implantable vascular prostheses, blood cell trauma caused by stenosis, and flow-induced thrombogenicity.

After (p20 in Discussion): We believe this study represents a significant advancement in the development of minimally invasive force/stress measurement in vitro systems that simulate hard-to-reach spaces within the human body.

3- Also in the abstract and along the MS the authors mention "comprised of a fluorogenic single polydiacetylene (PDA) particle, which is fabricated using the microfluidic method". Please be more specific. You should say for instance "by using a co-flow microfluidic approach".

We have made the necessary revisions to clarify the fabrication method of the PDA particle. In the revised manuscript, we explicitly state that the PDA particle was fabricated using the co-flow microfluidic approach.

Before (Abstract): Monitoring mechanical stresses in a microchannel is extremely challenging. Herein, we report the development of a mechanofluorescence sensor system comprised of a fluorogenic single polydiacetylene (PDA) particle, which is fabricated using the microfluidic method.

2
[Type here]

After (Abstract): Monitoring mechanical stresses in microchannels is extremely challenging. Herein, we report the development of a mechanofluorescence sensor system featuring a fluorogenic single polydiacetylene (PDA) particle, fabricated using a co-flow microfluidic method.

Before: The effort to develop this system was comprised of three parts, the first of which involved fabrication of highly conjugated PDA particles with a uniform size distribution by utilizing the microfluidic method (Fig. 1a)

After (p3): The effort to develop this system was comprised of three parts, the first of which involved fabrication of highly conjugated PDA particles with a uniform size distribution by utilizing a co-flow microfluidic method (Fig. 1a)

Before: Fabrication of PDA particles using the microfluidic method.

After (Caption of Fig. 1): Fabrication of PDA particles using the co-flow microfluidic method.

Before: The monodispersed stimulus-responsive fluorogenic PDA particle employed in this effort was fabricated using the microfluidic method. In the process, chloroform droplets containing 10 wt.% 10,12-pentacosadiynoic acid (PCDA) monomers were periodically generated along the flow of an aqueous outer phase using a co-flow microfluidic device (Fig. 1a and Fig. S1).

After (p5): The monodispersed stimulus-responsive fluorogenic PDA particle employed in this effort was fabricated using the co-flow microfluidic method. In the process, chloroform droplets containing 10 wt.% 10,12-pentacosadiynoic acid (PCDA) monomers were periodically generated along the flow of an aqueous outer phase (Fig. 1a and Supplementary Fig. 1).

Before: Fabrication of PDA particles using the microfluidic method.

After (p16): Fabrication of PDA particles using a co-flow microfluidic method.

4- A critical point is regarding the fluid dynamic results. The authors should significantly improve this part by providing the Reynolds number (Re), velocity profiles, shear stress, etc, around the measurement region. The injection flow rate does not give any relevant information about the velocities, shear and mechanical stresses in the region of interest where the flow measurements are made. Just by knowing these velocities and shear stress is possible to compare with in vivo features.

3

[Type here]

We have incorporated the CFD calculations for the region of interest (ROI) into the revised Fig. 5a-c and have added corresponding explanatory text. Given that the fluid velocities within the ROI are similar to those of the injected fluid, it is reasonable to assume that their Reynolds numbers would also be comparable. Detailed information for the injected fluids can be found in Supplementary Table 2.

Added paragraph (p17): The results of these calculations clarify the dynamic behavior of the mechano-FL sensor system, as illustrated in Fig. 5a-c. Specifically, Fig. 5a provides a detailed visual representation of the flow velocity patterns observed when injecting a 16.95 wt.% SNP solution with a flow velocity of $3.317 \text{ m}\cdot\text{s}^{-1}$ (Supplementary Table 2). The maximum flow velocities are found to be approximately $3 \text{ m}\cdot\text{s}^{-1}$ in the microchannel following impact with the PDA particle surface. Fig. 5b visualizes the corresponding shear and normal stress distributions. Notably, the normal stress peaks at a polar angle of $\delta P = 0^\circ$, which is the point where the SNPs collide perpendicularly with the PDA particle, as depicted in Fig. 5c. In contrast, the shear stress reaches a minimum value at the same polar angle ($\delta P = 0^\circ$) and increases as the angle δP widens. Note that the stress scale obtained in these calculations is comparable to the scales obtained through the surface forces apparatus for PDA Langmuir films upon the blue-to-red transition (i.e., $\sim 100 \text{ kPa}$ for normal stress and $\sim 3 \text{ kPa}$ for shear stress).⁴⁷

Revised Fig. 5:

[Image redacted]

Fig. 5 CFD analysis of the mechano-FL sensor system. a,b Examples showing distributions of flow velocity (a) and stress (b) for the 16.95 wt.% SNP suspension. c Variation of shear and normal stresses on the PDA surface as a function of the polar angle δP . Error bars represent the standard deviation over the stress values across the azimuthal angle $\delta\phi$. d,e Plots of δ_{H} and δ_{tot} as functions of glycerin (d) and SNPs (e) concentrations, respectively, at an injection rate of $25 \text{ mL}\cdot\text{min}^{-1}$ over 4.5 min. Corresponding experimentally measured FL intensity profiles are

overlaid for comparison. f Time-dependent profiles of δ_{tor} and FL intensity for the 16.95 wt.% SNP suspension injected at $25 \text{ mL}\cdot\text{min}^{-1}$ over an extended duration of 8.5 min. Inset indicates

4
[Type here]

the comparison of the nonlinear property of normalized FL intensity profiles from the mechano-FL sensor experiment and the thermal stress analysis (from Fig. 2g) against the normalized torsion energy δ_{tor} from a previous study.⁵⁵ The x-axis is normalized from 0 to 1, aligning maximum values of FL intensity and δ_{tor} to 1. For panels d-f, each experimental data point and corresponding error bar represent the mean intensity value of a minimum of three independent trials and its standard deviation, respectively.

Newly added Supplementary Table 2:

Supplementary Table 2. Flow information of the injected fluids into the mechano-FL sensor system.

[Table redacted]

4.1- In page 8, line 153 and 154 please revise this sentence "a flow rate range that is similar to velocity of $3\text{--}26 \text{ mL}\cdot\text{min}^{-1}$ in $0.8\text{--}1.8 \text{ mm}$ arteries in the human finger." The unit of the velocity is not $\text{mL}\cdot\text{min}^{-1}$. Please revise and improve.

We thank the reviewer for pointing this error out. We have corrected it in the revised manuscript.

Before: a flow rate range that is similar to velocity of $3\text{--}26 \text{ mL}/\text{min}$ in $0.8\text{--}1.8 \text{ mm}$ arteries in the human finger.

After (p10): a flow rate range that is similar to the blood flow rate of $3\text{--}26 \text{ mL}\cdot\text{min}^{-1}$ in $0.8\text{--}1.8 \text{ mm}$ arteries in the human finger.

5- The authors should also provide the hematocrits (Hct) for both red blood cells (RBCs) and yeast cells. The Hct depends on the size and location of the channel so the authors should clarify the Hcts that they have used in this study. What is the relevance of using yeast cells and silica

5
[Type here]

nanoparticles in comparison to RBCs. Nowadays there are particulate blood analogues easy to be made that could give information more relevant than yeast cells and silica nanoparticles.

In our experiments, we set the hematocrit level of the injected blood sample at $36.4 \pm 1.5\%$ before dilution and $6.1 \pm 0.3\%$ after dilution. For the Yeast-1 and Yeast-2 samples, the volume percentages, equivalent to the hematocrit level, were $12.8 \pm 0.4\%$ and $6.4 \pm 0.7\%$, respectively. The typical hematocrit levels in humans range from 36 to 50%. We have added this information in the revised Supplementary Table 3.

Regarding the use of the silica nanoparticles (SNPs) and yeast cells, we systematically examined various experimental results under well-controlled variable conditions with our developed PDA sensor system. We initially used SNPs as a screening tool for identifying key variables that influence the sensor system. SNPs are advantageous for these screening experiments because they are less expensive than RBCs and don't risk degradation, making storage and handling relatively convenient. However, in line with the reviewer's comments and given that the PDA sensor aims to mimic the vascular system, we conducted experiments using RBCs under conditions optimized with SNPs. Additionally, to expand the applicability of our PDA sensor system in the diverse biology fields, we also tested the system with yeast cells, another type of readily available soft biological material. These experiments helped validate the results we had observed with RBCs. We have revised the manuscript to clarify these points.

Revised Supplementary Table 3::

Supplementary Table 3. Properties of the injected materials into the mechano-FL sensor system. Note that the RBC number density and the normal hematocrit level observed human are within the range of $\sim 5 \times 10^6 \text{ L}^{-1}$ and 36% , respectively.

[Table redacted]

Before: We elucidated the effect of flow-induced mechanical stress on the FL response of this sensor (Fig. 1c), along with its dependence on several experimental variables including flow velocity, solution viscosity, and concentration of solid objects (i.e., silica nanoparticles (SNPs), red blood cells (RBCs) and yeast cells) were evaluated.

After (p3): We elucidated the effect of flow-induced mechanical stress on the FL response of this sensor (Fig. 1c) and assessed its dependency on multiple experimental variables such as flow rate, solution viscosity, and the concentration of solid objects (i.e., silica nanoparticles (SNPs), red blood cells (RBCs) and yeast cells). We identified the key variable conditions using SNPs and subsequently applied these to the experiments with RBCs and yeast cells.

6

[Type here]

6- In page 11 the authors mention that "the ~ 3 times greater intensity change promoted by the RBCs is directly related the collisions of solid objects, even when soft, with the PDA particle" and for the yeast cells "the collision effect clearly governs the response of this mechano-FL sensor system." However, there are no images/videos, cell visualizations and results (both experimental and numerical) showing these collisions. Significant improvements should be made on this point.

We thank the reviewer's suggestion regarding this matter. In response, we have incorporated FL images from the mechano-FL sensor system experiments using RBCs and yeast cells. These additions can be found in the updated Fig. 4, and the associated text has been revised accordingly.

Before: Specifically, given that the viscosity of the RBC solution (~ 2.5 cP, Fig. S5c) is comparable to that of the 30 wt.% glycerin solution, the ~ 3 times greater intensity change promoted by the RBCs is directly related the collisions of solid objects, even when soft, with the PDA particle.

After (p14): Specifically, given that the viscosity of the RBC solution (~ 2.5 cP, Supplementary Fig. 11c) is comparable to that of the 30 wt.% glycerin solution, the ~ 3 times greater intensity change promoted by the RBCs (Fig. 3h and Fig. 4d) is directly related the collisions of solid objects, even when soft, with the PDA particle.

Before: Injection of solutions of yeast cell with concentrations of $\sim 4.8 \times 10^4$ and $\sim 2.4 \times 10^4 \text{ L}^{-1}$ (Table S2) into the mechano-FL sensor system caused FL intensity responses of $\sim 0.2265 \pm 0.01147$ and $\sim 0.171 \pm 0.012$ a.u., respectively (Fig. S5d), which are also greater than those promoted by glycerin solutions. Here again, because the viscosities of the two yeast solutions are ~ 1.3 and ~ 1.2 cP (Fig. S5c), the collision effect clearly governs the response of this mechano-FL sensor system.

After (p14): Injection of solutions of yeast cell with concentrations of $\sim 4.8 \times 10^4$ and $\sim 2.4 \times 10^4 \text{ L}^{-1}$ (Supplementary Table 3) into the mechano-FL sensor system caused FL intensity responses of $\sim 0.2265 \pm 0.01147$ and $\sim 0.171 \pm 0.012$ a.u., respectively (Fig. 3h and Fig. 4e,f), which are also greater than those promoted by glycerin solutions. Here again, because the viscosities of the two yeast solutions are ~ 1.3 and ~ 1.2 cP (Supplementary Fig. 11c), the collision effect clearly governs the response of this mechano-FL sensor system.

Updated Fig. 4:

[Type here]

[Image redacted]

Fig. 4 FL images of the PDA particle in the mechano-FL sensor system. a-d Representative FL images illustrating the variation in PDA particle FL emission upon injecting different types of solutions: pure water (a), 20 wt.% glycerin (b), 16.95 wt.% SNPs (c), 1.54×10^5 RBCs (d), 4.80×10^4 yeast (e), and 2.40×10^4 yeast (f), corresponding to the results depicted in Fig. 3h. The images were captured at an injection rate of $25 \text{ mL} \cdot \text{min}^{-1}$ and a suction rate of $5 \text{ mL} \cdot \text{min}^{-1}$, with the flow injection time varied.

7- For the CFD simulations the authors decided to select a turbulent model but there is no information/discussion about the Re ? Can this numerical approach show the impact and influence of the cells collisions with the PDA particle. Is the FVM the most appropriate numerical method for this application? Why not using mesh less methods ? Please provide additional information about the selected numerical method and model.

For the Reynolds numbers, we tabulated the values by converting the flow rate injected into 8

[Type here]

the mechano-FL sensor to the flow velocity (Supplementary Table 2). Note that the flow velocity around the PDA particle, as depicted in Fig. 5a, is on a similar scale to the injected velocity.

Regarding our choice of the turbulence model, we have updated the manuscript to provide a more comprehensive justification for our selection of the $k\text{-}\epsilon$ SST model. Briefly, the $k\text{-}\epsilon$ SST model combines the strengths of the low Reynolds $k\text{-}\mu$ turbulence model in free-stream flows with the benefits of the $k\text{-}\epsilon$ turbulence model near walls. This approach can address certain limitations of both the pure $k\text{-}\epsilon$ and $k\text{-}\mu$ models, resulting in enhanced accuracy and stability over a wide range of flow conditions (<https://doi.org/10.2514/6.1993-2906>; <https://www.comsol.com/blogs/which-turbulence-model-should-choose-cfd-application/>).

Before (SI):

8. Numerical simulation

The CFD simulation (Fig. S6) of the mechano-FL sensor system was carried out using the finite volume method (FVM) with Ansys-Fluent (Academic Workbench 2021R2). The effects of injection flow rate, viscosity, and SNP concentration on the absorbed energy by the PDA particle were evaluated. The computational domain is subdivided into control volumes, and differential equations are integrated to produce a set of algebraic equations. These approaches are intrinsically conservative and may describe various arbitrary geometries.⁶ The ANSYS meshing tool was used to create meshes for the presented control volumes. The unstructured triangular/tetrahedral hybrid mesh elements were chosen in this study due to their flexibility and capacity to cope with complex geometries. For simplicity, the PDA particle was fixed at the holding tube without introducing the negative pressure on it. The governing equations for numerical analyses of fluid flows and SNPs motions inside the microchannel were solved using the Ansys-Fluent FVM CFD codes. To reduce the computational cost but keep the accuracy on a reasonable level, the Eulerian approach with the $k\text{-}\epsilon$ SST turbulence model was implemented for the steady-state incompressible Newtonian fluid flow, in which pressure and velocity distributions were determined via mass and momentum conservation equations.^{7, 8}

After (Supplementary Note 1 in SI):

Supplementary Note 1: Numerical simulation

The CFD simulation (Supplementary Fig. 12) of the mechano-FL sensor system was carried out using the widely used finite volume method (FVM) with Ansys-Fluent (Academic Workbench 2021R2) due to its versatility, robustness, and accuracy. We refrained from using the meshless method due to potential challenges such as convergence and stability issues, difficulty in enforcing boundary conditions, and high computational costs for large numbers of nodes. We evaluated the effects of injection flow rate, viscosity, and SNP concentration on the absorbed energy by the PDA particle. The computational domain is subdivided into control volumes, and differential equations are integrated to produce a set of algebraic equations. These

approaches are intrinsically conservative and can describe various arbitrary geometries.⁶ The ANSYS meshing tool was used to create meshes for the presented control volumes. The unstructured triangular/tetrahedral hybrid mesh elements were chosen in this study due to their flexibility and capacity to cope with complex geometries. For simplicity, the PDA particle was fixed at the holding tube without introducing the negative pressure on it. The governing

9
[Type here]

equations for numerical analyses of fluid flows and SNPs motions inside the microchannel were solved using the Ansys-Fluent FVM CFD codes. To reduce the computational cost but maintain reasonable accuracy, the Eulerian approach with the $k\text{-}\epsilon$ SST turbulence model was implemented for the steady-state incompressible Newtonian fluid flow, in which pressure and velocity distributions were determined via mass and momentum conservation equations.^{7, 8} The $k\text{-}\epsilon$ SST model combines the strengths of the low Reynolds $k\text{-}\mu$ turbulence model in free-stream flows with the benefits of the $k\text{-}\epsilon$ turbulence model near walls. This approach can address certain limitations of both the pure $k\text{-}\epsilon$ and $k\text{-}\mu$ models, resulting in enhanced accuracy and stability over a wide range of flow conditions.^{9, 10}

8- In general the captions of several Figures need improvement and additional information. The captions of Figure 4 and S1 should be improved. In Figure S3 is missing the scale and quantitative velocities around the region of interest.

We have accordingly revised the captions and figures to improve their clarity and provide necessary details. Regarding the quantitative velocities around the region of interest in Supplementary Fig. 4 (originally Fig. S3), we refer the reviewer to the revised Fig. 5a and 5c.

Before (Caption of Fig. 4): Fig. 4 FL images of the PDA particle. a-d Representative FL images of the PDA particle at different experimental conditions with varying the flow injection time corresponding to the results in Fig.3e.

After (Caption of Fig. 4): Fig. 4 FL images of the PDA particle in the mechano-FL sensor system. a-d Representative FL images illustrating the variation in PDA particle FL emission upon injecting different types of solutions: pure water (a), 20 wt.% glycerin (b), 16.95 wt.% SNPs (c), 1.54×10^5 RBCs (d), 4.80×10^4 yeast (e), and 2.40×10^4 yeast (f), corresponding to the results depicted in Fig. 3h. The images were captured at an injection rate of $25 \text{ mL}\cdot\text{min}^{-1}$ and a suction rate of $5 \text{ mL}\cdot\text{min}^{-1}$, with the flow injection time varied.

Before (Fig. S1):

10
[Type here]

Fig. S1 Schematic of the microfluidic system for fabricating the PCDA microdroplets (not drawn to scale). The bottom microscopic image indicates the generation of PCDA-chloroform droplets in the capillary channel.

After (Supplementary Fig. 1):

[Image redacted]

Supplementary Fig. 1 Schematic of the co-flow microfluidic system. The diagram represents the microfluidic system utilized for the fabrication of PCDA microdroplets (not drawn to scale). The optical microscope image at the bottom indicates the generation of PCDA-chloroform droplets in the capillary channel (refer to the Supplementary Method section for

detailed methodology).

11

[Type here]

Before (Fig. S3):

[Image redacted]

Fig. S3 Optimization of the mechano-FL sensor system. a,b Primitive designs without (a) and with (b) the injection tube. c Optimized design with the injection tube and the narrowed outer tube. Left and right columns are optical and FL images after subjecting the following flows. The aqueous flow was introduced with an injection rate of 25 mL/min for 4.5 min. The suction rate was 5 mL/min to hold a PDA particle at the holding tube. Pure water was used in a,b and water containing 16.95 wt% SNPs was used in c.

After (Supplementary Fig. 4):

[Image redacted]

12

[Type here]

Supplementary Fig. 4 Optimization of the mechano-FL sensor system. a,b Primitive designs without (a) and with (b) the injection tube. c Optimized design with the injection tube and the narrowed outer tube. Left and right columns are optical and FL images after subjecting the following flows. The aqueous flow was introduced with an injection rate of 25 mL·min⁻¹ for 4.5 min. The suction rate was 5 mL·min⁻¹ to hold a PDA particle at the holding tube. The buffer solution was used in a,b and water containing 16.95 wt% SNPs was used in c.

9- Should be nice to show a schematic Figure showing more details for each step for the fabrication of the mechano-FL sensor device.

We have included a schematic figure and detailed fabrication procedure of the mechano-FL sensor device in Supplementary Information.

Before: For the third geometry, the outer tube was narrowed to mimic the stenotic vessel. The

13

[Type here]

center region of the outer tube was locally heated and stretched until its inner diameter reached ~800 Åμm. Then, the fire-polished hold tube and the tapered capillary tube were inserted into the outer tube (Fig. S3c and Fig. S7).

After (p4 in SI): For the third geometry, the outer square capillary tube was narrowed to mimic a stenotic vessel. The center region of the outer tube was locally heated and stretched using micropipette puller until its inner diameter reached ~800 Åμm. During this process, the geometry of the outer tube became circular. Then, the fire-polished holding tube and the tapered capillary tube were inserted into the outer tube (Supplementary Fig. 4c and 5).

Newly added Supplementary Fig. 5:

[Image redacted]

Supplementary Fig. 5 Schematic illustration of the mechano-FL sensor device fabrication process. The fire-polished holding tube and the tapered capillary tube were inserted into the outer tube, whose center region was narrowed to simulate a stenotic vessel.

14

[Type here]

Reviewer #2 (Remarks to the Author):

Summary

This paper describes a novel method for developing large polydiacetylene (PDA) particles and a microfluidic approach to measuring mechanical stress induced fluorescence. The paper describes a useful way to make these particles, but significant work is needed to clarify the mechanochromic behavior and some of their observations. It is unlikely that the developments presented will enable mechanical stress measurement systems that can be placed in small and difficult-to-access areas deep within the human body. A glaring issue in the manuscript is that all results are presented in energy. Why aren't stresses reported here?

We appreciate the reviewer's acknowledgment of the novelty in our developed PDA mechano-FL sensor system, particularly their commendation of the usefulness of our microfluidic approach for PDA particle fabrication. In response to the comment on mechanochromic behavior and stress measurements, we have included quantitative data on the mechanical stress experienced by PDA particles in the revised Fig. 5b and 5c. The presentation of results in terms of energy was done to facilitate comparison with previously reported energy values associated with the blue-to-red transition of PDA.

As for the reviewer's concerns about the system's applicability in small and hard-to-reach areas within the human body, we wish to clarify that our system is better suited for simulating specific problematic regions in an in vitro environment rather than for direct in vivo use. We have made this point explicit in the revised manuscript.

Before (Abstract): We anticipate that this study will provide an important foundation for designing minimally invasive mechanical stress measurement systems that can be placed in small and difficult-to-access areas deep within the human body.

After (Abstract): We believe this study offers a unique advantage for simulating specific problematic regions of the human body in an in vitro environment, potentially paving the way for future exploration of difficult-to-access areas within the body.

Before (Discussion): This study represents a significant advancement in the development of minimally invasive force/stress measurement systems that can be located in hard-to-reach small spaces within the human body, including implantable vascular prostheses, blood cell trauma caused by stenosis, and flow-induced thrombogenicity.

After (p20 in Discussion): We believe this study represents a significant advancement in the development of minimally invasive force/stress measurement in vitro systems that simulate hard-to-reach spaces within the human body.

Below are our responses to the reviewer's comments.

15

[Type here]

Major Issues:

1. The use of a DSLR with an unspecified microscope configuration is problematic. Was autofluorescence an issue? Please provide a thorough and in-depth description of the fluorescent microscopy setup. What set of fluorescent filters were used? What light source was used? Because it appears a color camera was used to assess the fluorescence and an RGB calculation was done to extract red fluorescence, these parameters matter significantly.

Furthermore: Complementary, independent measurement of the blue and red phase should be done: ideally UV-vis spectroscopy of blue and red particles and true, monochromatic fluorescence using a high quality monochrome scientific camera. The Raman spectroscopy, does not appear to have been done in-situ nor for the bombarded particles.

How were experiment-to-experiment and day-to-day fluorescent intensity variations handled? Often the intensity from image to image is not directly comparable, and a reference color card is used.

* The use of an RGB to fluorescence is a poor indication of PDA behavior, especially without a second independent validation method. Spectroscopy or well tailored fluorescent filtering provide significantly more quantitative descriptions of the behavior. This RGB calculation should be validated with other in-situ techniques. These particles are large enough to easily be resolved by even a low quality UV-vis spectrometer. Data acquisition of the RGB-fluor. Is it not clear did they specify a region of interest and track it? Did they integrate over the image, further details on acquisition and processing are important and should be included.

For capturing fluorescence (FL) response images, we employed a digital single-lens reflex (DSLR) camera (EOS 700, Canon, Japan) mounted on an optical microscope (Ti-U, Nikon, Japan). The microscope was fitted with a metal halide illuminator (PhotoFluor LM-75, 89 North, USA) and Nikon excitation FL filters to selectively isolate FL emissions within the range of 570-700 nm. Throughout all measurements, we maintained consistent camera sensitivity and exposure time settings, specifically at ISO 100 and 0.4 s, respectively. Furthermore, FL measurements of PDA particles were consistently conducted in a darkroom environment. By maintaining such constant conditions, we ensured the minimization of any potential errors (e.g., experiment-to-experiment and day-to-day fluorescent intensity variations). To quantify the fluorescence values, each acquired microscopic image was processed to separate its color components (red, green, and blue (RGB)) using the "Splitting Channels" function in ImageJ software. During this process, the color of each pixel, determined by the intensity combination across the RGB channels, was separated into individual grayscale images representing each color channel. In these grayscale images, pixel intensity corresponds to the strength of the respective color in the original image. Subsequently, the mean gray value of the red channel was acquired and normalized within a range of 0 to 1. In the mechano-FL sensor experiments, images obtained from different PDA particles under identical experimental conditions were processed in the same manner. Note that because the 570-700 nm emission filter was used, any signals from the green and blue channels were barely detectable. The normalized red intensity values derived from the multiple measurements were averaged, and

16

[Type here]

the resulting mean values were used for further data analysis to ensure consistency.

Before:

3. Characterization of the PDA particles

The PDA particle morphology was examined using an optical microscope (IX83, Olympus, Japan) and a field-emission scanning electron microscopy (FE-SEM, LEO SUPRA 55, Carl Zeiss, Germany). The chemical analyses were performed with a Raman spectroscopy (inVia Basis, Renishaw, UK). Thermal response of PDA particles was conducted with a hot-stage (HS82, Mettler-Toledo Inc., USA). The FL response images were captured using a digital single-lens reflex camera (EOS 700, Canon, Japan) mounted on an optical microscope (Ti-U, Nikon, Japan) equipped with a metal halide illuminator (PhotoFluor LM-75, 89 North, USA). For all FL measurements, the camera sensitivity and exposure time were set to ISO 100 and 0.4 s, respectively.

After (p3 in SI):

4. Characterization of the PDA particles

The morphology of the PDA particles was analyzed using an optical microscope (IX83, Olympus, Japan) and a field-emission scanning electron microscopy (FE-SEM, LEO SUPRA 55, Carl Zeiss, Germany). For visualizing the interior of PDA particles, they were dispersed within polydimethylsiloxane (PDMS), followed by curing the sample for two days under ambient conditions after adding a 10 wt.% curing agent. Subsequently, the PDMS-embedded PDA particle was carefully bisected with a sharp razor. The bisected PDA particle was submerged in chloroform for an hour, dried, and finally imaged using SEM. Thermal response of PDA particles was examined with a microscope hot-stage (HS82, Mettler-Toledo Inc., USA). Blue-to-red transition of the PDA particles was analyzed using a Raman spectroscopy (inVia Basis, Renishaw, UK) equipped with an adapted research-grade Lecia DM 2700 microscope and a high-power near-infrared Renishaw diode laser (>250 mW at 785 nm). Note that the use of 785 nm laser was beneficial to reduce background effects and maintain relatively high signal intensity. Confocal microscopy measurements were performed using an Olympus IX83/FV3000 inverted confocal laser microscope (Olympus, Japan), equipped with a charge-coupled device (CCD) camera (DP80, Olympus, Japan). An objective with a numerical aperture of NA = 0.45 (LUCPlanFL N 20Å, Olympus, Japan) and a 561 nm diode laser were used in this study.

Before:

7. Mechano-FL sensing data acquisition

Each FL image was analyzed to obtain the red color intensity value x using the following equation,

$$\delta\% = 0.4124 \cdot 0.3576 \cdot 0.1805 \cdot \delta$$

$$[\delta] = [0.2126 \ 0.7152 \ 0.0722] [\delta^\circ]$$

$$\delta\% = 0.0193 \cdot 0.1192 \cdot 0.9505 \cdot \delta\mu$$

The difference in the red color intensities $\delta\%$ before and after applying stresses to the PDA

17

[Type here]

particle corresponded to the FL intensity.

After (p4 in SI):

5. Mechano-FL sensing data acquisition

In order to capture fluorescence (FL) response images, a digital single-lens reflex (DSLR) camera (EOS 700, Canon, Japan) mounted on an optical microscope (Ti-U, Nikon, Japan) was employed. The microscope was equipped with a metal halide illuminator (PhotoFluor LM-75, 89 North, USA) offering a full output spectrum from 340 nm to 800 nm. To selectively isolate red FL emission region, a Nikon FL filter (C-FL Epi-fluorescence TRITC filter, Nikon, Japan) was used. All FL measurements and observations were performed in a dark environment to eliminate potential interference from external light sources, ensuring the accuracy of the FL signals. Additionally, the PDA particles were only exposed to the FL laser during the FL measurements, and the DSLR camera settings were kept consistent (i.e., ISO 100 for camera sensitivity and 0.4 s exposure time) during the capturing of FL images. Note that each PDA particle was utilized only once for the mechano-FL sensor experiments; no reused particles were incorporated into any subsequent FL sensor experiments. All the FL measurements were conducted independently at least three times. The mean intensity values, along with their respective standard deviations, were reported in the main text and Supplementary Information. By maintaining these stringent conditions, we ensured the minimization of any potential errors from FL measurements.

To quantify the FL values, each acquired FL image was processed to separate its color components (red, green, and blue (RGB)) using the "Splitting Channels" function in ImageJ software.³ During this process, the color of each pixel, determined by the intensity combination across the RGB channels, was separated into individual grayscale images representing each color channel. In these grayscale images, pixel intensity corresponds to the strength of the respective color in the original image. Subsequently, the mean gray value of the red channel was acquired and normalized within a range of 0 to 1 using the maximum intensity value of 255. In the mechano-FL sensor experiments, images obtained from different PDA particles under identical experimental conditions were processed in the same manner. Note that because the 570–700 nm emission filter was used, any signals from the green and blue channels were barely detectable. The normalized red intensity values derived from the multiple measurements were averaged, and the resulting mean values were used for further data analysis to ensure

consistency.

Regarding the use of Raman spectroscopy, we recognize and greatly appreciate the suggestion that in-situ Raman measurements could notably benefit our mechano-FL sensor experiments. However, the spatial limitations present challenges when attempting to mount the mechano-FL sensor system on the stage of our Raman spectroscope. Looking forward, it would be advantageous to develop a setup that allows for in-situ Raman measurements and carry out further related experiments in our future work.

Additionally, we conducted solid-state UV-Vis spectrophotometry measurements on both blue and red phase PDAs. Initially, we attempted to prepare PDA particle-coated samples using the convective assembly method, where aqueous droplets containing PDA particles were placed

18
[Type here]

onto a glass substrate and allowed to dry under ambient conditions. The resulting PDA particle-coated film intrinsically exhibited a non-uniform surface due to its spherical geometry with a ~ 200 nm diameter. Consequently, the results of UV-Vis absorbance lacked consistency. It should be noted that such non-uniformity in film thickness can influence the absorption and reflection of light, complicating the resulting spectra and making accurate interpretation more challenging. Instead, we decided to prepare uniform PDA films on a glass substrate using the same protocol and environment as when fabricating the PDA particles via the microfluidic method. This was done to confirm whether our microfluidic-produced particles exhibited typical UV-Vis behaviors consistent with those of typical PDA materials. Specifically, we drop-casted a 10 wt% PCDA solution in chloroform onto a glass substrate and covered the drop with a 2 wt% PVA water solution, thereby mimicking the conditions used in the microfluidic fabrication. Once the solvent had completely evaporated, the PCDA film underwent polymerization under a UV lamp (254 nm VL-6LC, Vilber Lourmat, France) for 1, 5, and 10 min, respectively. The resulting blue phase film was then washed with DI water to remove any excess PVA. To prepare the red phase PDA film, the blue phase film was immersed in two different pH conditions, ~ 8.7 and ~ 9.7 , for 30 min. Upon conducting UV-Vis measurements of these samples, we found that they exhibited typical behaviors of PDA materials. The maximum absorbance occurred at ~ 640 nm for the blue phase samples and ~ 550 nm for the red phase samples, as displaced in Fig. R1.

[Image redacted]

Fig. R1 (Reviewer Only) Solid-state UV-Vis spectrophotometry of PDA films subjected to various conditions. Details on the preparation of these PDA films are provided in the preceding text.

2 The overall literature review is poor and focuses almost exclusively on recent and crude measurement of PDAs. I suggest referencing the following papers for the following sub-regions of PDAs mechanochromism:

* <https://pubs.acs.org/doi/10.1021/acs.nanolett.0c04027>

19

[Type here]

* <https://iopscience.iop.org/article/10.1088/0953-8984/16/23/R01>

* <https://onlinelibrary.wiley.com/doi/full/10.1002/admi.202201808>
origin of the blue to red transition

* <https://onlinelibrary.wiley.com/doi/abs/10.1002/3527607323.ch3>

* <https://pubs.acs.org/doi/10.1021/jp0638437>

* <https://www.sciencedirect.com/science/article/pii/S0009261499011379>

* <https://www.nature.com/articles/nphys196>

* <https://pubs.acs.org/doi/10.1021/la991580k>

* <https://pubs.rsc.org/en/content/articlelanding/2010/CP/B915527A>

<https://pubs.acs.org/doi/10.1021/la970831r>
<https://pubs.acs.org/doi/10.1021/la8029038>
<https://pubs.acs.org/doi/10.1021/la991580k>
<https://pubs.rsc.org/en/content/articlelanding/2007/cc/b703691d>
<https://pubs.acs.org/doi/10.1021/acs.macromol.0c00718>

We appreciate the valuable suggestion to enhance the literature review by referencing specific papers on the PDAs mechanochromism. We have revisited the reference section and incorporated key papers that address various aspects of PDAs' color changes, including the articles recommended by the reviewer.

3. It's unclear what the nanoparticles are doing to the PDA

- * are the particles impacting the PDA particle and then deflecting or embedding themselves into the particle?
- * Are they traveling through the PDA particle?
- * Some interesting tests would be to take a big particles (like 100-200 micron) and smash them into the PDA particle. This is almost something that could be examined using the classroom physics billiard balls momentum calculation.

We used the silica nanoparticles (SNPs) to investigate the physical impact on a PDA particle captured within the mechano-FL sensor system. After colliding with the PDA particle, most SNPs likely exit the sensor channel following the fluid flow. However, Raman spectroscopy measurements of the SNP-bombarded PDA particle suggest that some SNPs become embedded in the PDA layer (Supplementary Fig. 2).

Newly added Supplementary Fig. 2:

20
[Type here]

[Image redacted]

Supplementary Fig. 2 Raman spectrum of a PDA particle bombarded with SNPs during the mechano-FL sensor experiment. The presence of silica nano crystal peak in the bombarded PDA sample demonstrates that some SNPs became embedded in the PDA layer after colliding with the PDA particle.

We chose SNPs to identify key variable factors influencing the sensor system. SNPs are advantageous for these screening experiments because they are less expensive than RBCs, and there is no risk of degradation, making their storage and handling relatively convenient. The critical variable conditions identified during these experiments were subsequently applied to the mechanical stress tests involving RBCs. We have clarified this point in the revised manuscript.

Before: We elucidated the effect of flow-induced mechanical stress on the FL response of this sensor (Fig. 1c), along with its dependence on several experimental variables including flow velocity, solution viscosity, and concentration of solid objects (i.e., silica nanoparticles (SNPs), red blood cells (RBCs) and yeast cells) were evaluated.

After (p3): We elucidated the effect of flow-induced mechanical stress on the FL response of this sensor (Fig. 1c) and assessed its dependency on multiple experimental variables such as flow rate, solution viscosity, and the concentration of solid objects (i.e., silica nanoparticles (SNPs), red blood cells (RBCs) and yeast cells). We identified the key variable conditions using SNPs and subsequently applied these to the experiments with RBCs and yeast cells.

We find the reviewer's proposed experiment, in which a big PDA particle of similar size collides with another PDA particle captured within the mechano-FL sensor, to be quite intriguing. Such an experimental setup could potentially shed light on how locally initiated

mechanical stress propagates. However, this suggestion may not align well with the primary focus of our current study, which aims to investigate the mechanical stress caused by small components, such as RBCs, in flow channels. Therefore, we believe this experiment would be better suited for future follow-up studies, and we have noted this in the revised Discussion section.

21

[Type here]

Added statements (p21 in Discussion): Furthermore, we plan to initiate a single collision of a PDA particle (or other types of particles) to another PDA particle positioned within the mechano-FL sensor channel. By manipulating variables such as size, rigidity, and collision speed, we aim to compare the area where FL is emitted to the actual collision area.

The treatment of the blue to red transition in this work is unclear and at times contradictory.

* Figure 1 panel c suggests that distortion of the polymer backbone is responsible for the blue to red transition

* Page 6 Line 108-109 suggests that the blue to red transition is due to a transition from planar to non-planar geometry.

The authors should refer to the above references to clarify the following:

It's been well established that the red phase may be even more ordered than blue phase PDA, but simply non-planar. In some cases, it may be that an increase in disorder is associated with the blue to red transition. However, this is not exclusively true. Highly ordered red chains are well documented in literature. The authors should revise figure 1 and their statements to avoid making any conclusions on backbone conformation from these macro-scale measurements without significant experiment evidence.

See: the above references on the origin of the blue to red transition, this paper on a potentially disordered PDA <https://onlinelibrary.wiley.com/doi/abs/10.1002/adfm.201301044>

We are grateful for the reviewer's instructive comments concerning the treatment of the blue to red transition in our work. We have clarified the color transition mechanism and revised the description of the associated structural changes depicted in Fig 1c.

Before: The blue-to-red color change of the PDA particles is a consequence of conformational changes taking place in the highly conjugated backbone of the polymer, in which the blue-phase corresponds to a PDA with an intact and planar backbone structure, whereas the red-phase has a twisted nonplanar geometry (Fig. 1c).

After (p6): The blue-to-red color transition of the PDA particles is a consequence of conformational changes within the highly conjugated polymer backbone.⁴²⁻⁴⁵ In general, the blue phase represents a PDA state with a planar backbone structure, whereas the red phase exhibits a twisted, nonplanar geometry (Fig. 1c). Interestingly, some research has challenged the initial belief that disorder is a necessary condition for the red phase formation in PDA.^{46, 47}

22

[Type here]

Revised Fig. 1:

[Image redacted]

4- Furthermore, the authors make a very unusual series of conclusions based on their simulations regarding the torsion of PDAs. The packing and angles of PDAs at the molecular scale is not fully elucidated and is still relatively challenging to investigate even with atomistic simulations and molecular scale experiments. The CFD model and association molecular

torsion needs to be re-examined quite closely. To make those conclusions, the authors need to draw a molecular picture of what this might look like. The length scales between where this occurs (angstroms) to the 100s of microns over which this simulation is done, makes this whole assessment seemingly spurious. It is unclear how normal or shear stresses applied via particle bombardment or viscous stress is imparted onto the molecule in the form of torsion of the polymer backbone. Atomic force microscopy measurements show that torsion is not necessary, applying lateral stress along the PDA backbone is sufficient to induce the blue-to-red transition. It seems extremely difficult to correlate nano-scale phenomena with micron-scale continuum mechanics simulations.

Their energy calculations are also very unusual, they should report shear and normal stresses at the surfaces and compare to existing mechanical stress data found in:

<https://pubs.acs.org/doi/10.1021/acs.nanolett.0c04027>

<https://onlinelibrary.wiley.com/doi/full/10.1002/admi.202201808>

<https://pubs.acs.org/doi/10.1021/la9815998>

23

[Type here]

<https://doi.org/10.1023/A:1019113218650>

We acknowledge the possibility of errors in our initial attempt to explain the results obtained in the mechano-FL sensor experiments solely using the previously reported torsion energy model. We are also aware of the findings by Juhasz et al. (Nano Lett. 21, 543, 2021), which demonstrated, through well-controlled nanoscale experiments using friction force microscopy, that the lateral force component in thin PDA two-dimensional (2D) films plays a dominant role in PDA mechanochromism. However, unlike the relatively well-ordered 2D PDA films, our particle geometry is expected to be arranged in a more random fashion in three dimensions. Consequently, the force applied by the external physical impact on the PDA particle can act in complex directions relative to the PDA backbone. In addition, to our knowledge, there is no reported study showing that lateral stress applied to the PDA backbone induces nonlinear mechanochromism. Therefore, we believe it would be premature to conclude that the results of Juhasz et al. (i.e., the dominant contribution of the lateral stress) can be solely and directly applied to our system.

Note that the nonlinear increase of torsion energy in relation to the torsion angle shows a resemblance to the FL intensity profiles we experimentally obtained (as depicted in the inset of Fig. 5f). This similarity allows us to reasonably postulate a close correlation between these two factors. Nonetheless, we have specified in our revised manuscript that not only torsional but also lateral stresses could play a significant role in mechanochromism. We have added related references accordingly and removed the quantitative torsion energy calculation parts to avoid any unnecessary misunderstanding.

Additionally, activation energy values for the complete blue-to-red transition, as reported for various types and conditions of PDA materials, are around 17.6 to 22.5 kcal/mol [Carpick et al., Langmuir 2000, 16, 4639]. Considering the ~27% polymerization of our experimentally derived PDA particles, the activation energy required for the PDA particle's blue-to-red transition is estimated to be ~289 to 369 kJ/mol. This result aligns well with the energy requirement of 307 kJ/mol predicted via our calculations. We have updated the manuscript to emphasize and clarify these points.

Furthermore, as suggested by this reviewer, we have incorporated the CFD calculation results for normal and shear stresses applied to the PDA particle surface in the revised Fig. 5b and 5c. The stress scale obtained in these calculations (~25 kPa for normal stress and ~1 kPa for shear stress) is comparable to the scales obtained through the surface forces apparatus for PDA Langmuir films upon the blue-to-red transition (~100 kPa for normal stress and ~3 kPa for shear stress) (Adv. Mater. Interfaces 10, 2201808, 2023).

Before: The calculations show that bombardment by the SNPs leads to an increase in energy absorbed by the PDA particle, considered to be the sum of the computed energy and SNP collision energy calculated using the Hertz contact equation (Table S6). As shown in Fig. 5e, energy absorbed increases in a linear manner upon increasing the concentration of the SNP in the 0, 5.65, 11.3 and 16.95 wt.% range injected for 4.5 min. Similarly, because the conjugated structure of the PDA was not considered in this simulation, the energy values do not accurately reflect the observed nonlinear SNP concentration dependent increase in FL intensities. However, the

24

[Type here]

sudden increase in the FL intensity observed in 11.3 to 16.95 wt.% region suggests that a certain critical torsion energy threshold exists between 33.75 and 50.73 \hat{J} .

To determine the critical torsion energy level, δ_{tor} values were calculated as a function of the SNP injection time and then compared to the torsion energy δ_{tor} (Tables S7 and S8).^{46, 47} The calculated variation in torsion energy as a function of the δ_{tor} and the SNP injection time (t) show that two rapid increases in torsion energy take place in the time intervals of 60 s \hat{t} \hat{t} 120 s and 210 s \hat{t} \hat{t} 270 s, as indicated by the orange bars in Fig. 5f. The corresponding torsion angles are 11 $^{\circ}$ \hat{t} \hat{t} 22 $^{\circ}$ and 39 $^{\circ}$ \hat{t} \hat{t} 50 $^{\circ}$, which are consistent with PDA polymorphs displaying most prominent FL emission.⁴⁶ The calculated nonlinear δ_{tor} profile is in good agreement with the experimentally observed FL intensity profile, demonstrating the role played by the unique structural characteristics of the PDA particle in determining FL emission. In addition, because the δ_{tor} value obtained from the CFD simulation is the same order of magnitude as δ_{tor} , the range of $\sim 40\hat{t}50 \hat{J}$ is estimated to be the critical energy level, around which the PDA particle undergoes a notable FL intensity increase.

After (p18): The calculations show that bombardment by SNPs leads to an increase in δ_{tot} absorbed by the PDA particle, considered to be the sum of the computed δ_{H} and SNP collision energy δ_{col} calculated using the Hertz contact equation (Supplementary Table 7).⁶⁰ δ_{tot} demonstrates an increasing trend as the SNP concentration increases, spanning 0, 5.65, 11.3, and 16.95 wt.% when injected for 4.5 min (Fig. 5e) and with an extended injection duration of 8.5 min at 16.95 wt.% concentration (Fig. 5f). Similarly, because the conjugated structure of the PDA was not considered in this simulation, the δ_{tot} values do not accurately reflect the observed nonlinear behavior of FL intensities dependent on the SNP concentration.

Nevertheless, this computational data suggests that the δ_{tot} required to achieve a complete blue-to-red transition for a single PDA particle is approximately 307 \hat{J} . The activation energy δ_{act} for irreversible blue-to-red transitions in a variety of PDAs has been estimated to be $\sim 17.6\hat{t}22.5 \text{ kcal}\hat{t}\text{mol}^{-1}$.⁴⁴ For instance, for a three-layer PDA film fabricated from the PCDA monomers, the predicted δ_{act} was approximately 17.6 $\text{kcal}\hat{t}\text{mol}^{-1}$. Though our 200 $\hat{t}4\text{m}$ PDA particles might differ from the previously reported PDA conditions, we presumed $\delta_{\text{act}} \hat{t} 17.6\hat{t}22.5 \text{ kcal}\hat{t}\text{mol}^{-1}$ to roughly estimate δ_{act} for our PDA particles. Considering $\sim 27\%$

$\delta_{\text{act}} \hat{t} \delta_{\text{PDA}}$

polymerization (Fig. 2d) yields 0.27 \hat{t} PDA $\delta_{\text{act}} = 289\hat{t}369 \hat{J}$, where the PDA density

δ_{PCDA}

$3 \hat{t} 4\hat{t}3$

is $\delta_{\text{PDA}} \hat{t} 1.3 \hat{t} 10 \text{ kg}\hat{t}\text{m}^{-3}$, the PDA volume is $\delta_{\text{PDA}} = 3$, and the molecular weight of PCDA is $\delta_{\text{PCDA}} = 0.375 \text{ kg}\hat{t}\text{mol}^{-1}$. This result aligns well with the energy requirement of $\sim 307 \hat{J}$ for the complete blue-to-red transition, as derived from our calculations.

The nonlinear FL intensity profile possesses a resemblance to the previously reported torsional energy δ_{tor} calculations, which also exhibit nonlinear increases for various types of PDA materials.^{53, 54} Therefore, we compared the normalized δ_{tor} profile with the normalized FL intensity profiles derived from both the mechano-FL sensor experiment (Fig. 5f) and the thermal stress analysis (Fig. 2g). As displayed in the inset of Fig. 5f, the experimental FL intensity's nonlinear trajectory seems fairly consistent with the shape of δ_{tor} . This similarity allows us to reasonably postulate a close correlation between these two factors. Notably, recent experimental results have demonstrated that lateral stress applied parallel to the PDA backbone significantly influences the mechanochromism of thin PDA films.⁶¹ Therefore, further

25

[Type here]

experimental and/or theoretical studies are needed to clarify how torsional and lateral stresses contribute relative to each other depending on the PDA structures under various conditions.

Added paragraph (p17): The results of these calculations clarify the dynamic behavior of the mechano-FL sensor system, as illustrated in Fig. 5a-c. Specifically, Fig. 5a provides a detailed visual representation of the flow velocity patterns observed when injecting a 16.95 wt.% SNP solution with a flow velocity of 3.317 $\text{m}\hat{t}\text{s}^{-1}$ (Supplementary Table 2). The maximum flow velocities are found to be approximately 3 $\text{m}\hat{t}\text{s}^{-1}$ in the microchannel following impact with the PDA particle surface. Fig. 5b visualizes the corresponding shear and normal stress distributions. Notably, the normal stress peaks at a polar angle of $\delta_{\text{P}} = 0^{\circ}$, which is the point where the SNPs collide perpendicularly with the PDA particle, as depicted in Fig. 5c. In contrast, the

shear stress reaches a minimum value at the same polar angle ($\delta P = 0A^\circ$) and increases as the angle δP widens. Note that the stress scale obtained in these calculations is comparable to the scales obtained through the surface forces apparatus for PDA Langmuir films upon the blue-to-red transition (i.e., ~ 100 kPa for normal stress and ~ 3 kPa for shear stress).⁵⁹

Revised Fig. 5:

[Image redacted]

Fig. 5 CFD analysis of the mechano-FL sensor system. a,b Examples showing distributions of flow velocity (a) and stress (b) for the 16.95 wt.% SNP suspension. c Variation of shear and normal stresses on the PDA surface as a function of the polar angle δP . Error bars represent the standard deviation over the stress values across the azimuthal angle $\delta \frac{1}{4}$. d,e Plots of δ_{H} and δ_{tot} as functions of glycerin (d) and SNPs (e) concentrations, respectively, at an injection rate of $25 \text{ mL} \cdot \text{min}^{-1}$ over 4.5 min. Corresponding experimentally measured FL intensity profiles are overlaid for comparison. f Time-dependent profiles of δ_{tot} and FL intensity for the 16.95 wt.% SNP suspension injected at $25 \text{ mL} \cdot \text{min}^{-1}$ over an extended duration of 8.5 min. Inset indicates the comparison of the nonlinear property of normalized FL intensity profiles from the mechano-FL sensor experiment and the thermal stress analysis (from Fig. 2g) against the normalized torsion energy δ_{tor} from a previous study.⁵⁴ The x-axis is normalized from 0 to 1,

26
[Type here]

aligning maximum values of FL intensity and δ_{tor} to 1. For panels d-f, each experimental data point and corresponding error bar represent the mean intensity value of a minimum of three independent trials and its standard deviation, respectively.

5-The torsional calculations are naive. From the supplemental, it seems that they assume the entire particle is polymerized in the blue phase, prior to application of mechanical stress. It is well known that significant amounts of monomer remain in PDAs as only the topochemically aligned structures are capable of polymerizing. Even in well packed, ordered Langmuir films, polymerization is generally incomplete, especially in the blue phase. Complete polymerization is usually only achieved through prolonged irradiation of the polymer and in the red phase. Hence, assuming the entire particle polymerized is completely polymerized is unrealistic, especially without supporting evidence. The easiest way to validate this is to expose to a calibration uv light source and then measure the absorbance spectra as a function of UV light exposure. The maximum intensity of the particle will very likely be in the red phase.

We thank the reviewer for pointing out the concern regarding the incomplete polymerization of PDA particles. As noted, due to the finite penetration depth of UV irradiation, the PDA particles underwent partial polymerization, as previously reported by Spagnoli et al. (Langmuir 33, 1419-1426, 2017). To assess the degree of polymerization within the particles we used in our experiments, we applied sufficient thermal stress to the PDA particles and observed their fluorescence using a confocal microscope. As depicted in the revised Fig. 2d, when the focal plane was set at the center of the PDA particle, fluorescence was observed only at its periphery. Through image analysis, we estimated that the polymerized shell has a thickness of approximately $10 \text{ } \mu\text{m}$. When translated to the volume of a $200 \text{ } \mu\text{m}$ -sized PDA particle, this corresponds to a maximum volume degree of polymerization of $\sim 27\%$. Furthermore, SEM analyses of bisected PDA particles reaffirmed the partial polymerization of the PDA particles. The interior of the bisected particle appeared to be filled with a solid substance before chloroform treatment, as seen in Fig. 2e. However, after chloroform treatment, the interior appeared hollow, as shown in Fig. 2f. These results demonstrate that the center region of the particle remained unpolymerized and contained residual PCDA monomers. We have updated these results in the revised manuscript.

Added paragraph (p6): Importantly, due to the finite penetration depth of the UV irradiation, the PDA particles underwent partial polymerization.³⁹ As evident from the confocal images of the red phase PDA particles in Fig. 2d, the polymerized shell is clearly visible with an approximate thickness of $h_{\text{shell}} \hat{=} 10.0 \pm 0.03 \text{ } \mu\text{m}$, as averaged over nine different particles. This

corresponds to a maximum volume degree of polymerization of ~27%. Furthermore, SEM analyses of bisected PDA particles reaffirmed the partial polymerization of the PDA particles. The interior of the bisected particle appeared to be filled with a solid substance before the chloroform treatment (Fig. 2e) but appeared hollow after the treatment (Fig. 2f). Given that PCDA monomers are soluble in chloroform, these results demonstrate that the center region of the particle remained unpolymerized and contained residual PCDA monomers.

27

[Type here]

Added statements (p17): The activation energy δ_{act} for irreversible blue-to-red transitions in a variety of PDAs has been estimated to be $\sim 17.6 \pm 2.5 \text{ kcal} \cdot \text{mol}^{-1}$.⁴⁴ For instance, for a three-layer PDA film fabricated from the PCDA monomers, the predicted δ_{act} was approximately $17.6 \text{ kcal} \cdot \text{mol}^{-1}$. Though our 200 nm PDA particles might differ from the previously reported PDA conditions, we presumed $\delta_{act} \sim 17.6 \pm 2.5 \text{ kcal} \cdot \text{mol}^{-1}$ to roughly estimate δ_{act} for our δ_{PDA} PDA particles. Considering ~27% polymerization (Fig. 2d) yields 0.27 PDA $\delta_{act} =$ δ_{PCDA} $289 \pm 369 \text{ J}$, where the PDA density is $\delta_{PDA} \sim 1.3 \times 10^3 \text{ kg} \cdot \text{m}^{-3}$, the PDA volume is $\delta_{PDA} = 4 \times 10^{-3}$, and the molecular weight of PCDA is $\delta_{PCDA} = 0.375 \text{ kg} \cdot \text{mol}^{-1}$. This result aligns well with the energy requirement of $\sim 307 \text{ J}$ for the complete blue-to-red transition, as derived from our calculations.

Revised Fig. 2:

[Image redacted]

Fig. 2 Structural and optical properties of PDA particles. a,b Optical microscopic images of PCDA-chloroform emulsions in water (a) and solid PDA particles in water (b). c SEM image of PDA particles. d Confocal microscope image of a red phase PDA particle heated at $80 \text{ }^\circ\text{C}$ for 10 min, revealing a polymerized shell of a thickness h_{shell} . Two additional images from different particles are shown below. e,f SEM images of a bisected PDA particle before (e) and after (f) chloroform treatment. g Thermo-FL response of PDA particles subjected to gradual

28

[Type here]

heating from $40 \text{ }^\circ\text{C}$ to $100 \text{ }^\circ\text{C}$. h Raman spectra of PCDA monomers (i), a blue PDA particle (ii), a red PDA particle produced by heating at $80 \text{ }^\circ\text{C}$ for 10 min (iii), and a mechanically crushed PDA particle (iv).

6-Are these solid particles of PDA? What is inside? Figure 2d doesn't give a complete picture of the internal structure, or what that internal structure is. The classic liposome cross section often shown in SEM images of vesicles should be utilized here.

7-Additionally, if it is assumed that the PDA is fully polymerized, the domains should be readily viewable with a polarized light microscope. Were there domains large enough to be observed?

The comments-6 and -7 have been addressed in our response to this reviewer's 5th comment.

8-The concept of the blue to red transition propagating through the particle needs further examination, especially given the size of the particle. It should be readily viewable how the blue to red transition occurs spatially within the particle and yet the data they show is not really clear. Figure 4 e is the most clear, but experimental details are not clear enough. Experiment to try: what if they injected nanoparticles for a second, then stopped adding nanoparticles: does

the fluorescence continue to grow? Then it propagates, does it stop after flow of nanoparticles stops?

Text says "this phenomenon is likely the result of mechanical stress propagation." I think this part should be reviewed

We thank the reviewer for pointing out the important aspect of mechanical stress propagation. As suggested by this reviewer, we subjected a PDA particle to a collision by injecting 16.95 wt.% SNPs for 4.5 min, and then we stopped the injection and continued to monitor any changes in the FL intensity. As shown in Supplementary Fig. 10, the changes in FL intensity observed post-collision were negligible. Compared to the complete transition after an extended collision lasting 8.5 min (Fig. 5f and Supplementary Fig. 9), we surmise that for the energy applied locally to a PDA particle to spread across the entire region, it would necessitate a consistent and accumulating external energy input.

In our original manuscript, based on experimental data such as the observation of gradual FL changes in PDA particles over time in our mechano-FL sensor experiments (Fig. 4c), FL emission from entire particles under prolonged stress exposure (Fig. 5f), and the SEM image analysis of an isolated PDA particle post-SNPs collision showing local indentations focused on the center region of the particle (Fig. 3e-g), we inferred that the FL in entire PDA particles might originate from localized mechanical stress. This has led us to speculate that mechanical stress could propagate in a finite manner. Nonetheless, a more in-depth study is needed to explore the intricacies of this propagation process. For this, we are planning the following subsequent studies:

29

[Type here]

(1) We plan to utilize optical laser tweezers apparatus [Nat. Commun. 14, 3838, 2023; J. Phys. Chem. Lett. 13, 10018, 2022]. This can optically trap a polymeric microparticle and apply oscillatory impacts to the surface of a PDA particle anchored to a substrate. Given that optical laser tweezers can precisely control the applied force down to the piconewton scale, we believe that sufficient mechanical stress can be exerted on a thin-layered PDA particle.

(2) We intend to induce a single collision of a PDA particle (or other types of particles) to another PDA particle positioned within the mechano-FL sensor channel. By controlling the size, rigidity, and collision speed of the colliding particles, we aim to compare the area where FL is emitted to the actual collision area.

In light of these considerations, we have revised the manuscript and also updated the explanation part of Fig. 4.

Before: Finally, the confocal image given in the inset in Fig. 3f shows that the complete PDA particle fluoresces in the red region after injection of the SNPs for 4.5 min, demonstrating the applied mechanical stress propagates throughout the entire particle.

After (p12): Finally, the confocal image shown in the inset of Fig. 3e reveals that the entire surface region of the PDA particle fluoresces after injection of the SNPs for 4.5 min. This illustrates how the localized mechanical stress, when continuously applied and accumulated, can affect the entire region of the PDA particle.

Before: The time-dependence of the response of the new sensor system was examined by continuously monitoring FL intensities at times following their injection (Fig. 3e and Fig. 4). We found that as the flow injection time increases the consequent FL intensity increases. For example, in contrast to the negligible effect caused by pure water, injection of 16.95 wt.% SNP promotes the most prominent time dependent FL intensity increases. Interestingly, as the SNP injection time is increased, the area over which FL emission occurs gradually increases (Fig. 4e). Considering that the location of the impact of the SNPs remains constant throughout the injection time period, this phenomenon is likely the result of mechanical stress propagation. Notably, using the 16.95 wt.% SNP solution, the FL intensity increases nonlinearly, and two sharp FL increases occur using injection times of ~1 min and ~4 min (Fig. 3e).

After (p12): The time-dependent response of the mechano-FL sensor system was examined by continuously monitoring FL intensities at various time points following injection (Fig. 3h and

Fig. 4). We found that as the flow injection time increases the consequent FL intensity increases. For example, in contrast to the negligible effect caused by pure water (Fig. 4a) and relatively small FL intensity changes for the case of 20 wt.% glycerin (Fig. 4b), injection of 16.95 wt.% SNP promotes the most prominent time dependent FL intensity increases. Interestingly, as the SNP injection time is increased, the area of FL emission also gradually increases (Fig. 4c). Given that the impact location of SNPs remains constant throughout the injection, the continuous SNPs collision likely results in accumulated mechanical stress, causing the entire

30

[Type here]

particle to fluorescence. Notably, using the 16.95 wt.% SNP solution, the FL intensity increases nonlinearly (Fig. 3h).

Added paragraph (p13): We conducted further experiments to elucidate the effect of physical impact of SNPs on PDA particles on FL emission. The time-dependent FL response at different SNPs concentrations (5.65, 11.3, and 16.95 wt.%) was compared, as shown in Supplementary Fig. 8a. It was observed that the FL intensity value rapidly increased with time at higher concentrations, while the change was relatively minor at lower concentrations. The FL intensity change was then re-plotted against the number of injected SNPs, resulting in the data for the three different concentrations converging to a single master curve (Supplementary Fig. 8b). Additionally, we extended the injection time of 16.95 wt.% SNPs to 8.5 min (510 s) and found that the normalized FL intensity approached 1, indicating near saturation (Supplementary Fig. 9). Moreover, a 16.95 wt.% SNPs solution was injected and halted at 4.5 min (270 s), and the FL intensity was continuously monitored even after stopping the injection. It was found that the FL change after stopping was negligible (Supplementary Fig. 10). Based on these additional analyses, we confirmed that the FL intensity values are primarily influenced by the number of SNPs colliding with the PDA particles. Furthermore, the gradual changes in PDA mechanochromism, along with the halt of FL intensity changes upon the removal of mechanical stress, underscore the important role that continuous application and accumulation of mechanical stress play in mechanochromism. This might be closely related to the nonlinear behavior of PDA mechanochromism.^{53, 54} For more information on the injected fluids used in this study, one can refer to Supplementary Table 2.

Newly added Supplementary Fig. 10:

[Image redacted]

Supplementary Fig. 10 Monitoring of FL intensity change of the mechano-FL sensor system over time after stopping SNPs injection. A 16.95 wt.% SNPs solution was injected and halted at 4.5 min, with the injection and suction rates maintained at 25 mL·min⁻¹ and 5 mL·min⁻¹, respectively. The FL intensity was continuously observed even after stopping the injection, and it was found to be negligibly altered. The error bars indicate the standard deviations derived from at least three independent trials.

31

[Type here]

9-The particles shown in Figure 2c look completely different from the particle in figure 3f. What happened to the fuzzy domains? Are they stripped off? Furthermore the references used in line 92 (37, 38) are fine, but don't really explain the fuzzy polymer domains that form, and neither reference refers to the domains they investigated as "plate-like". The authors should investigate the following reference, where giant vesicles covered in similar domains were investigated.

See: <https://onlinelibrary.wiley.com/doi/abs/10.1002/adfm.200700726>

As per this reviewer's observation, the particle shown in Fig. 3e of the revised manuscript indeed appears to have been stripped off due to mechanical stress. We have clarified this point in the revised manuscript. Furthermore, we have updated the reference concerning the PDA particle's morphology as suggested by this reviewer.

Before: Inspection of a SEM image shows that the surface and interior of the PDA particles have lamellar structures (Fig. 2c, d), which is the typical plate-like morphology of PDAs formed by photopolymerization of self-assembled PCDA monomers.^{37, 38}

After (p6): Upon inspection of SEM images, it is apparent that the surface of the PDA particles features fuzzy domains, as depicted in Fig. 2c. This characteristic morphology has a resemblance to the structure of giant lipid/PDA vesicles, prepared using a simple solvent evaporation method.³⁸

Before: Also, the images show that several cracks radiate outward from this central area, and that numerous lamellar surface structures on the intact PDA particle have been mechanically etched as a consequence of the normal and shear stresses applied to the PDA surface during the SNPs injection.

After (p12): Also, the images show that several cracks radiate outward from this central area. Interestingly, numerous fuzzy domains present on the intact PDA particle, as shown in Fig. 2c, appear to have been mechanically etched as a consequence of mechanical stress applied to the PDA surface during the SNPs injection.

Additional Comments regarding the manuscript that should be clarified.

The microfluidic channel approach shows promise as an interesting way to both visualize and quantify the role of shear and normal stress in driving blue to red transitions. The particle synthesis approach potentially opens the door to further quantification of mechanical stress induced fluorescence in PDAs through a wide variety of traditional single molecule and microfluidic approaches for example optical tweezers, micropipette aspiration, etc.

We are grateful to this reviewer for emphasizing the significant contribution of our developed mechano-FL sensor system in visualizing localized mechanical stress. We also appreciate the

32
[Type here]

feedback on the convenience of our method for fabricating PDA particles using microfluidic devices. As the reviewer has mentioned, we are currently conducting follow-up studies on the mechanochromism and solvatochromism of PDA particles through precise position control of individual particles using optical laser tweezers.

10-The role of pH salt concentration in the blue-to-red transition needs further explanation. The authors are clearly aware of the effects that basic pH (8 to 9) have on PDAs, however their literature review of this portion of the PDA field is quite narrow and would provide significant insights into why this is occurring. I would suggest investigating and potentially citing some of the following papers. PDAs can be quite sensitive to changes in ionic strength, and pH. Furthermore their work with silica nanoparticles raises several questions

see:

<https://pubs.acs.org/doi/full/10.1021/ac0517794>,

<https://pubs.acs.org/doi/full/10.1021/ja0299001>,

<https://www.sciencedirect.com/science/article/pii/S0927775720302429>

among many many others...

We thank the reviewer for the comment and suggestions regarding the role of pH and salt concentrations in the blue-to-red transition of PDAs. We have added references in the revised manuscript, as the reviewer suggested.

Similar to how the blue-to-red transition phenomenon in typical PDA materials is sensitively

affected by pH, we anticipate that the PDA particles we used would also exhibit a similar trend. However, in our study, we focused on the physical impact of SNPs collisions on a PDA particle, and in order to minimize undesired color-change factors, we used a buffer solution instead of using the original SNPs solution. Importantly, as seen in Supplementary Table 1, the FL intensity values for all SNPs cases in the buffer environment we used for the mechano-FL sensor experiments were approximately in the range of 0.0098 to 0.0132 a.u. These values were sufficiently minor. The reason we chose to use LUDOX TM-40 was that its FL intensity was relatively the lowest in the buffer environment, although using other types of SNPs would not have resulted in any significant changes in the mechano-FL sensor experiment.

Before: This effect is likely caused by the presence of anionic components in the mixtures, which promote a high pH triggering PDA response.⁴² As a result, in the optimized vascular mimic system we used a 20 mM sodium phosphate buffer to mitigate the pH effect (Fig. S4c and Table S1) and LUDOX TM-40 (diameter ~34 nm), which displays a minimum FL intensity response upon static incubation.

After (p8): This effect is likely caused by the presence of anionic components in the mixtures, which promote a pH triggering PDA response.⁴⁸⁻⁵¹ The use of a 20 mM phosphate buffer significantly reduced the FL intensity of the PDA particles, with values in the range of 0.0098 to 0.0132 a.u. for all types of SNPs, which were sufficiently minor (Supplementary Fig. 6c and Supplementary Table 1). Consequently, we chose to use LUDOX TM-40 (diameter ~19.6 Å ±

33

[Type here]

5.6 nm, Supplementary Fig. 7), which displayed the relatively lowest FL intensity in the buffer environment. However, the use of other types of SNPs would not have resulted in any significant differences in the mechano-FL sensor experiment. Additionally, before proceeding with the mechano-FL sensor experiments, we confirmed that all the solutions used in the experiments did not induce significant changes in the FL emission of the PDA particles under static conditions.

11-When PDA particles are buffered down to around pH 7: do they aggregate? The stock solutions are quite basic. Were PDA particles and the buffer, glycerin or other solvents just added together and allowed to equilibrate? Did the PDAs slowly transform?

The PDA particles we used have a size of 200 nm, causing them to immediately sediment. This prevents us from directly observing whether they spontaneously form aggregates when dispersed in a solution. However, when we gently shook a container holding multiple PDA particles which initially appeared to be in contact with each other at a pH of ~7 the particles readily separated from one another. This observation suggests that the particles do not form aggregates under the condition. Furthermore, prior to conducting our mechano-FL sensor experiments under all fluid conditions, we verified in advance that the color changes in the PDA particles were negligible under static conditions.

Added statement (p9): Additionally, before proceeding with the mechano-FL sensor experiments, we confirmed that all the solutions used in the experiments did not induce significant changes in the FL emission of the PDA particles under static conditions.

12-The particle size distributions are not given for these nanoparticles. These are commercially available. A quick search of the product pages of these nanoparticles does not yield DLS or SEM characterization. The authors should strongly consider adding DLS particle size distributions (with full cumulants analysis and multimodal size distributions properly weighted as volume, intensity and number in the supplemental), both as stock and when buffered.

As the reviewer's suggestion, we have performed the DLS measurements of the 19.95 wt.% SNPs condition that we used in our experiments and have added these results as Supplementary Fig. 7. However, when the as-purchased SNPs solution (~40 wt.%) was measured directly without dilution, the results differed from those under our experimental conditions. Notably, the intensity distribution exhibited two peaks (Fig. R2). We believe that this discrepancy may

be attributed to various issues (e.g., aggregation, multiple scattering, etc.) that can arise during DLS measurements at high concentrations, which are likely to introduce errors.

Newly added Supplementary Fig. 7:

[Image redacted]

34

[Type here]

Supplementary Fig. 7 DLS measurements of a LUDOX TM-40 SNPs solution. a-c Volume (a), intensity (b), and number (c) percentage distributions of 16.95 wt.% SNPs in a 20 mM phosphate buffer solution. Measurements were performed using a Zetasizer (ZEN3600 Zetasizer Nano ZS, Malvern Instruments Ltd., UK).

[Image redacted]

Fig. R2 (Reviewer Only) DLS measurements of the as-purchased SNPs solution at a concentration of 40 wt.%.

13-Viscosity is an important parameter that is discussed in these experiments. However, there is no mention of how viscosity of any samples was measured (using say a rheometer), how it was calculated, or what references were used. Please provide references, and experimental details for all viscosity calculations. If the viscosity was measured, error and uncertainty should be reported (Figure 3 a-d).

We measured the viscosity experimentally using a Brookfield rotary viscometer (LV DV-I PRIME, Brookfield Engineering Laboratories, Inc., USA) equipped with a low-viscosity cylindrical spindle (LV-1 standard, Brookfield Engineering Laboratories, Inc., USA). Error bars corresponding to each measurement have been included in the Fig. 3c,d.

Added statements (p3 in SI):

3. Measurement of fluid viscosity

To determine the viscosity of the fluids, we utilized a Brookfield rotary viscometer (LV DV-I PRIME, Brookfield Engineering Laboratories, Inc., USA) equipped with a low viscosity cylindrical spindle (LV-1 standard, Brookfield Engineering Laboratories, Inc., USA). The measurements were performed at least three times to ensure accuracy and reproducibility.

Updated Fig. 3c,d:

35

[Type here]

[Image redacted]

14-The microfluidic setup is not described in sufficient detail. What Reynolds numbers were calculated? What flow regime was this system operated in? Significant program-specific jargon related to the $\hat{\text{a}}\text{Fluent}\hat{\text{a}}$ CFD package is referenced, but standard parameters for flow behavior within the channel are not described in the manuscript or supplemental. Furthermore, this model seems awfully complicated for what could maybe be approximated with back of the envelope calculations from a fluids textbook.

Regarding this comment, we direct this reviewer to our detailed responses to comments-4 and -7 of the 1st reviewer.

15-The figures need significant work. The color scheme and resolution on many of the figures is so poor that the data cannot be properly interpreted. Figure 3: Markers, error bars and viscosity bars need to be revised. Markers are too big and error bars are too small. The viscosity bars are too lightly shaded to be easily seen and compared between figures. The resolution of

the figures is terrible. Many micrographs are missing scale bars making interpretation difficult. Figure 5: too blurry to make out anything from the CFD simulations. The jet color scheme is misleading and a perceptually uniform colormap should be used instead. Velocity and pressure distributions are too small to be interpreted. Figure 2. panel e doesn't tell us anything that the inset doesn't already tell us. It's also too small and blurry to be useful.

In response to the reviewer's comments, we have improved the readability of all figures in both the revised manuscript and SI. We have also added missing scale bars and error bars for enhanced clarity. To improve resolution, all images in the revised manuscript and SI were inserted in PDF format into the MS Word document.

16-With regards to the PDA particle size: what governs it? The microfluidic system? Solvent choice? Solvent choice of chloroform is interesting as it is denser than water.

A key advantage of our microfluidic device is its ability to offer precise control over the size

36

[Type here]

of PDA particles, ranging from tens to hundreds of micrometers (Fig. R3a-d). This is achieved by manipulating the flow rates introduced into the microfluidic system or by altering the geometry of the constituent capillary tubes [J. Colloid Interface Sci. 560, 838, 2020]. For the solvent, we used chloroform due to its effective solubilization of PCDA monomers, its immiscibility with water (which serves as the continuous phase), and its high volatility. Another important feature of chloroform is its higher density compared to water. When chloroform droplets containing PCDA are collected in an aqueous medium, they sink to the bottom of the collecting container. In this water-immersed environment, the chloroform can evaporate completely, leaving behind solidified PCDA particles. Using a solvent with a lower density than water would risk having the droplets float on the water surface, which could result in droplet rupture.

Added statements (p20 in Discussion): Given the diverse application requirements, our microfluidic setup allows us the flexibility to produce PDA particles of different sizes, ranging from tens to hundreds of micrometers. This can be achieved by adjusting the flow rates fed into the microfluidic device or by altering the geometry of the capillary tubes constituting the device.64

[Image redacted]

Fig. R3 (Reviewer Only). PDA particles with various sizes fabricated using the co-flow microfluidic approach (a-d) and the emulsion method (e). The scale bars are 100 μm .

17-Please provide a detailed description of the confocal microscope used to acquire the inset in figure 3 either in the experimental section or in the supplemental information. How were confocal images processed?

We have expanded the SI to include detailed descriptions of all methods used for characterizing PDA particles, not just the confocal measurements. Additionally, we have provided detailed information on the image processing method used to obtain the FL intensity data. For more on these additional details and enhancements, we refer the reviewer to our response to the 1st comment of this reviewer.

18-The discussion section is incredibly short and reads more like a conclusions section. Why

37

[Type here]

does anyone care about this beyond some high level discussion of arterial function? I'm unsure if this will appeal to the general audience of Nat. Comm.

We thank the reviewer for the critical comment. We have significantly expanded and rewritten the Discussion section to emphasize the broader implications and potential applications of our research findings. Specifically, we have critically evaluated the limitations of our current mechano-FL system and proposed concrete methods to overcome these challenges. We have also added suggestions for future work, focusing on the propagation of mechanical stress at the macroscopic scale, an area that has not been systematically reported on to date. We believe that these revisions make the study more appealing to the general audience of Nature Communications.

Revised Discussion section:

Discussion

Mechanical stress is a key factor to the incidence of plaque rupture in blood arteries and the formation of emboli, both of which lead to severe cardiovascular and neurological disorders. Despite extensive study in the area of hemodynamics to comprehend the impact of mechanical stress on stenotic vessels, visualizing and accurately quantifying flow-induced mechanical stress inside stenotic channels in real time remains challenging. To address this, we performed a proof-of-concept study to demonstrate the potential of a single PDA particle-based mechano-FL sensor system for effectively analyzing and visualizing flow-induced mechanical stresses in stenotic vessel-mimicking microfluidic channels. Using uniformly sized PDA particles fabricated through the co-flow focusing microfluidic method, we could directly monitor mechanical stress. Among various parameters, such as fluid velocity, viscosity, and the concentration of particulate objects, the most significant factor influencing the FL emission of the PDA particle was the physical impact of solid particles. Notably, we observed nonlinear behaviors in FL intensity. This behavior closely resembled the previously reported energy changes based on the torsion angle of the PDA backbone. Given these similarities, we hypothesized a direct link between the two phenomena. However, considering our experimental conditions, which utilized spherical particles with a polymerized layer of $\sim 10 \text{ \AA}$ composed of PCDA monomers, clear quantification of torsion energy and its relation to the mechanical energy applied to the PDA particle will be crucial in future studies. We believe this study represents a significant advancement in the development of minimally invasive force/stress measurement in vitro systems that simulate hard-to-reach spaces within the human body. However, we recognize that several aspects in our current system need refinement. For instance, positioning the PDA probe particle within a narrowed channel could influence fluid behaviors. As illustrated in Supplementary Fig. 14, one potential solution is to embed the PDA particles into simulated vessel walls using materials such as hydrogels or polydimethylsiloxane (PDMS). To more accurately simulate actual blood flows in stenosed vessels, a 3D-printed vascular stenosis model can be employed.¹⁰ The sensitivity of the PDA particles as sensors could be enhanced by exploring different diacetylene monomers.^{41, 62, 63} A more straightforward approach might involve adjusting the UV exposure time for polymerization, thus controlling the degree of PDA particle polymerization. Given the diverse application requirements, our microfluidic setup allows us the flexibility to produce

38

[Type here]

PDA particles of different sizes, ranging from tens to hundreds of micrometers. This can be achieved by adjusting the flow rates fed into the microfluidic device or by altering the geometry of the capillary tubes constituting the device.⁶⁴

Lastly, the system we developed offers a promising experimental methodology to further understandings of the unique phenomena related to PDAs that remain elusive. For instance, we still lack a clear understanding of how mechanical stress, when applied locally, propagates through the PDA's conjugated system. When considering a linearly arranged PDA main chain backbone of finite length, it is unlikely to expect that mechanical energy introduced at one end would spontaneously propagate, like dominos, to the opposite end. This notion is evident from what we observed experimentally in Supplementary Fig. 10. Therefore, we surmise that for the energy applied locally to a PDA particle to spread across the entire region, it would necessitate a consistent and accumulating external energy input. In our pursuit to understand the propagation of mechanical stress at micro and/or macroscopic scales, we plan to utilize optical laser tweezers apparatus.⁶⁵⁻⁶⁷ This can optically trap a polymeric microparticle and apply oscillatory impacts to the surface of a PDA particle anchored to a substrate. Given

that optical laser tweezers can precisely control the applied force down to the piconewton scale, we believe that sufficient mechanical stress can be exerted on thin-layered PDA particles. Furthermore, we plan to initiate a single collision of a PDA particle (or other types of particles) to another PDA particle positioned within the mechano-FL sensor channel. By manipulating variables such as size, rigidity, and collision speed, we aim to compare the area where FL is emitted to the actual collision area.

19-The data availability statement is not sufficient as the raw data is all in the manuscript or supplemental. Information on how statistics and error should be provided.

We carried out the main experiments pertaining to PDA FL intensity, fluid viscosity, and confocal microscopy through a minimum of three independent trials. The data presented as mean values, and the corresponding standard deviations are indicated by error bars at each data point. We have clearly documented this information in each figure caption and in the Methods section of SI where applicable.

39

[Type here]

Reviewer #3 (Remarks to the Author):

In this paper, the authors demonstrated a droplet microfluidics-based fabrication technique to develop fluorescent PDA particles with unique mechanosensitive properties. After thoroughly characterizing the particles, the authors explore their potential as a probe to monitor mechanical stress in the microchannel and particulate flows. In the key experiments, the authors individually trap these particles at the center of a microchannel and induce/tune the injection flow parameters, such as flow rate, flow duration, viscosity, and particle concentration. Observable mechanofluorescent response occurs when a colloidal suspension is injected (10-20% particle concentration, somewhat similar to physiological RBC flows). Empirically and based on the SEM images, it is possible to hypothesize that colloids bombard the mechanosensitive particle, transferring energy and distorting the PDA particle's backbone, thereby causing mechanofluorescent sensors. However, this response is nonlinear: typically, fluorescent signal increases after 1 minute, plateaus after 2 min, and rises again after a few minutes of exposure (Fig.3e, Fig. S5d, Fig.5f). This response is critical for its stress-sensing application in a biological flow. Finally, the authors apply an ambitious CFD model to elucidate this response, simulations estimate the transferred hydraulic and collision energies to the PDA particle, and they particularly relate collision energy to the critical torsion energy needed to cause an optimal fluorescent response.

While the reviewer thinks that the overall concept and its application are interesting, the reviewer do not think it meets the standards and requirements of Nature Communications (see major comments).

We appreciate the time and effort the reviewer has taken to provide constructive feedback on our work. In line with the observations of the first reviewer, we wish to emphasize that our study presents a significant contribution to the challenging field of visualizing flow-induced mechanical stress in designed environments. We believe that our research offers a robust framework that has the potential to be adapted and extended to more complex and refined settings, thus offering closer approximations to real physiological conditions. Furthermore, we concur with the second reviewer's positive remarks concerning our use of the microfluidic approach to fabricate PDA particles. This methodology, as the second reviewer noted, "potentially opens the door to further quantification of mechanical stress induced fluorescence in PDAs through a wide variety of traditional single molecules and microfluidic approaches for example optical tweezers, micropipette aspiration, etc."

Major comments:

1. The reviewer is unable to see an obvious application for this sensor probe. For instance, the authors mentioned that "this study represents a significant advancement in the development of minimally invasive force measurement systems that can be located in hard-to-reach spaces within the human body, including implantable vascular prostheses, blood cell trauma caused

by stenosis, and flow-induced thrombogenicity. How do the authors plan to translate this system to establish a mechanical stress measurement system within the human body? In the experiment, sensor particles have a certain size (200 μm diameter), and they are manually

40
[Type here]

placed and trapped at the center of the microchannel flow to induce colloidal bombardment, which causes an irreversible fluorescent response measured by a microscope setup. In large vessels, these particles will probably flow along with the RBCs, in small vessels these particles will themselves induce blockage and cause high shear stresses? Furthermore, how do the authors measure the fluorescence coming from deep within human body? What would happen to the PDA particles?

We thank the reviewer for the insightful comments and concerns about the applicability and translation of our mechano-FL sensor system for mechanical stress measurements within the human body. Indeed, our system was developed to offer a unique advantage for simulating specific problematic regions of the human body in an in vitro environment. It enables the visualization and quantification of mechanical stresses or forces exerted on the region of interest. We agree that directly implanting the mechano-FL sensor system into the human body presents challenges. To eliminate potential confusion and to clarify this point, we have appropriately revised the statement highlighted by the reviewer.

Additionally, while we utilized PDA particles of 200 μm in size in this study, a key advantage of our microfluidic fabrication method is its adaptability. By adjusting the flow rates introduced into the microfluidic device or by altering the geometry of the capillary tubes constituting the device, we can systematically control the particle size, ranging from tens to hundreds of micrometers. Moreover, through an emulsion method, we can fabricate PDA particles on the scale of a few micrometers. We have indeed manufactured PDA particles of various sizes using these methods (Fig. R3) and are pursuing further studies with them.

As previously noted, our current mechano-FL sensor system is ideally suited for in vitro approaches that mimic specific human body regions. However, if improvements are made for in vivo applications, monitoring the resulting signals could be performed using fiber optic-based fluorescence imaging techniques (Opt. Fiber Technol., 19, 760, 2013). In fact, fiber optic fluorescence imaging has been recognized as a useful tool in various biomedical applications, especially in diagnosing and monitoring vascular diseases, tumors, and other lesions.

Before: We anticipate that this study will provide an important foundation for designing minimally invasive mechanical stress measurement systems that can be placed in small and difficult-to-access areas deep within the human body.

After (Abstract): We believe this study offers a unique advantage for simulating specific problematic regions of the human body in an in vitro environment, potentially paving the way for future exploration of difficult-to-access areas within the body.

41
[Type here]

[Image redacted]

Fig. R3 (Reviewer Only). PDA particles with various sizes fabricated using the co-flow microfluidic approach (a-d) and the emulsion method (e). The scale bars are 100 μm .

Before (Discussion): This study represents a significant advancement in the development of minimally invasive force/stress measurement systems that can be located in hard-to-reach small spaces within the human body, including implantable vascular prostheses, blood cell trauma caused by stenosis, and flow-induced thrombogenicity. These systems can be utilized to construct a 3D-printed vascular stenosis model to examine locally applied mechanical stresses under conditions that more closely simulate actual blood flow in stenosed vessels. Furthermore,

it is speculated that PDAs generated from DA monomers functionalized with hydrogen-bonding moieties can be used to develop unique reversible sensor systems that respond to diverse stimuli.

After (p18, Discussion): We believe this study represents a significant advancement in the development of minimally invasive force/stress measurement in vitro systems that simulate hard-to-reach spaces within the human body. However, we recognize that several aspects in our current system need refinement. For instance, positioning the PDA probe particle within a narrowed channel could influence fluid behaviors. As illustrated in Supplementary Fig. 14, one potential solution is to embed the PDA particles into simulated vessel walls using materials such as hydrogels or polydimethylsiloxane (PDMS). To more accurately simulate actual blood flows in stenosed vessels, a 3D-printed vascular stenosis model can be employed.¹⁰ The sensitivity of the PDA particles as sensors could be enhanced by exploring different diacetylene monomers.^{41, 62, 63} A more straightforward approach might involve adjusting the UV exposure time for polymerization, thus controlling the degree of PDA particle polymerization. Given the diverse application requirements, our microfluidic setup allows us the flexibility to produce PDA particles of different sizes, ranging from tens to hundreds of micrometers. This can be achieved by adjusting the flow rates fed into the microfluidic device or by altering the geometry of the capillary tubes constituting the device.⁶⁴

2. If the authors do not plan to establish the system within a human body and only plan to apply the probe in 3D-printed, vessel-mimicking microfluidic channels, does the sphere affect the flow profile and hence affect the measurement? The probe is shaped like a sphere requiring direct contact with the fluidic flow. It is difficult to understand how the sphere (200 Åμm diameter) could be placed in a fluidic channel without altering the flow profile of the fluidic channels.

42
[Type here]

In this proof-of-concept study, we primarily aimed to demonstrate the potential of a mechano-FL sensor system based on single PDA particles. This system turned out effectively analyzing and visualizing flow-induced mechanical stresses in stenotic vessel-mimicking microfluidic channels. Based on the success of this research, we are planning subsequent studies that will design and construct models that more closely resemble real conditions. As this reviewer rightly pointed out, the positioning of the PDA particle in the center of the channel could alter flow dynamics. Alternatively, we are designing the geometry to anchor PDA particles to the channel walls using materials like hydrogels or polydimethylsiloxane (PDMS), as illustrated in Supplementary Fig. 14. We have incorporated these considerations into the Discussion section of the revised manuscript. For a detailed overview of these changes, we direct the reviewer to our response to the previous comment.

Newly added Supplementary Fig. 14:

[Image redacted]

Supplementary Fig. 14 Biomimetic stenosis-based PDA sensor. a Schematic representation of a stenotic blood vessel. b Proposed model for future studies, where PDA particles are partially embedded into a hydrogel structure that simulates the stenotic region.

43
[Type here]

3. The key sensor response seems highly specific to the induced flow and does not appear to have a linear and high dynamic range.

We appreciate the reviewer's attention to the specificity of the sensor response in relation to the induced flow, as well as concerns about its dynamic range. Indeed, as illustrated in Fig. 3h, the FL response of our PDA particle under mechanical stress aligns well to the characteristic behavior of typical PDA materials, notably the nonlinear FL response exhibited in thermochromism, as displayed in Fig. 2g. This nonlinearity appears to be closely linked with the torsion energy variation pattern as the PDA main chain backbone is distorted, as supported by the inset in Fig. 5f.

To further quantify the operating range of the FL response of our PDA particles, we performed a 16.95 wt.% SNP injection experiment over an extended injection time of 8.5 min (510 s). We found that the normalized FL intensity of the PDA particle approached 1 at ~420 s, and further injection barely altered the intensity value. We have updated these results in Fig. 5f and Supplementary Fig. 9. In this case for the complete transition, we estimated the PDA operating range in terms of the total energy exerted on the PDA particle to be ~307 μ J. This value is consistent with previous studies predicted for various types of PDA materials [Langmuir, 16, 4639, 2000].

Additionally, we note that the sensitivity of the PDA FL response to mechanical stress can be modulated by altering the polymerization degree of the PDA particle or employing different types of diacetylene monomers [J. Am. Chem. Soc. 127, 12782, 2005; Adv. Mater. 30, 1801121, 2018]. Furthermore, our microfluidic setup allows us the flexibility to produce PDA particles of different sizes, ranging from tens to hundreds of micrometers (Fig. R3a-d). This can be achieved by adjusting the flow rates fed into the microfluidic device or by altering the geometry of the capillary tubes constituting the device [J. Colloid Interface Sci. 560, 838, 2020]. This adaptability introduces a level of versatility, making the system amenable to fine-tuning for specific applications. These points have been incorporated into the revised manuscript for greater clarity and context.

Added statement (p13): Additionally, we extended the injection time of 16.95 wt.% SNPs to 8.5 min (510 s) and found that the normalized FL intensity approached 1, indicating near saturation (Supplementary Fig. 9).

Newly added Supplementary Fig. 9:

[Image redacted]

44

[Type here]

Supplementary Fig. 9 Mechano-FL system response over an extended injection time. A 16.95wt.% SNP solution was injected for 8.5 min (510 s), with the injection and suction rates maintained at 25 mL \cdot min⁻¹ and 5 mL \cdot min⁻¹, respectively. This extended injection time yielded a normalized FL intensity approaching 1, indicating near saturation. The error bars represent the standard deviations derived from at least three independent trials. Note that this plot is the same with the profile of FL intensity versus time presented in Fig. 5f of the main text.

Before: The calculations show that bombardment by the SNPs leads to an increase in δ_{ij} absorbed by the PDA particle, considered to be the sum of the computed δ_{ij} and SNP collision energy δ_{ij} calculated using the Hertz contact equation (Table S6).50 As shown in Fig. 5e, δ_{ij} increases in a linear manner upon increasing the concentration of the SNP in the 0, 5.65, 11.3 and 16.95 wt.% range injected for 4.5 min. Similarly, because the conjugated structure of the PDA was not considered in this simulation, the δ_{ij} values do not accurately reflect the observed nonlinear SNP concentration dependent increase in FL intensities. However, the sudden increase in the FL intensity observed in 11.3 to 16.95 wt.% region suggests that a certain critical torsion energy threshold exists between 33.75 and 50.73 μ J.

To determine the critical torsion energy level, δ_{ij} values were calculated as a function of the SNP injection time and then compared to the torsion energy δ_{ij} (Tables S7 and S8).46, 47 The calculated variation in torsion energy as a function of the δ_{ij} and the SNP injection time (t) show that two rapid increases in torsion energy take place in the time intervals of 60 s $\hat{=}$ δ_{ij} $\hat{=}$ 120 s and 210 s $\hat{=}$ δ_{ij} $\hat{=}$ 270 s, as indicated by the orange bars in Fig. 5f. The corresponding torsion angles are 11 $\hat{=}$ δ_{ij} $\hat{=}$ 22 $\hat{=}$ and 39 $\hat{=}$ δ_{ij} $\hat{=}$ 50 $\hat{=}$, which are

consistent with PDA polymorphs displaying most prominent FL emission.⁴⁶ The calculated nonlinear δ_{tot} profile is in good agreement with the experimentally observed FL intensity profile, demonstrating the role played by the unique structural characteristics of the PDA particle in determining FL emission. In addition, because the δ_{tot} value obtained from the CFD simulation is the same order of magnitude as δ_{col} , the range of $\sim 40\text{--}50 \text{ J}$ is estimated to be the critical energy level, around which the PDA particle undergoes a notable FL intensity increase.

45

[Type here]

After (p17): The calculations show that bombardment by SNPs leads to an increase in δ_{tot} absorbed by the PDA particle, considered to be the sum of the computed δ_{H} and SNP collision energy δ_{col} calculated using the Hertz contact equation (Supplementary Table 7).⁶⁰ δ_{tot} demonstrates an increasing trend as the SNP concentration increases, spanning 0, 5.65, 11.3, and 16.95 wt.% when injected for 4.5 min (Fig. 5e) and with an extended injection duration of 8.5 min at 16.95 wt.% concentration (Fig. 5f). Similarly, because the conjugated structure of the PDA was not considered in this simulation, the δ_{tot} values do not accurately reflect the observed nonlinear behavior of FL intensities dependent on the SNP concentration.

Nevertheless, this computational data suggests that the δ_{tot} required to achieve a complete blue-to-red transition for a single PDA particle is approximately 307 J . The activation energy δ_{act} for irreversible blue-to-red transitions in a variety of PDAs has been estimated to be $\sim 17.6\text{--}22.5 \text{ kcal}\cdot\text{mol}^{-1}$.⁴⁴ For instance, for a three-layer PDA film fabricated from the PCDA monomers, the predicted δ_{act} was approximately $17.6 \text{ kcal}\cdot\text{mol}^{-1}$. Though our 200 nm PDA particles might differ from the previously reported PDA conditions, we presumed $\delta_{\text{act}} \sim 17.6\text{--}22.5 \text{ kcal}\cdot\text{mol}^{-1}$ to roughly estimate δ_{act} for our PDA particles. Considering $\sim 27\%$

$\delta_{\text{act}}^{\text{PDA}}$

polymerization (Fig. 2d) yields $0.27 \text{ PDA } \delta_{\text{act}} = 289\text{--}369 \text{ J}$, where the PDA density

δ_{PCDA}

$3 \times 10^3 \text{ kg}\cdot\text{m}^{-3}$

is $\delta_{\text{PDA}} \sim 1.3 \times 10^3 \text{ kg}\cdot\text{m}^{-3}$, the PDA volume is $\delta_{\text{PDA}} = 3$, and the molecular weight of PCDA is $\delta_{\text{PCDA}} = 0.375 \text{ kg}\cdot\text{mol}^{-1}$. This result aligns well with the energy requirement of $\sim 307 \text{ J}$ for the complete blue-to-red transition, as derived from our calculations.

The nonlinear FL intensity profile possesses a resemblance to the previously reported torsional energy δ_{tor} calculations, which also exhibit nonlinear increases for various types of PDA materials.^{53, 54} Therefore, we compared the normalized δ_{tor} profile with the normalized FL intensity profiles derived from both the mechano-FL sensor experiment (Fig. 5f) and the thermal stress analysis (Fig. 2g). As displayed in the inset of Fig. 5f, the experimental FL intensity's nonlinear trajectory seems fairly consistent with the shape of δ_{tor} . This similarity allows us to reasonably postulate a close correlation between these two factors. Notably, recent experimental results have demonstrated that lateral stress applied parallel to the PDA backbone significantly influences the mechanochromism of thin PDA films.⁶¹ Therefore, further experimental and/or theoretical studies are needed to clarify how torsional and lateral stresses contribute relative to each other depending on the PDA structures under various conditions.

Revised Fig. 5f:

[Image redacted]

46

[Type here]

Fig. 5 CFD analysis of the mechano-FL sensor system. f Time-dependent profiles of δ_{tot} and FL intensity for the 16.95 wt.% SNP suspension injected at $25 \text{ mL}\cdot\text{min}^{-1}$ over an extended duration of 8.5 min. Inset indicates the comparison of the nonlinear property of normalized FL intensity profiles from the mechano-FL sensor experiment and the thermal stress analysis (from Fig. 2g) against the normalized torsion energy δ_{tor} from a previous study.⁵⁴ The x-axis is normalized from 0 to 1, aligning maximum values of FL intensity and δ_{tor} to 1. For panels d-f, each experimental data point and corresponding error bar represent the mean intensity value of a minimum of three independent trials and its standard deviation, respectively.

Newly added statements (p20, Discussion): The sensitivity of the PDA particles as sensors could be enhanced by exploring different diacetylene monomers.^{41, 62, 63} A more straightforward approach might involve adjusting the UV exposure time for polymerization, thus controlling the degree of PDA particle polymerization. Given the diverse application requirements, our microfluidic setup allows us the flexibility to produce PDA particles of different sizes, ranging from tens to hundreds of micrometers. This can be achieved by adjusting the flow rates fed into the microfluidic device or by altering the geometry of the capillary tubes constituting the device.⁶⁴

Minor comments:

4â€¢ What are the standard deviations in the plots? What is the number of particles (n) tested in each experiment?

In both the main text and SI, each FL intensity data point represents the average of a minimum

47

[Type here]

of three independent trials, and the error bars indicate the corresponding standard deviations. We have clarified this point in the revised manuscript.

Before (SI): For all FL measurements, the camera sensitivity and exposure time were set to ISO 100 and 0.4 s, respectively.

After (p4 in SI): Additionally, the PDA particles were only exposed to the FL laser during the FL measurements, and the DSLR camera settings were kept consistent (i.e., ISO 100 for camera sensitivity and 0.4 s exposure time) during the capturing of FL images. Note that each PDA particle was utilized only once for the mechano-FL sensor experiments; no reused particles were incorporated into any subsequent FL sensor experiments. All the FL measurements were conducted independently at least three times. The mean intensity values, along with their respective standard deviations, were reported in the main text and Supplementary Information. By maintaining these stringent conditions, we ensured the minimization of any potential errors from FL measurements.

Added statement (Caption of Fig. 3): All FL experiments were performed independently at least three times. The data points and error bars in panels a-d and h indicate the mean intensity values and standard deviations, respectively.

Added statement (Caption of Fig. 5): For panels d-f, each experimental data point and corresponding error bar represent the mean intensity value of a minimum of three independent trials and its standard deviation, respectively.

Added statement (Caption of newly added Supplementary Fig. 8): Each FL intensity data point represents the average of a minimum of three independent trials, and the error bars indicate the corresponding standard deviations.

Added statement (Caption of newly added Supplementary Fig. 9 and Supplementary Fig. 10): The error bars indicate the standard deviations derived from at least three independent trials.

5â€¢ Would it be possible to tune the mechanofluorescent response? (e.g., more or less sensitive to stress and particle injection)

The sensitivity control of PDA particles is potentially feasible, and for a detailed response to this, we refer the reviewer to our answer to this reviewer's third comment.

6â€¢ Is there any other response after the FL intensity of 0.3 a.u? Is this about the maximum?

Does the particle start to lose its mechanical integrity in the flow after that (e.g. if one performs these experiment for >10 minutes)?

48

[Type here]

We thank the reviewer for point this out. In accordance with the reviewer's inquiry, we conducted an extended time injection experiment (8.5 min) for 16.95 wt.% SNPs, and as previously mentioned, we confirmed that the normalized FL intensity reached 1 a.u. after ~420 s. We refer to our detailed response to this reviewer's third comment for further information.

Reviewer #4 (Remarks to the Author):

In their manuscript, Ahmadi et al. describe the development of monodisperse polydiacetylene (PDA) microparticles and their fluorescence responses to a flow of fluid that contains silica nanoparticles or red blood cells. The concept and the results obtained are highly novel and interesting, but there are several points that must be corrected before the manuscript can be published.

We deeply appreciate the reviewer's emphasis on the novelty of the development of the mechano-FL sensor system using monodisperse PDA particles. We have carefully reviewed all the reviewer's comments and have made the following revisions.

1- About the preparation and characterization of PDA particles:

1.1) As far as I understand the particles fabrication protocol, the injection rates of aqueous PVA solution and PCDA monomer solution in chloroform are important. Could the authors explain what happen if these flow rates are changed? Is it possible to modify the size of the particles by modifying these flow rates or does this have an impact on the size dispersion?

By adjusting the flow rates introduced into the microfluidic device or by altering the geometry of the capillary tubes constituting the device, we can systematically control the particle size, ranging from tens to hundreds of micrometers (Fig. R3a-d). In the current study, after constructing the microfluidic device according to the protocol outlined in the Supplementary Method, we carefully adjusted the injection flow rate of each fluid until we achieved stable generation of PCDA-chloroform droplets in the microfluidic channel. Having identified the optimal conditions, we deliberately fabricated dried PDA particles of approximately 200 μm in diameter. This was achieved by using a flow rate of 400 $\mu\text{L}\cdot\text{min}^{-1}$ for the 2 wt.% PVA aqueous solution and a flow rate of 60 $\mu\text{L}\cdot\text{min}^{-1}$ for the PCDA-chloroform solution. We have revised the manuscript to clarify these points.

49

[Type here]

[Image redacted]

Fig. R3 (Reviewer Only). PDA particles with various sizes fabricated using the co-flow microfluidic approach (a-d) and the emulsion method (e). The scale bars are 100 μm .

Before: The prepared solutions were loaded into two gas-tight syringes, and separately delivered into the microfluidic device using two syringe pumps. Sol.A was injected through a needle installed in the gap between CC-1 and SC-1 at a rate of 400 $\mu\text{L}/\text{min}$. Sol.B was added to CC-2 at a rate of 60 $\mu\text{L}/\text{min}$. Injection of the fluids resulted in periodic generation of PCDA-chloroform droplets that traveled along CC-1. The generated PCDA droplets were collected in a water-filled glass petri dish for 5 min, and subsequently stored under dark conditions for at

least for ~12 h, leading to formation of solid PCDA particles.

After (p22 in Methods): The prepared solutions were loaded into two gas-tight syringes, and separately delivered into the microfluidic device using two syringe pumps. Sol.A was injected through a needle positioned in the gap between CC-1 and SC-1, and Sol.B was added to CC-2. The injection rate of each fluid phase was carefully adjusted to ensure the stable generation of PCDA-chloroform droplets with a desired size and size distribution. Once the optimal conditions were identified, the flow rates used in this study were established as 400 $\mu\text{L}\cdot\text{min}^{-1}$ for Sol.A and 60 $\mu\text{L}\cdot\text{min}^{-1}$ for Sol.B. The produced PCDA-chloroform droplets were then directed along CC-1 and collected in a water-filled glass petri dish for 5 min. To facilitate the evaporation of chloroform from the individual droplets, they were stored under dark conditions for ~12 h, leading to the formation of solid PCDA particles.

Added statements (p20 in Discussion): Given the diverse application requirements, our microfluidic setup allows us the flexibility to produce PDA particles of different sizes, ranging from tens to hundreds of micrometers. This can be achieved by adjusting the flow rates fed into the microfluidic device or by altering the geometry of the capillary tubes constituting the device.⁶⁴

1.2) What is the role of PVA in this protocol and are the obtained particles coated with PVA at the end of the process?

PVA is typically utilized as a stabilizer in microfluidic fabrication methods and can be easily removed from final solidified particles simply by washing them with water. In our study, we

50
[Type here]

also used a 2 wt.% PVA aqueous solution as a continuous phase to prevent the coalescence of the generated PCDA-chloroform droplets within the microfluidic channels.

Before: Sol.A was an aqueous PVA (2 wt.%) solution and Sol.B was the PCDA monomer solution (10 wt.%) containing 3.7 mg PCDA dissolved in 1 mL chloroform.

After (p22, Methods): Sol.A was an aqueous solution containing 2 wt.% PVA, used as a stabilizer, and Sol.B was a 10 wt.% PCDA monomer solution, which consisted of 3.7 mg PCDA dissolved in 1 mL of chloroform.

1.3) Could the authors give some indications about the stability of the PDA particles (colloidal stability and photostability during the fluorescence experiments)?

The PDA particles with ~200 μm in size quickly settle to the bottom when dispersed in an aqueous solution. Consequently, any potential aggregation that might occur among typical colloidal particles of a few or several micrometers can be disregarded. Even in the event where two PDA particles adhere to each other, separation can easily be achieved via the application of hydrodynamic force using a micropipette.

As for photostability, PDA materials are broadly recognized for their remarkable photostability, showing resistance to photobleaching. While PDAs could theoretically undergo photobleaching under specific extreme conditions, such as intense, prolonged light exposure or the presence of certain chemical reactants, our experimental conditions unlikely fit these extreme scenarios.

However, to further minimize any potential errors in our experiments with PDA particles, we consistently store the fabricated PDA particles in darkness at 4 $^{\circ}\text{C}$ to prevent any detrimental effects resulting from heat and light exposure under typical ambient conditions. Furthermore, all fluorescence (FL) measurements and observations were performed in a dark environment to preclude interference from external light sources, ensuring the accuracy of the FL signals. Additionally, the PDA particles were only exposed to a FL laser during the FL measurements, and the DSLR camera settings were kept consistent (i.e., ISO 100 for camera sensitivity and 0.4 s exposure time) when capturing FL images. We also evaluated the superior photostability

of the blue phase PDA particles, which remained continuously exposed to a FL laser for 5 min in ambient conditions without the use of a dark environment. FL images were captured every 30 s. As shown in Fig. R4, the FL intensity change of the blue phase PDA particles remained almost zero. Moreover, we ensure that no PDA particles are reused.

All these factors affirm that our experimental setups did not cause any significant errors in the FL measurements of PDA particles during the mechano-FL sensor experiments. We have revised the manuscript to clarify these points.

Before: Finally, the obtained PDA particles were washed with water, and then stored in fresh

51
[Type here]

water at 4 °C for further mechano-FL experiments.

After (p22, Methods): Finally, the obtained PDA particles were washed with water, and then stored in fresh water. The particles were kept under dark conditions at 4 °C until they were needed for subsequent mechano-FL experiments.

Before (Supplementary Information): The FL response images were captured using a digital single-lens reflex camera (EOS 700, Canon, Japan) mounted on an optical microscope (Ti-U, Nikon, Japan) equipped with a metal halide illuminator (PhotoFluor LM-75, 89 North, USA). For all FL measurements, the camera sensitivity and exposure time were set to ISO 100 and 0.4 s, respectively.

After (p4 in SI): In order to capture fluorescence (FL) response images, a digital single-lens reflex (DSLR) camera (EOS 700, Canon, Japan) mounted on an optical microscope (Ti-U, Nikon, Japan) was employed. The microscope was equipped with a metal halide illuminator (PhotoFluor LM-75, 89 North, USA) offering a full output spectrum from 340 nm to 800 nm. To selectively isolate red FL emission region, a Nikon FL filter (C-FL Epi-fluorescence TRITC filter, Nikon, Japan) was used. All FL measurements and observations were performed in a dark environment to eliminate potential interference from external light sources, ensuring the accuracy of the FL signals. Additionally, the PDA particles were only exposed to the FL laser during the FL measurements, and the DSLR camera settings were kept consistent (i.e., ISO 100 for camera sensitivity and 0.4 s exposure time) during the capturing of FL images. Note that each PDA particle was utilized only once for the mechano-FL sensor experiments; no reused particles were incorporated into any subsequent FL sensor experiments. All the FL measurements were conducted independently at least three times. The mean intensity values, along with their respective standard deviations, were reported in the main text and Supplementary Information. By maintaining these stringent conditions, we ensured the minimization of any potential errors from FL measurements.

52
[Type here]

[Image redacted]

Fig. R4 (Reviewer Only) Photostability evaluation of blue phase PDA particles. Normalized mean FL intensities of the PDA particles over time, monitored while continuously exposed to the FL laser for 5 min under ambient conditions. These PDA particles had been stored in the dark at 4 °C for 3 months prior to this experiment. The inset shows a magnified portion of the plot to better visualize the error bars, which indicate the standard deviation from five independent measurements performed on different particles. Throughout the experiment, the FL intensity of the blue phase PDA particles remained almost zero, demonstrating their excellent photostability.

1.4) The photopolymerization procedure should be described more precisely: what is the power per surface unit received by the particle suspension? Could the authors provide some

information about the UV-Vis absorption spectrum of the polymerized particles? This is important because it is known that for polydiacetylene the shape of the UV-Vis spectrum can be modified when the photopolymerization progresses with sometimes the apparition of red PDA phase that can change not only the absorption properties but also the fluorescence emission properties (see for example Y. Lifshitz, A. Upcher, O. Shusterman, B. Horovitz, A. Berman and Y. Golan, Phys. Chem. Chem. Phys., 2010, 12, 713â722 for a description of the evolution of absorbance of PDA films with the photopolymerization rate).

In our study, the dried PCDA particles were exposed to 254 nm UV irradiation using a VL-6LC UV lamp (Vilber Lourmat, France) at a power density of 0.4 mWÂ·cmâ2 for 10 min.

Additionally, we conducted solid-state UV-Vis spectrophotometry measurements on both blue and red phase PDAs. Initially, we attempted to prepare PDA particle-coated samples using the convective assembly method, where aqueous droplets containing PDA particles were placed onto a glass substrate and allowed to dry under ambient conditions. The resulting PDA particle-coated film intrinsically exhibited a non-uniform surface due to its spherical geometry with a ~200 Î¼m diameter. Consequently, the results of UV-Vis absorbance lacked consistency. It should be noted that such non-uniformity in film thickness can influence the absorption and reflection of light, complicating the resulting spectra and making accurate interpretation more challenging. Instead, we decided to prepare uniform PDA films on a glass substrate using the same protocol and environment as when fabricating the PDA particles via the microfluidic method. This was done to confirm whether our microfluidic-produced particles exhibited typical UV-Vis behaviors consistent with those of typical PDA materials. Specifically, we drop-casted a 10 wt% PCDA solution in chloroform onto a glass substrate and covered the drop with a 2 wt% PVA water solution, thereby mimicking the conditions used in the microfluidic fabrication. Once the solvent had completely evaporated, the PCDA film underwent polymerization under a UV lamp (254 nm VL-6LC, Vilber Lourmat, France) for 1, 5, and 10 min, respectively. The resulting blue phase film was then washed with DI water to remove any excess PVA. To prepare the red phase PDA film, the blue phase film was immersed in two different pH conditions, ~8.7 and ~9.7, for 30 min. Upon conducting UV-Vis measurements of these samples, we found that they exhibited typical behaviors of PDA materials. The maximum absorbance occurred at ~640 nm for the blue phase samples and ~550 nm for the red phase samples, as displaced in Fig. R1.

53

[Type here]

Before: Photopolymerization of the PCDA particles was conducted using 254 nm UV irradiation (1 W, VL-6LC, Vilber Lourmat, France) for ~10 min.

After (p22 in Methods): Photopolymerization of the PCDA particles, dispersed in the water-filled petri dish, was conducted under 254 nm UV irradiation (1 W, VL-6LC, Vilber Lourmat, France) at a power density of 0.4 mWÂ·cmâ2 for 10 min.

[Image redacted]

Fig. R1 (Reviewer Only) Solid-state UV-Vis spectrophotometry of PDA films subjected to various conditions. Details on the preparation of these PDA films are provided in the preceding text.

1.5) As far as I understand, the authors seem to suppose that the diacetylene monomers in the particles are completely polymerized to give polydiacetylene chains. In particular, this hypothesis is used in the calculation of the PDA torsion energy. However, it has been demonstrated on several PDAs that upon UV irradiation the polymerization is far from complete. This has been shown in particular on PCDA in this reference: Spagnoli et al., Langmuir 2017, 33, 1419â1426. As a consequence, the authors should estimate the polymerization rate in their microparticles, and correct the PDA torsion energy calculation accordingly.

We appreciate the reviewer's comment concerning the degree of polymerization within the PDA particles. We acknowledge that our initial assumption, that the entire PDA particle is polymerized, was oversimplified. Furthermore, we overlooked the recent report that the finite

penetration depth of UV irradiation can lead to partial polymerization, as pointed out by the reviewer. To address this concern, we employed confocal microscope measurements on red phase PDA particles to ascertain the thickness of the polymerized shell (denoted as h_{shell}), which corresponds to the UV penetration depth within the particles. Our findings indicate the h_{shell} value of $10.0 \pm 0.03 \mu\text{m}$ over nine different particles (as indicated in the revised Fig. 2d).

54
[Type here]

The corresponding volume degree of polymerization was $\sim 27\%$, consistent with prior research (Spagnoli et al., Langmuir, 33, 1419, 2017).

Additionally, we conducted a SEM analysis to further substantiate the partial polymerization of the PDA particles. The particles were dispersed within polydimethylsiloxane (PDMS), followed by curing the sample for two days under ambient conditions after adding a 10 wt.% curing agent. We then bisected the PDMS-embedded PDA particle with a sharp razor. The bisected PDA particle was submerged in chloroform for an hour, dried, and subsequently imaged via SEM. As shown in the revised Fig. 2e,f, the interior of the bisected particle appeared to be filled with a solid substance before the chloroform treatment. However, post-treatment, the interior appeared hollow. Given that PCDA monomers are soluble in chloroform, these results confirm the presence of PCDA monomers inside the polymerized shell.

These findings have been incorporated into the revised manuscript, and the calculations for torsion energy have been modified to reflect the degree of polymerization.

Before: Inspection of a SEM image shows that the surface and interior of the PDA particles have lamellar structures (Fig. 2c, d), which is the typical plate-like morphology of PDAs formed by photopolymerization of self-assembled PCDA monomers.

After (p6): Upon inspection of SEM images, it is apparent that the surface of the PDA particles features fuzzy domains, as depicted in Fig. 2c. This characteristic morphology has a resemblance to the structure of giant lipid/PDA vesicles, prepared using a simple solvent evaporation method.³⁸

Importantly, due to the finite penetration depth of the UV irradiation, the PDA particles underwent partial polymerization.³⁹ As evident from the confocal images of the red phase PDA particles in Fig. 2d, the polymerized shell is clearly visible with an approximate thickness of $h_{\text{shell}} \approx 10.0 \pm 0.03 \mu\text{m}$, as averaged over nine different particles. This corresponds to a maximum volume degree of polymerization of $\sim 27\%$. Furthermore, SEM analyses of bisected PDA particles reaffirmed the partial polymerization of the PDA particles. The interior of the bisected particle appeared to be filled with a solid substance before the chloroform treatment (Fig. 2e) but appeared hollow after the treatment (Fig. 2f). Given that PCDA monomers are soluble in chloroform, these results demonstrate that the center region of the particle remained unpolymerized and contained residual PCDA monomers.

Before: The PDA particle morphology was examined using an optical microscope (IX83, Olympus, Japan) and a field-emission scanning electron microscopy (FE-SEM, LEO SUPRA 55, Carl Zeiss, Germany).

After (p3 in SI): The morphology of the PDA particles was analyzed using an optical microscope (IX83, Olympus, Japan) and a field-emission scanning electron microscopy (FE-SEM, LEO SUPRA 55, Carl Zeiss, Germany). For visualizing the interior of PDA particles, they were dispersed within polydimethylsiloxane (PDMS), followed by curing the sample for two days under ambient conditions after adding a 10 wt.% curing agent. Subsequently, the

55
[Type here]

PDMS-embedded PDA particle was carefully bisected with a sharp razor. The bisected PDA particle was submerged in chloroform for an hour, dried, and finally imaged using SEM.

Revised Fig. 2:

[Image redacted]

Fig. 2 Structural and optical properties of PDA particles. a,b Optical microscopic images of PCDA-chloroform emulsions in water (a) and solid PDA particles in water (b). c SEM image of PDA particles. d Confocal microscope image of a red phase PDA particle heated at 80 °C for 10 min, revealing a polymerized shell of a thickness h_{shell} . Two additional images from different particles are shown below. e,f SEM images of a bisected PDA particle before (e) and after (f) chloroform treatment. g Thermo-FL response of PDA particles subjected to gradual heating from 40 °C to 100 °C. h Raman spectra of PCDA monomers (i), a blue PDA particle (ii), a red PDA particle produced by heating at 80 °C for 10 min (iii), and a mechanically crushed PDA particle (iv).

Before: The calculations show that bombardment by the SNPs leads to an increase in δ_{tot} absorbed by the PDA particle, considered to be the sum of the computed δ_H and SNP collision energy δ_{col} calculated using the Hertz contact equation (Table S6). As shown in Fig. 5e, δ_{tot} increases in a linear manner upon increasing the concentration of the SNP in the 0, 5.65, 11.3 and 16.95 wt.% range injected for 4.5 min. Similarly, because the conjugated structure of

56
[Type here]

the PDA was not considered in this simulation, the δ_{tot} values do not accurately reflect the observed nonlinear SNP concentration dependent increase in FL intensities. However, the sudden increase in the FL intensity observed in 11.3 to 16.95 wt.% region suggests that a certain critical torsion energy threshold exists between 33.75 and 50.73 μJ .

To determine the critical torsion energy level, δ_{tot} values were calculated as a function of the SNP injection time and then compared to the torsion energy δ_{tor} (Tables S7 and S8).^{46, 47} The calculated variation in torsion energy as a function of the δ_{tor} and the SNP injection time (t) show that two rapid increases in torsion energy take place in the time intervals of 60 s $\hat{=}$ 120 s and 210 s $\hat{=}$ 270 s, as indicated by the orange bars in Fig. 5f. The corresponding torsion angles are 11° $\hat{=}$ 22° and 39° $\hat{=}$ 50°, which are consistent with PDA polymorphs displaying most prominent FL emission.⁴⁶ The calculated nonlinear δ_{tot} profile is in good agreement with the experimentally observed FL intensity profile, demonstrating the role played by the unique structural characteristics of the PDA particle in determining FL emission. In addition, because the δ_{tot} value obtained from the CFD simulation is the same order of magnitude as δ_{tor} , the range of $\sim 40\text{--}50 \mu\text{J}$ is estimated to be the critical energy level, around which the PDA particle undergoes a notable FL intensity increase.

After (p18): The calculations show that bombardment by SNPs leads to an increase in δ_{tot} absorbed by the PDA particle, considered to be the sum of the computed δ_H and SNP collision energy δ_{col} calculated using the Hertz contact equation (Supplementary Table 7).⁶⁰ δ_{tot} demonstrates an increasing trend as the SNP concentration increases, spanning 0, 5.65, 11.3, and 16.95 wt.% when injected for 4.5 min (Fig. 5e) and with an extended injection duration of 8.5 min at 16.95 wt.% concentration (Fig. 5f). Similarly, because the conjugated structure of the PDA was not considered in this simulation, the δ_{tot} values do not accurately reflect the observed nonlinear behavior of FL intensities dependent on the SNP concentration.

Nevertheless, this computational data suggests that the δ_{tot} required to achieve a complete blue-to-red transition for a single PDA particle is approximately 307 μJ . The activation energy δ_{act} for irreversible blue-to-red transitions in a variety of PDAs has been estimated to be $\sim 17.6\text{--}22.5 \text{ kcal}\cdot\text{mol}^{-1}$.⁴⁴ For instance, for a three-layer PDA film fabricated from the PCDA monomers, the predicted δ_{act} was approximately 17.6 $\text{kcal}\cdot\text{mol}^{-1}$. Though our 200 μm PDA particles might differ from the previously reported PDA conditions, we presumed $\delta_{act} \hat{=}$ 17.6 $\text{--} 22.5 \text{ kcal}\cdot\text{mol}^{-1}$ to roughly estimate δ_{act} for our PDA particles. Considering $\sim 27\%$

δ_{act} PDA
polymerization (Fig. 2d) yields 0.27 μm^3 PDA $\delta_{act} = 289\text{--}369 \mu\text{J}$, where the PDA density
 δ_{PCDA}

3 $\hat{=}$ 4 $\times 10^3 \text{ kg}\cdot\text{m}^{-3}$

is $\delta_{PCDA} \hat{=}$ 1.3 $\times 10^3 \text{ kg}\cdot\text{m}^{-3}$, the PDA volume is $\delta_{PDA} = 3 \mu\text{m}^3$, and the molecular weight of PCDA is $\delta_{PCDA} = 0.375 \text{ kg}\cdot\text{mol}^{-1}$. This result aligns well with the energy requirement of $\sim 307 \mu\text{J}$ for the complete blue-to-red transition, as derived from our calculations.

The nonlinear FL intensity profile possesses a resemblance to the previously reported torsional energy δ_{tor} calculations, which also exhibit nonlinear increases for various types of PDA materials.^{53, 54} Therefore, we compared the normalized δ_{tor} profile with the normalized

FL intensity profiles derived from both the mechano-FL sensor experiment (Fig. 5f) and the thermal stress analysis (Fig. 2g). As displayed in the inset of Fig. 5f, the experimental FL intensity's nonlinear trajectory seems fairly consistent with the shape of δ_{tor} . This similarity allows us to reasonably postulate a close correlation between these two factors. Notably, recent

57

[Type here]

experimental results have demonstrated that lateral stress applied parallel to the PDA backbone significantly influences the mechanochromism of thin PDA films.⁶¹ Therefore, further experimental and/or theoretical studies are needed to clarify how torsional and lateral stresses contribute relative to each other depending on the PDA structures under various conditions.

Revised Fig. 5:

[Image redacted]

Fig. 5 CFD analysis of the mechano-FL sensor system. a,b Examples showing distributions of flow velocity (a) and stress (b) for the 16.95 wt.% SNP suspension. c Variation of shear and normal stresses on the PDA surface as a function of the polar angle δP . Error bars represent the standard deviation over the stress values across the azimuthal angle $\delta \frac{1}{4}$. d,e Plots of δ_{H} and δ_{tot} as functions of glycerin (d) and SNPs (e) concentrations, respectively, at an injection rate of 25 mL·min⁻¹ over 4.5 min. Corresponding experimentally measured FL intensity profiles are overlaid for comparison. f Time-dependent profiles of δ_{tot} and FL intensity for the 16.95 wt.% SNP suspension injected at 25 mL·min⁻¹ over an extended duration of 8.5 min. Inset indicates the comparison of the nonlinear property of normalized FL intensity profiles from the mechano-FL sensor experiment and the thermal stress analysis (from Fig. 2g) against the normalized torsion energy δ_{tor} from a previous study.⁵⁴ The x-axis is normalized from 0 to 1, aligning maximum values of FL intensity and δ_{tor} to 1. For panels d-f, each experimental data point and corresponding error bar represent the mean intensity value of a minimum of three independent trials and its standard deviation, respectively.

1.6) Could the authors specify in ESI which excitation wavelength they use for fluorescence experiments and also for Raman characterization? This would be useful to ensure good reproducibility of the results.

For the fluorescence experiments, we used a PhotoFluor LM-75 metal halide illuminator (89 North, USA) that has a full output spectrum ranging from 340 nm to 800 nm. To selectively

58

[Type here]

isolate the red fluorescence emission regions, we employed a Nikon FL filter (C-FL Epi-fluorescence TRITC filter, Nikon, Japan). For Raman characterizations, we used a high-power near-infrared Ranishaw diode laser (>250 mW at 785 nm). Note that the use of 785 nm laser was beneficial to reduce background effects and maintain relatively high signal intensity. We clarified these points in the revised manuscript.

Before: The FL response images were captured using a digital single-lens reflex camera (EOS 700, Canon, Japan) mounted on an optical microscope (Ti-U, Nikon, Japan) equipped with a metal halide illuminator (PhotoFluor LM-75, 89 North, USA). For all FL measurements, the camera sensitivity and exposure time were set to ISO 100 and 0.4 s, respectively.

After (p4 in SI): In order to capture fluorescence (FL) response images, a digital single-lens reflex (DSLR) camera (EOS 700, Canon, Japan) mounted on an optical microscope (Ti-U, Nikon, Japan) was employed. The microscope was equipped with a metal halide illuminator (PhotoFluor LM-75, 89 North, USA) offering a full output spectrum from 340 nm to 800 nm. To selectively isolate red FL emission region, a Nikon FL filter (C-FL Epi-fluorescence TRITC

filter, Nikon, Japan) was used. All FL measurements and observations were performed in a dark environment to eliminate potential interference from external light sources, ensuring the accuracy of the FL signals. Additionally, the PDA particles were only exposed to the FL laser during the FL measurements, and the DSLR camera settings were kept consistent (i.e., ISO 100 for camera sensitivity and 0.4 s exposure time) during the capturing of FL images. Note that each PDA particle was utilized only once for the mechano-FL sensor experiments; no reused particles were incorporated into any subsequent FL sensor experiments. All the FL measurements were conducted independently at least three times. The mean intensity values, along with their respective standard deviations, were reported in the main text and Supplementary Information. By maintaining these stringent conditions, we ensured the minimization of any potential errors from FL measurements.

Before: The chemical analyses were performed with a Raman spectroscopy (inVia Basis, Renishaw, UK).

After (p3 in SI): Blue-to-red transition of the PDA particles was analyzed using a Raman spectroscopy (inVia Basis, Renishaw, UK) equipped with an adapted research-grade Lecia DM 2700 microscope and a high-power near-infrared Ranishaw diode laser (>250 mW at 785 nm). Note that the use of 785 nm laser was beneficial to reduce background effects and maintain relatively high signal intensity.

2- About the sensitivity of the PDA particles to stimuli:

2.1) I am intrigued by the results displayed on figure S2. Could the authors give an estimate of the force (or the stress) applied on the PDA particle by the three methods (shearing, poking, pressing)? In particular, in the following references (Chem. Commun. 2019, 55 (97), 59 [Type here]

14566â14569 and Nano Lett. 2021, 21, 543â549 which could both usefully be cited in the article) PDA are described to be sensitive to anisotropic friction stress, which seems to be in contradiction with the results displayed on figure S2 where visually the PDA particle seems to be more sensitive to pressing than to shearing. Could the authors discuss the apparent contradiction?

The experiments performed in Fig. S2 (in the original SI) were designed to assess whether PDA particles manufactured by the microfluidic method exhibit FL emissions in response to common mechanical stresses. However, it should be noted that the shear and normal stresses exerted in these experimental conditions were not controlled meticulously. Therefore, to make a comparison with the results of the studies mentioned by the reviewer, additional experiments would need to be conducted under more carefully designed and controlled conditions. Such research would go beyond the focus of our current study, which centers on the mechano-FL sensor system within microfluidic channels. Instead, we made efforts to carry out the three mechanical stress experiments in as consistent an environment as possible. Specifically, for shearing, a PDA particle was placed between a cover glass and a glass slide, and the cover glass was moved back and forth continuously for 30 s. For poking, a PDA particle placed on a glass slide was repeatedly poked using the sharp head of a capillary tube for 30 s. For pressing, a PDA particle was placed between a cover glass and a glass slide, and the cover glass was pressed using a capillary tube and then lifted slightly, which were repeated for 30 s. Note that as previously mentioned, while the newly obtained results were conducted under more refined experimental conditions, it is still not advisable to use these findings to quantitatively evaluate the effects of normal stress and shear stress on PDA FL response. The revised results have been added to Supplementary Fig. 3.

Before: The mechanochromism of the PDA particles was further explored by applying macroscopic mechanical stresses, including shearing, poking and pressing a single PDA particle, which all induce the blue-to-red transition and FL emission (Fig. S2).

After (p8): The typical mechanochromism of the PDA particles, manufactured through the microfluidic method, was further confirmed by subjecting them to common mechanical stresses, such as shearing, poking, and pressing. Each of these stresses induced FL emission,

as shown in Supplementary Fig. S3.

Revised Supplementary Fig. 3:

[Image redacted]

60

[Type here]

Supplementary Fig. 3 FL response of PDA particles to mechanical stresses. Schematics, optical images, and FL images (from left to right) of single PDA particles under different mechanical stresses: shearing (a), poking (b), and pressing (c). For shearing, a PDA particle was placed between a cover glass and a glass slide, and the cover glass was moved back and forth continuously for 30 s. For poking, a PDA particle placed on a glass slide was repeatedly poked using the sharp head of a capillary tube for 30 s. For pressing, a PDA particle was placed between a cover glass and a glass slide, and the cover glass was pressed using a capillary tube and then lifted slightly, which were repeated for 30 s. All three cases led to PDA FL emissions.

2.2) Could the authors please add scale bars on figure S3?

We thank the reviewer for pointing this out. We added the scale bars in the revised Supplementary Fig. 4 (Fig. S3 in the original manuscript).

Revised Supplementary Fig. 4:

61

[Type here]

[Image redacted]

2.3) Could the authors explain how they optimized the conditions for force sensing using the PDA particle? In particular, what happens at silica NPs concentration higher than 16.96%? And what happens at times longer than 4.5 minutes? How is the potential photobleaching of the particle taken into account during the measurements?

Regarding the potential issue of photobleaching, we refer this reviewer to our detailed response to the comment-1.3 of this reviewer.

Regarding the reviewer's inquiries about employing SNPs concentrations higher than 16.95 wt.% and extending the injection time beyond 4.5 min (270 s), we performed further experiments. Specifically, we introduced a 16.95 wt.% SNPs solution into the mechano-FL sensor system and extended the injection time to 510 s. This extended injection time yielded a normalized FL intensity approaching 1, indicating near saturation (Supplementary Fig. 9). This

62

[Type here]

result demonstrates that the concentration conditions we employed were optimized for observing changes in FL intensity relative to the injection time during a moderate period. Additionally, we performed experiments to evaluate changes in FL intensity for PDA particles under three different SNPs concentrations (5.65, 11.3, and 16.95 wt.%). While maintaining both the injection rate and suction rate at 25 mL·min⁻¹ and 5 mL·min⁻¹, respectively, we observed that the FL intensity value increased rapidly with time at higher concentrations. In contrast, the change was relatively minor at lower concentrations (Supplementary Fig. 8a).

When we re-plotted the time-dependent FL intensity graph against the number of injected particles, the data for the three different concentration conditions converged to a single master curve (Supplementary Fig. 8b), demonstrating that a short injection time with a high concentration of SNPs could yield a FL response equivalent to that obtained with a longer injection time using a smaller concentration of SNPs. Furthermore, a 16.95 wt.% SNPs solution was injected and halted at 4.5 min, and the FL intensity was continuously monitored even after stopping the injection. It was found that the change after stopping was negligible (Supplementary Fig. 10). In short, these additional experiments and analyses confirmed that the FL intensity values are directly influenced by the number of SNPs colliding with the PDA particles. We have included these findings in the revised manuscript.

Added paragraph (p13): We conducted further experiments to elucidate the effect of physical impact of SNPs on PDA particles on FL emission. The time-dependent FL response at different SNPs concentrations (5.65, 11.3, and 16.95 wt.%) was compared, as shown in Supplementary Fig. 8a. It was observed that the FL intensity value rapidly increased with time at higher concentrations, while the change was relatively minor at lower concentrations. The FL intensity change was then re-plotted against the number of injected SNPs, resulting in the data for the three different concentrations converging to a single master curve (Supplementary Fig. 8b). Additionally, we extended the injection time of 16.95 wt.% SNPs to 8.5 min (510 s) and found that the normalized FL intensity approached 1, indicating near saturation (Supplementary Fig. 9). Moreover, a 16.95 wt.% SNPs solution was injected and halted at 4.5 min (270 s), and the FL intensity was continuously monitored even after stopping the injection. It was found that the FL change after stopping was negligible (Supplementary Fig. 10). Based on these additional analyses, we confirmed that the FL intensity values are primarily influenced by the number of SNPs colliding with the PDA particles. Furthermore, the gradual changes in PDA mechanochromism, along with the halt of FL intensity changes upon the removal of mechanical stress, underscore the important role that continuous application and accumulation of mechanical stress play in mechanochromism. This might be closely related to the nonlinear behavior of PDA mechanochromism.^{53, 54} For more information on the injected fluids used in this study, one can refer to Supplementary Table 2.

Newly added Supplementary Fig. 8:

[Image redacted]

63

[Type here]

Supplementary Fig. 8 Effect of the injected SNPs concentration on the mechano-FL sensor response. a Time-dependent FL intensity changes for varying SNPs concentrations, with the injection and suction rates maintained at 25 mL·min⁻¹ and 5 mL·min⁻¹, respectively. b, FL intensity change re-plotted against the number of injected SNPs, showing that the data for the three different concentration conditions converge to a single master curve. This demonstrates that a short injection time with a high concentration of SNPs can yield a FL response equivalent to that obtained with a longer injection time using a smaller concentration of SNPs. Each FL intensity data point represents the average of a minimum of three independent trials, and the error bars indicate the corresponding standard deviations.

Newly added Supplementary Fig. 9:

[Image redacted]

64

[Type here]

Supplementary Fig. 9 Mechano-FL system response over an extended injection time. A

16.95wt.% SNP solution was injected for 8.5 min (510 s), with the injection and suction rates maintained at 25 mL·min⁻¹ and 5 mL·min⁻¹, respectively. This extended injection time yielded a normalized FL intensity approaching 1, indicating near saturation. The error bars represent the standard deviations derived from at least three independent trials. Note that this plot is the same with the profile of FL intensity versus time presented in Fig. 5f of the main text.

Newly added Supplementary Fig. 10:

[Image redacted]

Supplementary Fig. 10 Monitoring of FL intensity change of the mechano-FL sensor system over time after stopping SNPs injection. A 16.95 wt.% SNPs solution was injected and halted at 4.5 min, with the injection and suction rates maintained at 25 mL·min⁻¹ and 5 mL·min⁻¹, respectively. The FL intensity was continuously observed even after stopping the injection, and it was found to be negligibly altered. The error bars indicate the standard deviations derived from at least three independent trials.

65

[Type here]

2.4) The authors present a model where the mechanofluorochromic response is linked to the energy absorbed by the PDA particle, with the highest fluorescence response obtained at high concentration of particles and longest observation time. The authors should detail whether or not a short injection time with a high concentration of silica particles is equivalent to a long injection time with a smaller concentration of particles. This would strongly comfort the proposed model and increase the practical usefulness of such a force sensor.

This comment has been addressed in our response to this reviewer's comment 2.3 (specifically, Supplementary Fig. 8).

2.5) What would be the detection limit of the proposed force sensor? And does the optimal response presented by the authors correspond to a saturation limit of the sensor (in other words does the fluorescence signal stay similar when higher mechanical stress is applied to the PDA particle)?

The detection limit of the mechano-FL sensor can be deduced from Supplementary Fig. 9, where the maximum FL intensity of the PDA particle is normalized within the range of 0 to 1. This range represents the appropriate stress required to induce a measurable change in the FL intensity of the PDA particle, leading up to the point of FL saturation. Regarding the saturation limit, it is important to note that the FL intensity would not exhibit significant changes after the normalized FL intensity reaches 1. This is because the PDA response occurs irreversibly, meaning that beyond this point, additional stresses do not result in a substantial change in the FL response. This characteristic defines the sensor's saturation limit.

Before: This thermally induced-FL response matches that observed in a previous study where a PDA

After (p7): Note that the FL intensity barely exhibit significant changes after the normalized FL intensity reaches ~1. This is because the PDA response occurs irreversibly, meaning that beyond this point, additional stresses unlikely result in a substantial change in the FL response. The observed thermally induced-FL response matches that observed in a previous study where a PDA

2.6) The authors notice a fluorescence response after simple incubation of the PDA particle with various silica nanoparticle and justify this by a sensitivity to pH. I am very surprised that pentacosadiynoic acid can be sensitive to pH variations between 8.2 and 9.1 since it is a carboxylic acid completely on its carboxylate form in this pH range. Could the authors explain

further?

66

[Type here]

The FL response did not show a significant difference or any consistent trend depending on the measured pH values after simply incubating the PDA particles with various SNPs dispersed solutions. Importantly, as seen in Supplementary Table 1, especially in the buffer environment we used for the mechano-FL sensor experiments, the FL intensity values for all SNPs cases were approximately in the range of 0.0098 to 0.0132 a.u. These values were sufficiently minor. The reason we chose to use LUDOX TM-40 was that its FL intensity was relatively the lowest in the buffer environment, although using other types of SNPs would not have resulted in any significant changes in the mechano-FL sensor experiment. We have removed the column showing the pH values for each solution to eliminate any potential for unnecessary confusion.

Before: As a result, in the optimized vascular mimic system we used a 20 mM sodium phosphate buffer to mitigate the pH effect (Fig. S4c and Table S1) and LUDOX TM-40 (diameter ~34 nm), which displays a minimum FL intensity response upon static incubation.

After (p8): The use of a 20 mM phosphate buffer significantly reduced the FL intensity of the PDA particles, with values in the range of 0.0098 to 0.0132 a.u. for all types of SNPs, which were sufficiently minor (Supplementary Fig. 6c and Supplementary Table 1). Consequently, we chose to use LUDOX TM-40 (diameter $\sim 19.6 \text{ \AA} \pm 5.6 \text{ nm}$, Supplementary Fig. 7), which displayed the relatively lowest FL intensity in the buffer environment. However, the use of other types of SNPs would not have resulted in any significant differences in the mechano-FL sensor experiment. Additionally, before proceeding with the mechano-FL sensor experiments, we confirmed that all the solutions used in the experiments did not induce significant changes in the FL emission of the PDA particles under static conditions.

Revised Supplementary Table 1:

Supplementary Table 1. Effect of various SNP suspensions on the FL response of PDA particles upon simple incubation.

[Table redacted]

2.7) About figure 3: in the text p10 1191 the authors write: "local indentations are present mainly at the center region of the PDA particle" and "complete PDA particle fluoresces in the red region after injection of the SNPs for 4.5 min, demonstrating the applied mechanical stress propagates throughout the entire particle." Is this propagation of mechanical stress instantaneous or is it possible to observe transient states where only the central region of the

67
[Type here]

particle is fluorescent? Such a mechanical stress propagation is described in Langmuir 2000, 16, 1270-1278 (reference which should be cited in the manuscript) however it is not instantaneous. Could the authors discuss potential differences between their work and the literature results on this specific point?

We thank the reviewer for pointing out the important aspect of mechanical stress propagation. To clarify this point, we subjected a PDA particle to a collision by injecting 16.95 wt.% SNPs for 4.5 min, and then we stopped the injection and continued to monitor any changes in the FL intensity. As shown in Supplementary Fig. 10, the changes in FL intensity observed post-collision were negligible. Compared to the complete transition after an extended collision lasting 8.5 min (Fig. 5f and Supplementary Fig. 9), we surmise that for the energy applied locally to a PDA particle to spread across the entire region, it would necessitate a consistent and accumulating external energy input.

In our original manuscript, based on experimental data such as the observation of gradual FL changes in PDA particles over time in our mechano-FL sensor experiments (Fig. 4c), FL

emission from entire particles under prolonged stress exposure (Fig. 5f), and the SEM image analysis of an isolated PDA particle post-SNPs collision showing local indentations focused on the center region of the particle (Fig. 3e-g), we inferred that the FL in entire PDA particles might originate from localized mechanical stress. This has led us to speculate that mechanical stress could propagate in a finite manner. Nonetheless, a more in-depth study is needed to explore the intricacies of this propagation process. For this, we are planning the following subsequent studies:

- (1) We plan to utilize optical laser tweezers apparatus [Nat. Commun. 14, 3838, 2023; J. Phys. Chem. Lett. 13, 10018, 2022]. This can optically trap a polymeric microparticle and apply oscillatory impacts to the surface of a PDA particle anchored to a substrate. Given that optical laser tweezers can precisely control the applied force down to the piconewton scale, we believe that sufficient mechanical stress can be exerted on a thin-layered PDA particle.
- (2) We intend to induce a single collision of a PDA particle (or other types of particles) to another PDA particle positioned within the mechano-FL sensor channel. By controlling the size, rigidity, and collision speed of the colliding particles, we aim to compare the area where FL is emitted to the actual collision area.

In light of these considerations, we have revised the manuscript and also updated the explanation part of Fig. 4.

Before: Finally, the confocal image given in the inset in Fig. 3f shows that the complete PDA particle fluoresces in the red region after injection of the SNPs for 4.5 min, demonstrating the applied mechanical stress propagates throughout the entire particle.

After (p12): Finally, the confocal image shown in the inset of Fig. 3e reveals that the entire surface region of the PDA particle fluoresces after injection of the SNPs for 4.5 min. This

68
[Type here]

illustrates how the localized mechanical stress, when continuously applied and accumulated, can affect the entire region of the PDA particle.

Before: The time-dependence of the response of the new sensor system was examined by continuously monitoring FL intensities at times following their injection (Fig. 3e and Fig. 4). We found that as the flow injection time increases the consequent FL intensity increases. For example, in contrast to the negligible effect caused by pure water, injection of 16.95 wt.% SNP promotes the most prominent time dependent FL intensity increases. Interestingly, as the SNP injection time is increased, the area over which FL emission occurs gradually increases (Fig. 4e). Considering that the location of the impact of the SNPs remains constant throughout the injection time period, this phenomenon is likely the result of mechanical stress propagation. Notably, using the 16.95 wt.% SNP solution, the FL intensity increases nonlinearly, and two sharp FL increases occur using injection times of ~1 min and ~4 min (Fig. 3e).

After (p12): The time-dependent response of the mechano-FL sensor system was examined by continuously monitoring FL intensities at various time points following injection (Fig. 3h and Fig. 4). We found that as the flow injection time increases the consequent FL intensity increases. For example, in contrast to the negligible effect caused by pure water (Fig. 4a) and relatively small FL intensity changes for the case of 20 wt.% glycerin (Fig. 4b), injection of 16.95 wt.% SNP promotes the most prominent time dependent FL intensity increases. Interestingly, as the SNP injection time is increased, the area of FL emission also gradually increases (Fig. 4c). Given that the impact location of SNPs remains constant throughout the injection, the continuous SNPs collision likely results in accumulated mechanical stress, causing the entire particle to fluorescence. Notably, using the 16.95 wt.% SNP solution, the FL intensity increases nonlinearly (Fig. 3h).

Added paragraph (p13): We conducted further experiments to elucidate the effect of physical impact of SNPs on PDA particles on FL emission. The time-dependent FL response at different SNPs concentrations (5.65, 11.3, and 16.95 wt.%) was compared, as shown in Supplementary Fig. 8a. It was observed that the FL intensity value rapidly increased with time at higher concentrations, while the change was relatively minor at lower concentrations. The FL intensity

change was then re-plotted against the number of injected SNPs, resulting in the data for the three different concentrations converging to a single master curve (Supplementary Fig. 8b). Additionally, we extended the injection time of 16.95 wt.% SNPs to 8.5 min (510 s) and found that the normalized FL intensity approached 1, indicating near saturation (Supplementary Fig. 9). Moreover, a 16.95 wt.% SNPs solution was injected and halted at 4.5 min (270 s), and the FL intensity was continuously monitored even after stopping the injection. It was found that the FL change after stopping was negligible (Supplementary Fig. 10). Based on these additional analyses, we confirmed that the FL intensity values are primarily influenced by the number of SNPs colliding with the PDA particles. Furthermore, the gradual changes in PDA mechanochromism, along with the halt of FL intensity changes upon the removal of mechanical stress, underscore the important role that continuous application and accumulation of mechanical stress play in mechanochromism. This might be closely related to the nonlinear

69

[Type here]

behavior of PDA mechanochromism.^{53, 54} For more information on the injected fluids used in this study, one can refer to Supplementary Table 2.

Newly added Supplementary Fig. 10:

[Image redacted]

Supplementary Fig. 10 Monitoring of FL intensity change of the mechano-FL sensor system over time after stopping SNPs injection. A 16.95 wt.% SNPs solution was injected and halted at 4.5 min, with the injection and suction rates maintained at 25 mL·min⁻¹ and 5 mL·min⁻¹, respectively. The FL intensity was continuously observed even after stopping the injection, and it was found to be negligibly altered. The error bars indicate the standard deviations derived from at least three independent trials.

2.8) Still about the propagation of mechanical stress, have the authors tried to flow a solution with a high concentration of silica nanoparticles for a short time followed by pure water for a longer time to check how this could influence the mechanical stress detection? If there is stress propagation from the central zone of the PDA particle to the entire particle, this could be highly beneficial for the sensitivity of the device, but this could also strongly perturb the force quantification. I would like the authors to further comment on this point.

We refer this reviewer to our response to the previous comment-2.7 of this reviewer.

2.9) Lastly, if I understand correctly figures 1e and 2e the fluorescence enhancement upon thermal stimulus is quantified between 0 and 1 arbitrary units while the fluorescence enhancement due to the mechanical stimulus is quantified between 0 and 0.3 arbitrary units. I suppose the arbitrary units are the same for the two experiments, and I wonder why if the PDA particle is entirely converted to the red phase after mechanical stimulation the fluorescence enhancement is limited to 0.3. Could the authors explain further?

We have provided a more detailed procedure for normalizing the FL intensity values in Supplementary Information. We also showed that an extended exposure of SNPs to a PDA particle can result in the normalized FL intensity approaching 1 (Supplementary Fig. 9).

70

[Type here]

Before:

7. Mechano-FL sensing data acquisition

Each FL image was analyzed to obtain the red color intensity value x using the following equation,

$\delta\epsilon = 0.4124 \ 0.3576 \ 0.1805 \ \delta$
 $[\delta] = [0.2126 \ 0.7152 \ 0.0722] [\delta^\circ]$.
 $\delta\epsilon = 0.0193 \ 0.1192 \ 0.9505 \ \delta\mu$

The difference in the red color intensities $\delta\epsilon$ before and after applying stresses to the PDA particle corresponded to the FL intensity.

After (p5 in SI): To quantify the FL values, each acquired FL image was processed to separate its color components (red, green, and blue (RGB)) using the "Splitting Channels" function in ImageJ software.³ During this process, the color of each pixel, determined by the intensity combination across the RGB channels, was separated into individual grayscale images representing each color channel. In these grayscale images, pixel intensity corresponds to the strength of the respective color in the original image. Subsequently, the mean gray value of the red channel was acquired and normalized within a range of 0 to 1 using the maximum intensity value of 255. In the mechano-FL sensor experiments, images obtained from different PDA particles under identical experimental conditions were processed in the same manner. Note that because the 570–700 nm emission filter was used, any signals from the green and blue channels were barely detectable. The normalized red intensity values derived from the multiple measurements were averaged, and the resulting mean values were used for further data analysis to ensure consistency.

2.10) Could the authors give information about how many times the experiment was repeated for a given condition and give some details on how they calculated the error bars in figures 3 and 5?

For each experimental condition, we performed a minimum of three independent trials to ensure the reliability and reproducibility of our results. From each trial, we derived the normalized FL intensity value using the image analysis protocol. The displayed values represent the average, and the error bars denote the standard deviation.

Added statement (Caption of Fig. 3): All FL experiments were performed independently at least three times. The data points and error bars in panels a-d and h indicate the mean intensity values and standard deviations, respectively.

Added statement (Caption of Fig. 5): For panels d-f, each experimental data point and corresponding error bar represent the mean intensity value of a minimum of three independent

71
[Type here]

trials and its standard deviation, respectively.

Before (SI): For all FL measurements, the camera sensitivity and exposure time were set to ISO 100 and 0.4 s, respectively.

After (p4 in SI): Additionally, the PDA particles were only exposed to the FL laser during the FL measurements, and the DSLR camera settings were kept consistent (i.e., ISO 100 for camera sensitivity and 0.4 s exposure time) during the capturing of FL images. Note that each PDA particle was utilized only once for the mechano-FL sensor experiments; no reused particles were incorporated into any subsequent FL sensor experiments. All the FL measurements were conducted independently at least three times. The mean intensity values, along with their respective standard deviations, were reported in the main text and Supplementary Information. By maintaining these stringent conditions, we ensured the minimization of any potential errors from FL measurements.

3- About the model used:

3.1) Could the authors give the numerical values of the Reynolds number in the system studied and justify the use of a turbulence model? Additionally, if the authors could provide numerical values for all the parameters they use in table S6, it would help the reader understand the model used.

We have included the Reynolds numbers for our system in Supplementary Table 2 and the parameter values associated with the Hertz contact theory equations in Supplementary Table 7. For the Reynolds numbers, we tabulated the values by converting the flow rate injected into the mechano-FL sensor to the flow velocity. Note that the flow velocity around the PDA particle, as depicted in Fig. 5a, is on a similar scale to the injected velocity.

Regarding our choice of the turbulence model, we have updated the manuscript to provide a more comprehensive justification for our selection of the $k\text{-}\bar{\epsilon}$ SST model. Briefly, the $k\text{-}\bar{\epsilon}$ SST model combines the strengths of the low Reynolds $k\text{-}\bar{\mu}$ turbulence model in free-stream flows with the benefits of the $k\text{-}\bar{\epsilon}$ turbulence model near walls. This approach can address certain limitations of both the pure $k\text{-}\bar{\epsilon}$ and $k\text{-}\bar{\mu}$ models, resulting in enhanced accuracy and stability over a wide range of flow conditions (<https://doi.org/10.2514/6.1993-2906>; <https://www.comsol.com/blogs/which-turbulence-model-should-choose-cfd-application/>).

72

[Type here]

Newly added Supplementary Table 2:

Supplementary Table 2. Flow information of the injected fluids into the mechano-FL sensor system.

[Table redacted]

Revised Supplementary Table 7 (Table S6 in the original manuscript):

Supplementary Table 7. Summary of Hertz contact theory equations, dimensionless numbers, and parameter values for a single SNP collision to PDA particle.

[Table redacted]

Before:

8. Numerical simulation

The CFD simulation (Fig. S6) of the mechano-FL sensor system was carried out using the finite volume method (FVM) with Ansys-Fluent (Academic Workbench 2021R2). The effects of injection flow rate, viscosity, and SNP concentration on the absorbed energy by the PDA particle were evaluated. The computational domain is subdivided into control volumes, and differential equations are integrated to produce a set of algebraic equations. These approaches are intrinsically conservative and may describe various arbitrary geometries.⁶ The ANSYS meshing tool was used to create meshes for the presented control volumes. The unstructured triangular/tetrahedral hybrid mesh elements were chosen in this study due to their flexibility and capacity to cope with complex geometries. For simplicity, the PDA particle was fixed at the holding tube without introducing the negative pressure on it. The governing equations for numerical analyses of fluid flows and SNPs motions inside the microchannel were solved using the Ansys-Fluent FVM CFD codes. To reduce the computational cost but keep the accuracy on a reasonable level, the Eulerian approach with the $k\text{-}\bar{\epsilon}$ SST turbulence model was implemented for the steady-state incompressible Newtonian fluid flow, in which pressure and velocity distributions were determined via mass and momentum conservation equations.^{7, 8}

After (p22 in SI):

Supplementary Note 1: Numerical simulation

The CFD simulation (Supplementary Fig. 12) of the mechano-FL sensor system was carried out using the widely used finite volume method (FVM) with Ansys-Fluent (Academic Workbench 2021R2) due to its versatility, robustness, and accuracy. We refrained from using the meshless method due to potential challenges such as convergence and stability issues, difficulty in enforcing boundary conditions, and high computational costs for large numbers of nodes. We evaluated the effects of injection flow rate, viscosity, and SNP concentration on the absorbed energy by the PDA particle. The computational domain is subdivided into control volumes, and differential equations are integrated to produce a set of algebraic equations. These approaches are intrinsically conservative and can describe various arbitrary geometries.⁶ The ANSYS meshing tool was used to create meshes for the presented control volumes. The unstructured triangular/tetrahedral hybrid mesh elements were chosen in this study due to their flexibility and capacity to cope with complex geometries. For simplicity, the PDA particle was

fixed at the holding tube without introducing the negative pressure on it. The governing equations for numerical analyses of fluid flows and SNPs motions inside the microchannel were solved using the Ansys-Fluent FVM CFD codes. To reduce the computational cost but maintain reasonable accuracy, the Eulerian approach with the $k\text{-}\bar{\epsilon}$ SST turbulence model was implemented for the steady-state incompressible Newtonian fluid flow, in which pressure and velocity distributions were determined via mass and momentum conservation equations.^{7, 8} The $k\text{-}\bar{\epsilon}$ SST model combines the strengths of the low Reynolds $k\text{-}\hat{\mu}$ turbulence model in free-stream flows with the benefits of the $k\text{-}\bar{\epsilon}$ turbulence model near walls. This approach can

74

[Type here]

address certain limitations of both the pure $k\text{-}\bar{\epsilon}$ and $k\text{-}\hat{\mu}$ models, resulting in enhanced accuracy and stability over a wide range of flow conditions.^{9, 10}

3.2) I cannot find details on how E_{torsion} is calculated (values of table S7 and table S8). Could the authors give more details? If these values are directly taken from reference 47 then it is a huge approximation because the polydiacetylene studied are different.

In an effort to quantitatively predict the nonlinear behavior of the FL intensity acquired through our mechano-FL sensor experiments, we referred to previously reported studies [J. Am. Chem. Soc. 132, 13313, 2010]. However, as pointed out by this reviewer, we agree that utilizing the torsion energy values from different types of PDA can introduce unnecessary errors. As a result, the quantitative calculation parts were removed from the main text and SI. Nevertheless, the nonlinear increase of torsion energy in relation to the torsion angle shows a resemblance to the FL intensity profiles we experimentally obtained (as depicted in the inset of Fig. 5f). This similarity allows us to reasonably postulate a close correlation between these two factors. Furthermore, activation energy values for the complete blue-to-red transition, as reported for various types and conditions of PDA materials, are around 17.6 to 22.5 kcal/mol [Carpick et al., Langmuir, 16, 4639, 2000]. Considering the ~27% polymerization of our experimentally derived PDA particles, the activation energy required for the PDA particle's blue-to-red transition is estimated to be ~ 289 to 369 $\hat{\mu}$ J. This result aligns well with the energy requirement of 307 $\hat{\mu}$ J predicted via our calculations. We have updated the manuscript to emphasize and clarify these points.

Before: The calculations show that bombardment by the SNPs leads to an increase in δ_{torsion} absorbed by the PDA particle, considered to be the sum of the computed δ_{torsion} and SNP collision energy $\delta_{\text{collision}}$ calculated using the Hertz contact equation (Table S6).⁵⁰ As shown in Fig. 5e, δ_{torsion} increases in a linear manner upon increasing the concentration of the SNP in the 0, 5.65, 11.3 and 16.95 wt.% range injected for 4.5 min. Similarly, because the conjugated structure of the PDA was not considered in this simulation, the δ_{torsion} values do not accurately reflect the observed nonlinear SNP concentration dependent increase in FL intensities. However, the sudden increase in the FL intensity observed in 11.3 to 16.95 wt.% region suggests that a certain critical torsion energy threshold exists between 33.75 and 50.73 $\hat{\mu}$ J.

To determine the critical torsion energy level, δ_{torsion} values were calculated as a function of the SNP injection time and then compared to the torsion energy δ_{torsion} (Tables S7 and S8).^{46, 47} The calculated variation in torsion energy as a function of the δ_{torsion} and the SNP injection time (t) show that two rapid increases in torsion energy take place in the time intervals of 60 s to 120 s and 210 s to 270 s, as indicated by the orange bars in Fig. 5f. The corresponding torsion angles are 11° to 22° and 39° to 50°, which are consistent with PDA polymorphs displaying most prominent FL emission.⁴⁶ The calculated nonlinear δ_{torsion} profile is in good agreement with the experimentally observed FL intensity profile, demonstrating the role played by the unique structural characteristics of the PDA particle in determining FL emission. In addition, because the δ_{torsion} value obtained from the CFD simulation is the same order of magnitude as δ_{torsion} , the range of ~40 to 50 $\hat{\mu}$ J is estimated

75

[Type here]

to be the critical energy level, around which the PDA particle undergoes a notable FL intensity increase.

After (p18): The calculations show that bombardment by SNPs leads to an increase in δ_{tot} absorbed by the PDA particle, considered to be the sum of the computed δ_H and SNP collision energy δ_{col} calculated using the Hertz contact equation (Supplementary Table 7).⁶⁰ δ_{tot} demonstrates an increasing trend as the SNP concentration increases, spanning 0, 5.65, 11.3, and 16.95 wt.% when injected for 4.5 min (Fig. 5e) and with an extended injection duration of 8.5 min at 16.95 wt.% concentration (Fig. 5f). Similarly, because the conjugated structure of the PDA was not considered in this simulation, the δ_{tot} values do not accurately reflect the observed nonlinear behavior of FL intensities dependent on the SNP concentration.

Nevertheless, this computational data suggests that the δ_{tot} required to achieve a complete blue-to-red transition for a single PDA particle is approximately 307 μJ . The activation energy δ_{act} for irreversible blue-to-red transitions in a variety of PDAs has been estimated to be $\sim 17.6\text{--}22.5 \text{ kcal}\cdot\text{mol}^{-1}$.⁴⁴ For instance, for a three-layer PDA film fabricated from the PCDA monomers, the predicted δ_{act} was approximately $17.6 \text{ kcal}\cdot\text{mol}^{-1}$. Though our 200 μm PDA particles might differ from the previously reported PDA conditions, we presumed $\delta_{act} \sim 17.6\text{--}22.5 \text{ kcal}\cdot\text{mol}^{-1}$ to roughly estimate δ_{act} for our PDA particles. Considering $\sim 27\%$

δ_{act} PDA

polymerization (Fig. 2d) yields 0.27 μm^3 PDA $\delta_{act} = 289\text{--}369 \mu\text{J}$, where the PDA density

ρ_{PCDA}

$3 \times 10^3 \text{ kg}\cdot\text{m}^{-3}$

is $\delta_{PDA} \sim 1.3 \times 10^{-10} \text{ kg}\cdot\text{m}^{-3}$, the PDA volume is $\delta_{PDA} = 3 \times 10^{-10} \text{ m}^3$, and the molecular weight of PCDA is $\delta_{PCDA} = 0.375 \text{ kg}\cdot\text{mol}^{-1}$. This result aligns well with the energy requirement of $\sim 307 \mu\text{J}$ for the complete blue-to-red transition, as derived from our calculations.

The nonlinear FL intensity profile possesses a resemblance to the previously reported torsional energy δ_{tor} calculations, which also exhibit nonlinear increases for various types of PDA materials.^{53, 54} Therefore, we compared the normalized δ_{tor} profile with the normalized FL intensity profiles derived from both the mechano-FL sensor experiment (Fig. 5f) and the thermal stress analysis (Fig. 2g). As displayed in the inset of Fig. 5f, the experimental FL intensity's nonlinear trajectory seems fairly consistent with the shape of δ_{tor} . This similarity allows us to reasonably postulate a close correlation between these two factors. Notably, recent experimental results have demonstrated that lateral stress applied parallel to the PDA backbone significantly influences the mechanochromism of thin PDA films.⁶¹ Therefore, further experimental and/or theoretical studies are needed to clarify how torsional and lateral stresses contribute relative to each other depending on the PDA structures under various conditions.

Revised Fig. 5f:

76

[Type here]

[Image redacted]

Fig. 5 CFD analysis of the mechano-FL sensor system. f Time-dependent profiles of δ_{tot} and FL intensity for the 16.95 wt.% SNP suspension injected at $25 \text{ mL}\cdot\text{min}^{-1}$ over an extended duration of 8.5 min. Inset indicates the comparison of the nonlinear property of normalized FL intensity profiles from the mechano-FL sensor experiment and the thermal stress analysis (from Fig. 2g) against the normalized torsion energy δ_{tor} from a previous study.⁵⁴ The x-axis is normalized from 0 to 1, aligning maximum values of FL intensity and δ_{tor} to 1. For panels d-f, each experimental data point and corresponding error bar represent the mean intensity value of a minimum of three independent trials and its standard deviation, respectively.

3.3) Could the authors provide data, either from their experiment or from the literature, about which torsion angle should be reached on PCDA to observe the blue to red colour change? The data from reference 46 are on polyTHD, which is a completely different diacetylene while the side chain interactions are likely to play a significant role in the torsion angle values.

Additionally, it is described in reference 46 that a PDA chain with a torsion angle of 4° is blue while a PDA chain with a torsion angle of 14° is red, which would be completely inconsistent with the results of this manuscript where the most significant fluorescence change is observed for torsion angle between 40° and 50° .

This comment has been addressed in our response to this reviewer's comment-3.2.

3.4) Could the authors discuss on the reversibility of the torsion at the scale of one PDA chain? If the torsion is fully reversible then a long experiment time with a little number of collisions on the PDA particle would be equivalent to a short experiment time with many collisions. Are there critical torsion angles beyond which the torsion is not reversible? Is there some thermal relaxation? It would be interesting to know the shape of the function $E_{tors}(\hat{l}_t)$ as a function of \hat{l}_t in this respect.

77

[Type here]

This comment has been addressed previously in our response to this reviewer's comment-3.2. Regarding the relationship between a long injection time with a few collisions and a short injection time with many collisions, both conditions are indeed equivalent. We refer the reviewer to Supplementary Fig. 8 and our earlier response to this reviewer's comment-2.3 for further clarification.

3.4.1) Lastly, I am not convinced by the last sentence of the abstract "We anticipate that this study will provide an important foundation for designing minimally invasive mechanical stress measurement systems that can be placed in small and difficult-to-access areas deep within the human body." While I fully acknowledge the originality of the presented results, I do not see how the system described in this manuscript can be implemented within the human body (furthermore I am not aware of studies on the potential toxicity of PDA particles). Could the authors comment and potentially modify the abstract accordingly?

We appreciate the reviewer's insightful comments and questions regarding the potential applications and translatability of our mechano-FL sensor system for measuring mechanical stress within the human body. In fact, our system was initially designed to serve as an in vitro platform that simulates specific problem areas within the human body, and it also facilitates the visualization and quantification of mechanical forces applied to these regions. At present, the direct implantation of the mechano-FL sensor system into the human body may pose challenges. However, we believe that these challenges could potentially be overcome by coating the PDA particles with biocompatible materials, making in vivo applications not entirely out of reach. To avoid any misunderstandings and to clarify this point, we have revised the relevant section as suggested by the reviewer.

Before: We anticipate that this study will provide an important foundation for designing minimally invasive mechanical stress measurement systems that can be placed in small and difficult-to-access areas deep within the human body.

After (Abstract): We believe this study offers a unique advantage for simulating specific problematic regions of the human body in an in vitro environment, potentially paving the way for future exploration of difficult-to-access areas within the body.

Before (Discussion): This study represents a significant advancement in the development of minimally invasive force/stress measurement systems that can be located in hard-to-reach small spaces within the human body, including implantable vascular prostheses, blood cell trauma caused by stenosis, and flow-induced thrombogenicity.

After (p18, Discussion): We believe this study represents a significant advancement in the development of minimally invasive force/stress measurement in vitro systems that simulate hard-to-reach spaces within the human body.

78

Version 1:

Reviewer comments:

Reviewer #1

(Remarks to the Author)

The authors have responded to most of my queries satisfactorily and the manuscript (MS) has benefited from all the changes made in the various sections of the MS.

However the authors should revise the values of the RBCs mean size at the Supplementary Table 3. "Properties of the injected materials into the mechano-FL sensor system."

RBCs mean size of 4.3um is extremely small and far from the well known size found in the literature. Please have a look, for instance, to these papers.

<https://doi.org/10.1007/s00348-020-03066-7>

<https://www.sciencedirect.com/science/article/pii/S2590007222000466?via=ihub>

<https://doi.org/10.1039/D1SM00106J>

The authors should revise and compare with the literature, the values of the RBCs mean size from table 3 and change/explain them accordingly before the acceptance of this manuscript.

Reviewer #2

(Remarks to the Author)

The authors have done an admirable job addressing the numerous reviewer comments. The most interesting aspects to this individual reviewer are the preparation method and the consistency of fluorescent intensity with stress (e.g. Fig 3h, 5def, S8ab). All measurements demonstrate the same unusual monotonic increase. This builds confidence in the "sensor".

There is one crucial point that this reviewer believes must still be addressed.

The mapping of the energy for the blue to red transition to a single condition is a significant weakness (Note the authors incorrectly cite Carpick et al. Langmuir 2000. It should be Caprick et al. 2004 DOI 10.1088/0953-8984/16/23/R01). The single condition (200 micron particles 27% cross-linked) agrees with Carpick et al. is used repeatedly by the authors to justify that they are correct enough in their CFD analysis that they understand and can predict the energy deposited or the shear stress exposure of the PDA particle. The easiest test to clearly demonstrate the utility of the sensor would then be to use a different sized PDA particle and see the same agreement at this new energy deposition requirement to drive the blue to red transition. However, this will require new CFD calculations. Conversely, one could decrease or increase the polymerization, which may or may not be easy. This reviewer strongly urges a demonstration with a simple change and mapping the energy deposited to the observed fluorescence is necessary to demonstrate the conclusions. The change in size to smaller PDA particles will also bolster statements like "We believe this study offers a unique advantage for simulating specific problematic regions of the human body in an in vitro environment, potentially paving the way for future exploration of difficult-to-access areas within the body."

Reviewer #3

(Remarks to the Author)

The authors were able to significantly improve the quality of the manuscript: the protocols for fabrication, simulation, experiments, and data analysis are clearer and more elaborate. Additional experiments like bisecting the particle and imaging before/after chloroform treatment reveal the composition of these particles. Raman spectroscopy after SNP bombardment, long-term exposure, and extended time-dependent data clarify the sensor mechanism and response better. Finally, the authors rightfully toned down the potential of this technique for in vivo applications, but Supplementary Fig. 14 at least visualizes a potential scenario to simulate vessel-mimicking capillary channels in vitro more realistically.

The revised manuscript claims that PDA particles have a polymerized shell, with a significant amount of PCDA monomers inside. However, Raman spectra of crushed PDA particle in Fig. 2h(iv) do not contain PCDA signal at all, why?

Minor typos:

Fig. 2h: Raman sifts

Supplementary s3: Lecia DM 2700

Reviewer #5

(Remarks to the Author)

The authors have done comprehensive work to address the reviewers' concerns. The manuscript is in a good shape to be accepted.

Author Rebuttal letter:

Response to the Reviewer's Comments

REVIEWER COMMENTS

Reviewer #1 (Remarks to the Author):

The authors have responded to most of my queries satisfactorily and the manuscript (MS) has benefited from all the changes made in the various sections of the MS.

However the authors should revise the values of the RBCs mean size at the Supplementary

Table 3. "Properties of the injected materials into the mechano-FL sensor system."

RBCs mean size of 4.3 μ m is extremely small and far from the well known size found in the literature. Please have a look, for instance, to these papers.

[1*] <https://doi.org/10.1007/s00348-020-03066-7>

[2*] <https://www.sciencedirect.com/science/article/pii/S2590007222000466?via=ihub>

[3*] <https://doi.org/10.1039/D1SM00106J>

The authors should revise and compare with the literature, the values of the RBCs mean size from table 3 and change/explain them accordingly before the acceptance of this manuscript.

We thank the reviewer for the positive feedback on our revised manuscript.

Regarding the concern raised about the red blood cells (RBCs) mean size, please note that we used sheep RBCs (~4 μ m) instead of human ones (~7 μ m). This choice was due to the greater accessibility of sheep RBCs in our country. We have mentioned this in the material section (p2 in SI: "Defibrinated sheep blood was purchased from Biozoa Biological Supply (Korea)," and also in the main text (p14: "For these experiments, we used ovine RBCs because they are commercially available,").

Reviewer #2 (Remarks to the Author):

The authors have done an admirable job addressing the numerous reviewer comments. The most interesting aspects to this individual reviewer are the preparation method and the consistency of fluorescent intensity with stress (e.g. Fig 3h, 5def, S8ab). All measurements demonstrate the same unusual monotonic increase. This builds confidence in the "sensor".

There is one crucial point that this reviewer believes must still be addressed.

The mapping of the energy for the blue to red transition to a single condition is a significant weakness (Note the authors incorrectly cite Carpick et al. Langmuir 2000. It should be Caprick et al. 2004 DOI 10.1088/0953-8984/16/23/R01). The single condition (200 micron particles 27% cross-linked) agrees with Carpick et al. is used repeatedly by the authors to justify that they are correct enough in their CFD analysis that they understand and can predict the energy deposited or the shear stress exposure of the PDA particle. The easiest test to clearly demonstrate the utility of the sensor would then be to use a different sized PDA particle and see the same agreement at this new energy deposition requirement to drive the blue to red [Type here]

transition. However, this will require new CFD calculations. Conversely, one could decrease or increase the polymerization, which may or may not be easy. This reviewer strongly urges a demonstration with a simple change and mapping the energy deposited to the observed fluorescence is necessary to demonstrate the conclusions. The change in size to smaller PDA particles will also bolster statements like "We believe this study offers a unique advantage for simulating specific problematic regions of the human body in an in vitro environment, potentially paving the way for future exploration of difficult-to-access areas within the body."

We thank the reviewer for the positive feedback on our efforts to address the previous comments.

Regarding the citation issue, we have included "Carpick et al. 2004" in the revised manuscript.

In response to the request for a comparative analysis of the energy required for blue-to-red transition in PDA particles of varying sizes, we performed additional CFD calculations for PDA particles approximately 50 and 140 μ m in size. The results, as depicted in Supplementary Fig. 14, indicate polymerization degrees of ~74% for the 50 μ m PDA and ~37% for the 140 μ m PDA. Considering these findings and the literature values of δ_{act} 17.6-22.5 kcal \cdot mol $^{-1}$, the energy required ranges of 12.7-16.3 μ J for the 50 μ m PDA and 136.7-174.7 μ J for the 140 μ m PDA. These values are consistent with the CFD-predicted δ_{tot} of ~12.4 μ J and ~141 μ J, respectively. We have added these details into the revised manuscript.

Newly added Supplementary Fig. 15:

[Type here]

[Image redacted]

Supplementary Fig. 14 Effect of PDA particle size on energy requirements for blue-to-red transition. a Representative confocal microscopic images illustrating the thickness of the polymerized shell of PDA particles. b Comparative analysis of δ_{tot} values derived from CFD simulations against estimations based on literature-reported activation energy, δ_{act} $\approx 17.6 \pm 22.5$ kcal \cdot mol $^{-1}$. The purple and pink shades depict the upper and lower boundaries of these literature values, respectively. The inset graph demonstrates the relationship between PDA particle size and both the polymerized shell thickness and the degree of polymerization. The error bars indicate the standard deviations derived from three independent measurements.

Added statement (p19): Additionally, we performed CFD calculations for different sizes of PDA particles and verified that the δ_{tot} values aligned with the estimates derived from the δ_{act} values, taking into account the degree of polymerization and particle size as shown in Supplementary Fig. 14.

Reviewer #3 (Remarks to the Author):

The authors were able to significantly improve the quality of the manuscript: the protocols for fabrication, simulation, experiments, and data analysis are clearer and more elaborate. Additional experiments like bisecting the particle and imaging before/after chloroform treatment reveal the composition of these particles. Raman spectroscopy after SNP bombardment, long-term exposure, and extended time-dependent data clarify the sensor mechanism and response better. Finally, the authors rightfully toned down the potential of this technique for in vivo applications, but Supplementary Fig. 14 at least visualizes a potential scenario to simulate vessel-mimicking capillary channels in vitro more realistically.

We appreciate the constructive feedback from the reviewer and are pleased that the revisions have enhanced the quality of the manuscript.

The revised manuscript claims that PDA particles have a polymerized shell, with a significant amount of PCDA monomers inside. However, Raman spectra of crushed PDA particle in Fig. 2h(iv) do not contain PCDA signal at all, why?

Minor typos:

Fig. 2h: Raman sifts

Supplementary s3: Lecia DM 2700

As the reviewer pointed out, the PCDA signal in the Raman spectra of the crushed PDA particle is present but varies depending on the measurement location. To maintain consistency, we replaced Fig. 2h(iv) with a spectrum that shows the PCDA monomer peak. Additionally, we have corrected the typos mentioned by the reviewer in the revised manuscript.

Revised Fig. 2h:

[Type here]

[Image redacted]

Before: λ pattern that has been observed in a variety of PDAs subjected to stresses.

After (p8): λ pattern that has been observed in a variety of PDAs subjected to stresses. Note that the PCDA monomer peak observed in the crushed sample consistently demonstrates the presence of unpolymerized PCDA within the PDA particle.

Reviewer #5 (Remarks to the Author):

The authors have done comprehensive work to address the reviewers' concerns. The manuscript is in a good shape to be accepted.

We sincerely appreciate the reviewer's positive evaluation of our manuscript.

Version 2:

Reviewer comments:

Reviewer #2

(Remarks to the Author)

The authors have not properly addressed my criticism. Instead of demonstrating that the different sized particles/polymerization levels result in quantifiable differences in the deposited energy to transition from blue to red fluorescence, the authors have done additional CFD calculations based on a single measurement in the literature by Carpick et al.

I do not agree with publishing the paper without this demonstration of the tunability and utility of the sensor.

Reviewer #3

(Remarks to the Author)

The authors have addressed all of my concerns.

Author Rebuttal letter:

Reviewer #2 (Remarks to the Author):

The authors have not properly addressed my criticism. Instead of demonstrating that the different sized particles/polymerization levels result in quantifiable differences in the deposited energy to transition from blue to red fluorescence, the authors have done additional CFD calculations based on a single measurement in the literature by Carpick et al.

I do not agree with publishing the paper without this demonstration of the tunability and utility of the sensor.

In response to the reviewer's request for a comparative analysis of the energy required for the blue-to-red transition in PDA particles of different sizes, we have complemented our previous CFD calculations with experimental studies using 50 μm and 140 μm diameter PDA particles.

Supplementary Fig. 14a,b presents previously obtained results with confocal images of polymerized shell of PDA particles and their corresponding CFD-predicted transition energies. Supplementary Fig. 14c-f depicts the new experimental FL intensity responses of varied-size PDA particles alongside their respective transition energies.

Analysis of the 50 μm PDA particles revealed a polymerization degree of $\sim 74\%$. Consequently, the estimated energy required for the transition, based on reported activation energies (δ_{act}) of $17.6\text{--}22.5 \text{ kcal}\cdot\text{mol}^{-1}$ (Carpick et al.), falls within the range of $12.7\text{--}16.3 \text{ kJ}$, that aligns closely with the CFD-predicted total energy (δ_{tot}) of $\sim 12.4 \text{ kJ}$. Similarly, the polymerization degree of the 140 μm PDA particle was determined to be $\sim 37\%$. The anticipated energy for transition, based on the same literature values for δ_{act} , spans between $136.7\text{--}174.7 \text{ kJ}$, which is consistent with the CFD-predicted δ_{tot} of $\sim 141 \text{ kJ}$. Moreover, our previous results on the 200 μm PDA particle yielded a polymerization degree of $\sim 27\%$ and an δ_{act} of $289\text{--}369 \text{ kJ}$, exhibiting an agreement with the CFD-predicted δ_{tot} of $\sim 307 \text{ kJ}$.

We have added these results into the revised manuscript.

Revised Supplementary Fig. 14:

[Type here]

[Image redacted]

Supplementary Fig. 14 Effect of PDA particle size on energy requirements for blue-to-red transition. a Representative confocal microscopic images illustrating the thickness of the polymerized shell of PDA particles. b Comparative analysis of δ_{tot} values derived from CFD simulations against estimations based on literature-reported activation energy, δ_{act} \hat{a} $17.6\text{--}22.5 \text{ kcal}\cdot\text{mol}^{-1}$. The purple and pink shades depict the upper and lower boundaries of these literature values, respectively. The inset graph demonstrates the relationship between PDA particle size and both the polymerized shell thickness and the degree of polymerization. c Normalized FL intensity response for different PDA particles sizes over the injection time. 16.95wt.% SNP solution was injected with the injection and suction rates maintained at 25 $\text{mL}\cdot\text{min}^{-1}$ and 5 $\text{mL}\cdot\text{min}^{-1}$, respectively. All error bars indicate the standard deviations derived

from three independent measurements. d-f Comparison of the experimental results in panel c with the calculated CFD-predicted transition energy in panel b.

[Type here]

Before: Additionally, we performed CFD calculations for different sizes of PDA particles and verified that the δ_{tot} values aligned with the estimates derived from the δ_{act} values, taking into account the degree of polymerization and particle size as shown in Supplementary Fig. 14.

Added statement (p19): Additionally, we performed CFD calculations and experiments for different sizes of PDA particles. Given the polymerization degree of ~74% for 50 μm and ~37% for 140 μm PDA particles (Supplementary Fig. 14a), the energy required for the complete blue-to-red transition was ~12.4 μJ and ~141 μJ , respectively. These values are in good agreement with the predictions based on δ_{act} values (Supplementary Fig. 14b). Furthermore, it was observed that the blue-to-red transition occurred more rapidly as the size of the PDA particles decreased (Supplementary Fig. 14c-f), demonstrating sensitivity tunability for PDA particles.

As mentioned by the reviewer and discussed in our Discussion section, it is anticipated that sensitivity can be adjusted by controlling the degree of polymerization. However, even without the addition of experimental data on this aspect, we believe that the research findings included in the current manuscript—namely, the fabrication of uniformly sized PDA particles using microfluidics, the study of mechanofluorescence triggered by physical collisions of nanoparticles and biomolecules in microfluidic channels, and the calculation of energy required for the blue-to-red transition using CFD—suffice to demonstrate the novelty, excellence, and broad interest of this system. Furthermore, these results will aid future research in conducting mechanofluorescent studies in more refined stenotic vessel-mimicking geometries or channels with more complex structures.

For the reviewer's information, we have recently conducted a study on the solvatochromism of several micrometer-sized individual PDA particles prepared via an emulsion method. In this study, optical laser tweezers were used to control the position of individual particles, trapping each particle dispersed in an aqueous phase and adsorbing them at an n-decane/water interface. This allowed for the investigation of solvatochromism as n-decane was absorbed by the PDA particles, revealing that the rate of the blue-to-red transition increases as the degree of polymerization decreases (Fig. R1). These research findings are currently under revision for another journal.

[Image redacted]

Fig. R1 (Reviewer Only) Influence of UV-irradiation time (δ_{UV}) on PDA solvatochromism. a [Type here]

Blue-to-red transition rate in PDA particles as a function of δ_{UV} . The inset contains the corresponding log-log plot. b SEM images of PDA particles generated using different polymerization durations.

Version 3:

Reviewer comments:

Reviewer #6

(Remarks to the Author)

In this article, the authors present a microfluidic system integrating a mechanofluorescent sensor composed of a fluorogenic single polydiacetylene (PDA) particle. The authors then demonstrate its use in the context of a stenotic vessel-mimicking capillary channel. The PDA particle is subjected to a fluid flow stress which is assessed by evaluating its fluorescence response.

This partial review concerns the CFD simulations part of the study (related to figure 14) which were performed to identify the relationship between the amount of mechanical energy absorbed by the PDA particle and the experimental FL intensity change.

Reviewer 2 has requested additional results which would serve to demonstrate a relation between the level of

polymerization/size of particles and the mechanical energy required to transition from blue to red fluorescence. This, with the aim to fully demonstrate the utility of this mechanofluorescent sensor and its capability to assess the energy absorbed or the shear stress experienced by the PDA particle at various sizes.

The authors have first responded by completing the study with CFD simulations for additional sizes (50 μm and 140 μm) and taking into account the particle size and also the degree of polymerization. Their results have shown that the E_{tot} values aligned with the literature-reported E_{act} values.

Subsequently, reviewer 2 has requested experimental tests to fully confirm these results and CFD calculations.

From my point of view, the authors have responded to this final request. They have indeed performed experimental studies using 50 μm and 140 μm diameter PDA particles. Their results show that in both cases, the anticipated energy for transition, based on the literature values for E_{act} (Caprick et al.), is in agreement with the CFD-predicted E_{tot} value.
